# FedFACT: A Provable Framework for Controllable Group-Fairness Calibration in Federated Learning

**Li Zhang**[1], **Zhongxuan Han**[1], **Xiaohua Feng**[1], **Jiaming Zhang**[1], **Yuyuan Li**[2], **Chaochao Chen**[1*]

[1]Zhejiang University, [2]Hangzhou Dianzi University

zhanglizl80@gmail.com, {zxhan, fengxiaohua, 22321350}@zju.edu.cn
y2li@hdu.edu.cn, zjuccc@zju.edu.cn

## Abstract

With the emerging application of Federated Learning (FL) in decision-making scenarios, it is imperative to regulate model fairness to prevent disparities across sensitive groups (e.g., female, male). Current research predominantly focuses on two concepts of group fairness within FL: *Global Fairness* (overall model disparity across all clients) and *Local Fairness* (the disparity within each client). However, the non-decomposable, non-differentiable nature of fairness criteria poses two fundamental, unresolved challenges for fair FL: (i) *Harmonizing global and local fairness, especially in multi-class setting*; (ii) *Enabling a controllable, optimal accuracy-fairness trade-off*. To tackle these challenges, we propose a novel controllable federated group-fairness calibration framework, named FedFACT. Fed-FACT identifies the Bayes-optimal classifiers under both global and local fairness constraints, yielding models with minimal performance decline while guaranteeing fairness. Building on the characterization of the optimal fair classifiers, we reformulate fair federated learning as a personalized cost-sensitive learning problem for in-processing and a bi-level optimization for post-processing. Theoretically, we provide convergence and generalization guarantees for FedFACT to approach the near-optimal accuracy under given fairness levels. Extensive experiments on multiple datasets across various data heterogeneity demonstrate that FedFACT consistently outperforms baselines in balancing accuracy and global-local fairness.

## 1 Introduction

Federated learning (FL) is a collaborative distributed machine learning paradigm that allows multiple clients to jointly train a shared model while preserving the privacy of their local data [55]. As FL is increasingly adopted in high-stakes domains—healthcare [80, 65, 62], finance [15, 11, 72], pattern recognition [52, 63, 94, 86, 19], and recommender systems [10, 82, 35, 73]—ensuring fairness is imperative to prevent discrimination against demographic groups based on sensitive attributes [32, 90, 88, 64], such as race, gender, age, etc. Although a rich literature addresses group fairness in centralized settings [2, 3, 41, 13], these methods depend on full access to the entire dataset and centralized processing, imposing excessive communication overhead and elevating privacy concerns when directly applied in the FL context.

To provide fairness guarantees for federated algorithms, recent works have concentrated on two group-fairness concepts in FL: *Global Fairness* and *Local Fairness* [33, 25, 95, 18, 31]. Global fairness aims to identify a model that provides similar treatment to protected groups across the entire data distribution. Local fairness concerns models that mitigate disparities and deliver unbiased predictions for sensitive groups within each client's local data. Previous work [33] theoretically demonstrated that, under statistical heterogeneity across clients, global and local fairness can differ,

---

[*]Corresponding author

39th Conference on Neural Information Processing Systems (NeurIPS 2025).

and both entail an inherent trade-off with predictive accuracy. As global and client-level biases can induce heterogeneous treatment disparities among sensitive groups, concurrently mitigating global and local disparities is vital for achieving group fairness in FL. For example, in constructing a federated prediction model for clinical decision-making within a hospital network [48], achieving global fairness substantially enhances performance for disadvantaged subgroups, while fairness at each hospital also carries heightened significance due to local deployment and legal requirements [21].

However, existing methods face certain challenges in controlling group fairness within FL: (i) *Harmonizing global and local fairness, especially in multi-class classification.* Divergent sensitive-group distributions from client heterogeneity separate global and local fairness, thereby imposing an intrinsic trade-off [33, 25]. Most fair FL approaches focus exclusively on either global or local fairness [31, 18, 93, 23, 85], thereby inevitably sacrificing the other objective and impeding the realization of both fairness criteria. Moreover, this research predominantly addresses fairness in the binary case, despite the ubiquity of multiclass tasks in practical FL scenarios. (ii) *Enabling a controllable, optimal accuracy-fairness trade-off with theoretical guarantee.* The non-decomposable, non-differentiable nature of group-fairness measures poses significant optimization challenges [56, 17]. Existing frameworks typically rely on surrogate fairness losses [85, 31, 93, 18, 54], yet the inevitable surrogate-fairness gap [83, 53] produces suboptimal performance and undermines convergence stability, thus hampering the controllability of the accuracy-fairness trade-off.

To address these challenges, we propose a novel **Fed**erated group-**FA**irness **C**alibra**T**ion framework, named FedFACT, comprising in-processing and post-processing approaches. Our framework is capable of ensuring controllable global and local fairness with minimal accuracy deterioration, underpinned by provable convergence and consistency guarantees. To harmonize global and local fairness, we seek to find the optimal classifier under both dual fairness constraints in the multi-class case. To this end, specific characterizations of the federated Bayes-optimal fair classifiers are established for both the in-processing and post-processing phases in FL. Building on the Bayes-optimal fair classifier's structure, we develop efficient, privacy-preserving federated optimization strategies that realize a controllable and theoretically optimal fairness–accuracy trade-off. In detail, FedFACT reduces the in-processing task into a series of personalized cost-sensitive classification problems, and reformulates post-processing as a bi-level optimization that leverages the explicit form of the federated Bayes-optimal fair classifiers. We further derive theoretical convergence and generalization guarantees, demonstrating that our methods achieve near-optimal model performance while enforcing tunable global and local fairness constraints.

Our extensive experiments on multiple real-world datasets verify the efficiency and effectiveness of FedFACT, highlighting that FedFACT delivers a superior, controllable accuracy-fairness balance while maintaining competitive classification performance compared to state-of-the-art methods.

Our main contributions are summarized as follows:

- To the best of our knowledge, we are the first to propose a multi-class federated group-fairness calibration framework that approaches the Bayes-optimal fair classifiers, explicitly tailored to achieve a provably optimal and controllable balance between global fairness, local fairness, and accuracy.

- We further develop efficient algorithms to derive optimal classifiers under global-local fairness constraints at in-processing and post-processing stages with provable convergence and consistency guarantees. The in-processing fair classification is reduced to a sequence of personalized cost-sensitive learning problem, while the post-processing is formulated as a bi-level optimization, using the closed-form representation of Bayes-optimal fair classifiers.

- We conduct extensive experiments on multiple datasets with various data heterogeneity. The experimental results demonstrate that FedFACT outperforms existing methods in achieving superior balances among global fairness, local fairness, and accuracy. Experiments also show that FedFACT enables the flexible adjustment of the accuracy-fairness trade-off in FL.

## 2 Related Work

**Group Fairness in Machine Learning.** As summarized in previous work [56], group fairness is broadly defined as the absence of prejudice or favoritism toward a sensitive group based on their inherent characteristics. Common strategies for realizing group fairness in machine learning can be

classified into three categories: pre-, in-, and post-processing methods. **Pre-processing** [47, 42, 43] approaches aim to modify training data to eradicate underlying bias before model training. **In-processing** [45, 50, 2, 81, 83, 87, 92, 84] methods are developed to achieve fairness requirements by intervening during the training process. **Post-processing** [13, 20, 91, 78, 79, 37, 46] methods adjust the prediction results generated by a given model to adapt to fairness constraints after the training stage. However, because these methods require access to the full dataset, they are confined to mitigating disparities only at the local level.

**Group Fairness in Federated Learning.** Current methods primarily utilize in-processing and post-processing strategies to address global or local fairness issues. Concerning local fairness, prior work [12] highlights potential detrimental effects of FL on the group fairness of each clients, while [18] and [85] employ unified and personalized multi-objective optimization algorithms, respectively, to navigate the trade-off between local fairness and accuracy. Concerning global fairness, two main approaches are adaptive reweighting techniques [58, 31, 93, 1] and optimizing relaxed fairness objectives within FL [23, 75, 26], generally replacing the fairness metrics with surrogate functions.

Furthermore, previous work [33] offered a theoretical study elucidating the divergence between local and global fairness in FL, while revealing the intrinsic trade-off between these fairness objectives and accuracy. [25] formulates local and global fairness optimization into a linear program for minimal fairness cost, but it does not realize the Bayes-optimal balance between accuracy and fairness. [95] derives the Bayes-optimal classifier and decomposes the overall problem into client-specific optimizations, yet their approach applies only to binary classification in post-processing. Persistent challenges include inadequate accuracy-fairness flexibility and limited convergence guarantee in mitigating disparities within multi-class classification across various stages of FL training.

# 3  Preliminary

**Notation.** Denote by $\mathbf{1}_m$ the $m$-dimensional all-ones vector and by $\mathbf{1}_{m \times m}$ the $m \times m$ all-ones matrix. Write the probability simplex as $\Delta_m = \{\, q \in [0,1]^m : \|q\|_1 = 1 \,\}$, and let $e_i$ be the $i$-th standard basis vector. Throughout this paper, bold lowercase letters denote vectors and bold uppercase letters denote matrices. For two equally sized matrices $\mathbf{A}$ and $\mathbf{B}$, their Frobenius inner product is $\langle \mathbf{A}, \mathbf{B} \rangle = \sum_{i,j} a_{ij} b_{ij}$. For positive integer $n$, $[n] = \{1, \ldots, n\}$.

**Fairness in classification.** Let $(X, A, Y)$ be a random tuple, where $X \in \mathcal{X}$ for some feature space $\mathcal{X} \subseteq \mathbb{R}^d$, labels $Y \in \mathcal{Y} = [m]$, and the discrete sensitive attribute $A \in \mathcal{A}$. Given the randomized classifiers $h : \mathcal{X} \times \mathcal{A} \to \Delta_m$, the prediction $\widehat{Y}$ is associated with the random outputs of $h$ defined by $\mathbb{P}(\widehat{Y} = y \mid \mathbf{x}) = h_y(\mathbf{x})$. In this work, we generally focus on three popular group-fairness criteria—Demographic Parity (DP) [27], Equalized Opportunity (EOP) [37] and Equalized Odds [37]—in multiclass classification tasks with multiple sensitive attributes, as defined in prior works [20, 78, 79].

**Group fairness in FL.** A federated system consists of numerous decentralized clients, so that we consider the population data distribution represented by a jointly random tuple $(X, A, Y, K)$ with total $N$ clients. The $k$-th client possesses a local data dataset $\mathcal{D}_k$, $k \in [N]$. Each sample in $\mathcal{D}_k$ is assumed to be drawn from local distribution, represented as $\{(x_{k,i}, a_{k,i}, y_{k,i})\}_{i=1}^{n_k}$, where $n_k$ represents the number of samples for client $k$. The Bayes score function is commonly used to characterize the performance-optimal classifier under fairness constraints [91, 13, 78, 20], which possesses a natural extension in the federated setting: $\eta_y(x, a, c) := \mathbb{P}(Y = y \mid X = x, A = a, K = k)$. We are interested in both local and global fairness criteria in the FL context following [33, 25, 95]:

**Definition 1. (Global Fairness)** The disparity regarding sensitive groups aroused by the federated model in global data distribution $\mathbb{P}(X, A, Y)$ across all clients.

**Definition 2. (Local Fairness)** The disparity regarding sensitive groups aroused by the federated model when evaluated on each client's data distribution $\mathbb{P}(X, A, Y \mid K)$.

**Confusion matrices.** Confusion matrices encapsulate the information required to evaluate diverse performance metrics and assess group fairness constraints in classification tasks [81, 60, 46]. The population confusion matrix is $\mathbf{C} \in [0,1]^{m \times m}$, with elements defined for $i, j \in [m]$ as $\mathbf{C}_{i,j} = \mathbb{P}(Y = i, \widehat{Y} = j)$. To capture both local and global fairness across multiple data distributions within

Table 1: Example of Confusion-Matrix-Based Group Fairness Constraints in Centralized & Federated Learning.

| Fairness Criterion | Demographic parity (DP) | Equal Opportunity (EOP) |
|---|---|---|
| Group Constraints (Centralized) | $\left\|\mathbb{P}(\widehat{Y}=y \mid A=a') - \mathbb{P}(\widehat{Y}=y)\right\|$ $\forall a' \in \mathcal{A}, \forall y \in \mathcal{Y}$ | $\left\|\mathbb{P}(\widehat{Y}=y \mid A=a', Y=y) - \mathbb{P}(\widehat{Y}=y \mid Y=y)\right\|$ $\forall a' \in \mathcal{A}, \forall y \in \mathcal{Y}$ |
| Matrix Notations (Centralized) | $\left\|\sum_a \sum_i \left(\mathbb{I}[a=a'] - p_a\right)\mathbf{C}_{i,y}^a\right\|$ $\forall a' \in \mathcal{A}, \forall y \in \mathcal{Y}$ | $\left\|\sum_a \left(\frac{p_{a'}}{p_{a',y}}\mathbb{I}[a=a'] - \frac{1}{p_y}\right)\mathbf{C}_{y,y}^a\right\|$ $\forall a' \in \mathcal{A}, \forall y \in \mathcal{Y}$ |
| Global Fairness (Federated) | $\left\|\sum_a \sum_k \sum_i \left(p_{k\mid a'}\mathbb{I}[a=a'] - p_{a,k}\right)\mathbf{C}_{i,y}^{a,k}\right\|$ $\forall a' \in \mathcal{A}, \forall y \in \mathcal{Y}$ | $\left\|\sum_a \sum_k \left(\frac{p_{a',k}}{p_{a',y}}\mathbb{I}[a=a'] - \frac{p_{a,k}}{p_y}\right)\mathbf{C}_{i,y}^{a,k}\right\|$ $\forall a' \in \mathcal{A}, \forall y \in \mathcal{Y}$ |
| Local Fairness (Federated) | $\left\|\sum_a \sum_i \left(\mathbb{I}[a=a'] - p_{a\mid k}\right)\mathbf{C}_{i,y}^{a,k}\right\|$ $\forall a' \in \mathcal{A}, \forall y \in \mathcal{Y}$ | $\left\|\sum_a \sum_k \left(\frac{p_{a',k}}{p_{a',y,k}}\mathbb{I}[a=a'] - \frac{p_{a,k}}{p_{y,k}}\right)\mathbf{C}_{i,y}^{a,k}\right\|$ $\forall a' \in \mathcal{A}, \forall y \in \mathcal{Y}$ |

FL, we propose the **decentralized group-specific confusion matrices** $\mathbf{C}^{a,k}, a \in \mathcal{A}, k \in [N]$, with elements defined for $i, j \in [m]$ as $\mathbf{C}_{i,j}^{a,k}(h) := \mathbb{P}(Y=i, \widehat{Y}=j \mid A=a, K=k)$.

**Fairness and performance metrics.** As presented in Table 1 (EO criterion and notation explanations see Appendix A), the global fairness constraints typically can be expressed by $|\mathscr{D}_{u_g}^g(h)| \leq \xi^g, u_g \in \mathcal{U}_g$, where $\mathscr{D}_{u_g}^g(h) = \sum_a \sum_k \langle \mathbf{D}_{u_g}^{a,k}, \mathbf{C}^{a,k}(h)\rangle$ represents the constraints required to achieve the global fairness criterion. Similarly, the local fairness constraints are $|\mathscr{D}_{u_k}^k(h)| \leq \xi^k, u_k \in \mathcal{U}_k, k \in [N]$, with $\mathscr{D}_{u_k}^k(h) = \sum_a \langle \mathbf{D}_{u_k}^{a,k}, \mathbf{C}^{a,k}(h)\rangle$. For performance metrics, we consider a risk metric expressed as a linear function of the population confusion matrix, i.e. $\mathcal{R}(h) = \langle \mathbf{R}, \mathbf{C}(h)\rangle = \sum_a \sum_k p_{a,k}\langle \mathbf{R}, \mathbf{C}^{a,k}(h)\rangle$. This formulation has been explored in multi-label and fair classification contexts [76, 81], and encompasses a variety of performance metrics, such as average recall and precision. In this paper, we primarily focus on standard classification error to set $\mathbf{R} = \mathbf{1}_{m \times m} - \mathbf{I}$.

## 4  Methodology

### 4.1  Federated Bayes-Optimal Fair Classifier

To investigate the optimal classifier with the group fairness guarantee within FL, we consider the situation that there is a unified fairness constraint at the global level, and each client has additional local fairness restrictions in response to personalized demands. Therefore, it is appropriate to consider a personalized federated model to minimize classification risk and ensure both local and global fairness. Denoting the set of classifiers as $\mathcal{H} = \{h : \mathcal{X} \to \Delta_m\}$, the federated Bayes-optimal fair classification problem can be formulated as

$$\min_{\mathbf{h} \in \mathcal{H}^N} \mathcal{R}(\mathbf{h}), \quad \text{s.t. } |\mathscr{D}^g(\mathbf{h})| \leq \xi^g, \quad |\mathscr{D}^k(\mathbf{h})| \leq \xi^k, \quad k \in [N], \tag{1}$$

where $\xi^k, \xi^g$ are positive bounds, $\mathscr{D}^g(\mathbf{h}) := \{\mathscr{D}_{u_g}^g(\mathbf{h})\}_{u_g \in \mathcal{U}_g}$, $\mathscr{D}^k(\mathbf{h}) := \{\mathscr{D}_{u_k}^k(\mathbf{h})\}_{u_k \in \mathcal{U}_k}$, and the inequality applies element-wise. FL model $\mathbf{h} = (h_1, \ldots, h_N)$ comprises $N$ local classifiers.

Before delving into the optimal solution for Problem (1), we present a formal result on the structure of the federated Bayes-optimal fair classifier. Proposition 1 indicates that the Bayes-optimal classifier can be decomposed into local deterministic classifiers for all clients. This observation provides valuable insights for the subsequent algorithm design. The proof and the discussion on feasibility are given in Appendix B.1.

**Proposition 1.** *If* (1) *is feasible for any positive* $\xi^k$ *and* $\xi^g$, *the client-wise classifier* $h_k^*$ *in federated Bayes-optimal fair classifier* $\mathbf{h}^* = (h_1^*, \ldots, h_N^*)$ *can be expressed as the linear combination of some deterministic classifiers* $\{h_{k,i}'\}_{i=1}^{d_k}$, *i.e.,* $h_k^*(x) = \sum_{i=1}^{d_k} \alpha_{k,i} h_{k,i}'(x), \ \alpha_k \in \Delta_{d_k}$.

### 4.2  In-processing Fair Federated Training via Cost Sensitive Learning

In this section, we aim to seek for the optimal solution of $\mathbf{h} = (h_1, \ldots, h_N)$, where each local classifier is attribute-blind and parameterized by $\phi_k$. A direct approach to solving (1) is to formulate an equivalent convex-concave saddle point problem in terms of its Lagrangian $\mathcal{L}(\mathbf{h}, \lambda, \mu)$:

$$\mathcal{R}(\mathbf{h}) + (\lambda^{(1)} - \lambda^{(2)})^\top \mathscr{D}^g(\mathbf{h}) - (\lambda^{(1)} + \lambda^{(2)})^\top \xi^g + \sum_{k=1}^N (\mu_k^{(1)} - \mu_k^{(2)})^\top \mathscr{D}^k(\mathbf{h}) - (\mu_k^{(1)} + \mu_k^{(2)})^\top \xi^k,$$

where $\lambda = \{\lambda^{(1)}, \lambda^{(2)}\}$ and $\mu = \{\mu_k^{(1)}, \mu_k^{(2)}\}_{k=1}^N$ are the dual parameters. Let $\Lambda := \{\lambda \in \mathbb{R}_{\geq 0}^{2|\mathcal{U}_g|} : \|\lambda\|_1 \leq B_d\}$ and $\mathcal{M} := \{\mu \in \mathbb{R}_{\geq 0}^{2\sum_k |\mathcal{U}_k|} : \|\mu\|_1 \leq B_d\}$. Since $\mathbf{h}$ is random classifier, by Sion's minimax theorem [66], the primal problem can be written as a saddle-point optimization

$$\max_{\lambda \in \Lambda, \mu \in \mathcal{M}} \min_{\mathbf{h} \in \mathcal{H}^N} \mathcal{L}(\mathbf{h}, \lambda, \mu) = \min_{\mathbf{h} \in \mathcal{H}^N} \max_{\lambda \in \Lambda, \mu \in \mathcal{M}} \mathcal{L}(\mathbf{h}, \lambda, \mu). \tag{2}$$

The boundedness of optimal $\lambda^*, \mu^*$ will be shown later. To derive the representation of optimal saddle point, we initially focus on the inner minimization optimization task, namely $\min_{\mathbf{h} \in \mathcal{H}} \mathcal{L}(\mathbf{h}, \lambda, \mu)$.

**Proposition 2.** *Given non-negative $\lambda$ and $\mu$, then an optimal solution $\mathbf{h}^* = (h_1^*, \ldots, h_N^*)$ to the inner problem $\min_{\mathbf{h} \in \mathcal{H}} \mathcal{L}(\mathbf{h}, \lambda, \mu)$ is realized by local deterministic classifiers $h_k^*(x)$, $k \in [N]$ satisfying*

$$h_k^*(x) = e_y, \ y \in \underset{j \in [m]}{\arg\max} \Big( \sum_{a \in \mathcal{A}} \mathbb{P}(A = a|x, k) \big[\mathbf{M}^{\lambda, \mu}(a, k)\big]^\top \eta(x, a, k) \Big)_j, \tag{3}$$

*where $\mathbf{M}^{\lambda, \mu}(a, k) := \mathbf{I} - \frac{1}{p_{a,k}} \Big[ \sum_{u_g \in \mathcal{U}_g} (\lambda_{u_g}^{(1)} - \lambda_{u_g}^{(2)}) \mathbf{D}_{u_g}^{a,k} - \sum_{u_k \in \mathcal{U}_k} (\mu_{k,u_k}^{(1)} - \mu_{k,u_k}^{(2)}) \mathbf{D}_{u_k}^{a,k} \Big].$*

The proof of Proposition 2 is given in Appendix B.2. Notice that the optimal solution in (3) remains computationally intractable, because the point-wise distributions $\mathbb{P}(A = a \mid x, k)$ and the Bayes-optimal classifier $\eta$ are unknown. We reformulate the task of solving $\mathbf{h}^*$ within a cost-sensitive learning framework, by designing sample-wise calibrated training losses for each $h_k^*(x)$ that can yield an equivalent objective.

**Proposition 3.** *Let the personalized cost-sensitive loss for client $k$ be defined by*

$$\ell_k(y, \mathbf{s}(x), a) = -\sum_{i=1}^m \overline{\mathbf{M}}_{y,i}^{\lambda, \mu}(a, k) \log \frac{\exp([\mathbf{s}(x)]_i)}{\sum_{j=1}^m \exp([\mathbf{s}(x)]_j)}, \tag{4}$$

*where $\mathbf{s} : \mathcal{X} \to \mathbb{R}^m$ is the scoring function, and $\overline{\mathbf{M}}^{\lambda, \mu}(a, k) = \mathbf{M}^{\lambda, \mu}(a, k) + \kappa \mathbf{1}_{m \times m}$ with $\kappa$ chosen to ensure all matrix entries are strictly positive. Denoting the optimal scoring function to minimize $\ell_k$ over the local data distribution as $\mathbf{s}_k^*(x)$, then the loss $\ell_k$ is calibrated for the inner problem $\min_{\mathbf{h} \in \mathcal{H}} \mathcal{L}(\mathbf{h}, \lambda, \mu)$, i.e., $h_k^*(x) = e_y, y \in \arg\max_{j \in [m]} [\mathbf{s}_k^*(x)]_j$ is equivalent to that in (3).*

In practice, $\mathbf{s}$ is parameterized by $\phi_k$ for $k \in [N]$, formulating the personalized optimization objective $L_k(f_{\phi_k}) = \sum_{i=1}^{n_k} \ell_k(y_{k,i}, \mathbf{s}(x_{k,i}; \phi_k), a_{k,i})$ in the FL setting. Appendix B.3 provides the proof of Proposition 3 and further presents that the loss $\ell_k$ in (4) is also calibrated for the unified federated Bayes-optimal fair classifier. Inspired by this property, we propose an efficient in-processing algorithm for group-fair classification within FL, as detailed in Algorithm 1.

At each iteration $t$, the personalized classifier $h_k^t$ is obtained by ensembling the unified model $\theta^t$ with the local model $\phi_k^t$. The ensemble weight $w_k^t$ and its update rule balance the contributions of the unified and local models. The following theorem establishes the personalized regret bound w.r.t. the best model parameter, and further demonstrates that our algorithm achieves an $\epsilon$-approximate stochastic saddle point.

**Theorem 4.** *Under mild assumptions, there exist constants $B_k, B_L$, such that the following cumulative regret upper bound is guaranteed for the ensemble personalized models:*

$$\frac{1}{T} \sum_{t=1}^T \mathbb{E}[L_k^t(f_{k,ens}^t) - L_k^t(f_{\phi_k^*})] \leq \frac{\|\phi_k^*\|^2}{\eta RT} + \eta B_k + \frac{\log(2)}{\eta_w T} + \eta_w B_L. \tag{5}$$

*Furthermore, suppose that personalized models achieve a $\rho^t$-approximate optimal response at iteration $t$, namely $\widehat{\mathcal{L}}(\mathbf{h}^t, \lambda^t, \mu^t) \leq \min_{\mathbf{h}} \widehat{\mathcal{L}}(\mathbf{h}, \lambda^t, \mu^t) + \rho^t$, denoting $\bar{\rho}^T = \sum_{t=1}^T \rho^t / T$, then the sequences of model and bounded dual parameters comprise an approximate mixed Nash equilibrium:*

$$\max_{\lambda^*, \mu^*} \frac{1}{T} \sum_{t=1}^T \widehat{\mathcal{L}}(\mathbf{h}^t, \lambda^*, \mu^*) - \inf_{\mathbf{h}^*} \frac{1}{T} \sum_{t=1}^T \widehat{\mathcal{L}}(\mathbf{h}^*, \lambda^t, \mu^t) \leq \epsilon^T := \bar{\rho}^T + 16B_d^2 \sqrt{\frac{1}{T}}. \tag{6}$$

The complete Theorem 4 with its proof is provided in the Appendix B.4. The regret bound yields a convergence rate of $\mathcal{O}(1/\sqrt{T})$ by appropriately choosing the learning rate, reflecting the stability of the proposed algorithm. Moreover, as $\rho^t$ decreases with $t$, the algorithm will gradually converge to the optimal equilibrium. For instance, if $\rho^t \propto C/\sqrt{t}$, $\epsilon$ will also exhibit an $\mathcal{O}(1/\sqrt{T})$ convergence rate. The **generalization error** between the optimal solutions of the empirical dual and primal problems under finite samples is given in **Appendix B.5**.

---

**Algorithm 1:** FedFACT (In-processing)

---

**Input** : Datasets $\{x_{k,i}, y_{k,i}, a_{k,i}\}_{i=1}^{N_k}$ from client $k$, $k \in \mathcal{K}$; Communication round $T$; Local
round $R$; Initial parameters $\lambda^0, \mu^0, \theta^0, \phi_k^0, w_k^0, k \in \mathcal{K}$; Learning rate $\{\eta, \eta_d, \eta_w\}_{t=1}^T$;

**for** $t = 0, 1, \ldots, T$ **do**

  **Each client $k \in \mathcal{K}$ in parallel do:**

  Ensemble unified and local model: $h_k^t(x) = e_y, y \in \arg\max_{j \in [m]} [f_{k,ens}^t(x)]_j$, where
  $f_{k,ens}^t(x) = w_k^t f_{\theta^t}(x) + (1 - w_k^t) f_{\phi_k^t}(x)$ and $f_\phi(x) := \text{softmax}(\mathbf{s}(x; \phi))$;

  Update calibration matrix $\overline{\mathbf{M}}^{\lambda^t, \mu_k^t} = \widehat{\mathbf{M}}^{\lambda^t, \mu_k^t} + \kappa \mathbf{1}_{m \times m}$;

  Update the weight $w_k^{t+1} = \frac{1}{1 + \mathcal{W}_k^t(w_k^t)}$, $\mathcal{W}_k^t(w_k^t) = \frac{1 - w_k^t}{w_k^t} \exp(-\eta_w [L_k(f_{\phi_k^t}) - L_k(f_{\theta^t})])$;

  Calculate update of global dual parameter $\Delta \lambda_k^{t+1}$, and update local dual parameter $\mu_k^{t+1}$,
  $$\Delta \lambda_{k,u_g}^{(i),t+1} = (3 - 2i) \sum_{a \in \mathcal{A}} \langle \widehat{\mathbf{D}}_{u_g}^{a,k}, \widehat{\mathbf{C}}^{a,k}(h_k^t) \rangle - \xi^g, \ i \in [1, 2], u_g \in \mathcal{U}_g,$$
  $$\mu_{k,u_k}^{(i),t+1} = \Pi_{\mathcal{M}} [(3 - 2i) \sum_{a \in \mathcal{A}} \langle \widehat{\mathbf{D}}_{u_k}^{a,k}, \widehat{\mathbf{C}}^{a,k}(h_k^t) \rangle - \xi^k], \ i \in [1, 2], u_k \in \mathcal{U}_k;$$

  Perform $R$ local-batch update of $\theta^t$ and $\phi_k^t$, guided by loss $L_k$ with $\overline{\mathbf{M}}^{\lambda^t, \mu_k^t}$,
  $\theta_k^{t,r+1} = \theta_k^{t,r} - \eta_k \nabla L_k(\theta^{t,r}; \mathcal{B}_k^{t,r}), \phi_k^{t,r+1} = \phi_k^{t,r} - \eta_k \nabla L_k(\phi^{t,r}; \mathcal{B}_k^{t,r}), \ r = 0, \cdots, R - 1$;

  Send last update $\Delta \theta_k^{t+1} = \theta_k^{t,R} - \theta^t$ to the server;

  **Server do:**

  Server aggregates $\{\Delta \theta_k^{t+1}\}$: $\theta^{t+1} = \theta^t + \sum_{k=1}^N p_k \Delta \theta_k^{t+1}$;

  Update global dual parameter: $\lambda^{t+1} = \Pi_\Lambda(\lambda^t + \eta_d \sum_{k=1}^N \Delta \lambda_k^{t+1})$;

  Send $\theta^{t+1}, \lambda^{t+1}$ to clients;

**end**

**Return** Personalized classifier $\overline{\mathbf{h}} = (\bar{h}_1, \cdots, \bar{h}_N)$, where $\bar{h}_k := \sum_{t=1}^T h_k^t / T$;

---

## 4.3 Label-Free Federated Post-Fairness Calibration based on Plug-In Approach

This section introduces a post-hoc fairness approach that calibrates the classification probabilities of
a pre-trained federated model. We formulate an closed-form representation of the federated Bayes
optimal fair classifier under standard assumptions, and then derive the primal problem into bi-level
optimization through the plug-in approach. To begin, we introduce the following assumption.

**Assumption 1.** ($\eta$-continuity). For each client $k$, denoting $\mathcal{P}_{a,k}^X := \mathbb{P}(X \mid A = a, K = k)$, let
the put forward distribution $\tau_{a,k} := \eta_\sharp \mathcal{P}_{a,k}^X$, $a \in \mathcal{A}$ be absolutely continuous with respect to the
Lebesgue measure restricted to $\Delta_m$.

Assumption 1, which can be met by adding minor random noises to $\tau_{a,k}$, is commonly used in the
literature on post-processing fairness [13, 81, 20, 16]. Next, we derive a more explicit characterization
of federated Bayes-optimal fair classifier.

**Theorem 5.** *Under Assumption 1, suppose that the primal problem* (1) *is feasible for any* $\xi^g, \xi^k > 0$
*and all non-zero columns of each* $\mathbf{D}_u^{a,k}$ *are distinct, the attribute-aware personalized classifier*
$\{h_k^*\}_{k \in [N]}$ *is Bayes-optimal under local and global fairness constraints, if* $h_k^*(x, a) = e_y, y \in$
$\arg\max_{j \in [m]} ([\mathbf{M}^{\lambda^*, \mu^*}(a, k)]^\top \eta(x, a, k))_j$, *where the dual parameters are determined from*
$(\lambda^*, \mu^*) \in \arg\min_{\lambda, \mu \geq 0} H(\lambda, \mu)$,

$$H(\lambda, \mu) = \sum_{k \in [N]} \sum_{a \in \mathcal{A}} p_{a,k} \mathop{\mathbb{E}}_{X \sim \mathcal{P}_{a,c}^X} \left[ \max_{y \in [m]} ([\mathbf{M}^{\lambda, \mu}(a, k)]^\top \eta(X, a, k))_y \right] + \xi^g \|\lambda\|_1 + \sum_{k \in [N]} \xi^k \|\mu_k\|_1. \tag{7}$$

*The optimal dual parameters* $(\lambda^*, \mu^*)$ *are bounded, and the optimality of* $\lambda$ *and* $\mu$ *respectively
guarantee global fairness and local fairness.*

Proof of Theorem 5 is given in Appendix B.6. Since the dual parameter $\mu$ only related to local fairness
constraints, each clients can finish update of this parameter without global aggregation. Therefore,

the federated Bayes-optimal classification problem can be reformulated into a bi-level optimization:

$$\min_{\lambda \in \Lambda} \left\{ \widehat{F}(\lambda) = \sum_{k=1}^{N} \hat{p}_k \widehat{F}_k(\lambda) \right\}, \quad \widehat{F}_k(\lambda) := \min_{\mu_k \in \mathcal{M}_k} \widehat{H}_k(\lambda, \mu_k), \tag{8}$$

where $\widehat{H}_k(\lambda, \mu_k) := \frac{1}{n_k} \sum_{i=1}^{n_k} \max_{y \in [m]} \left( \left[ \mathbf{M}^{\lambda,\mu}(a_i, k) \right]^\top \eta(x_i, a_i, k) \right)_y + \xi^g \|\lambda\|_1 + \frac{\xi^k}{\hat{p}_k} \|\mu_k\|_1$ is the plug-in estimation of (7). Considering that the non-smoothness of the optimization objective may lead to convergence issues in federated optimization [89], we replace the maximum operation in $\widehat{H}_k(\lambda, \mu_k)$ with soft-max weight function $\sigma_\beta(x) = \sum_{i=1}^{m} \frac{\exp(x_i/\beta)}{\sum_{j=1}^{m} \exp(x_j/\beta)} x_i$, which reduces to the hard-maximum if temperature $\beta \to 0$. Denoting the relaxed local objective as $\widehat{H}'_k(\lambda, \mu_k)$, we propose Algorithm 2 to solve the federated Bayes-optimal fair classifier.

---

**Algorithm 2:** FedFACT (Post-processing)

---

**Input** : Datasets $\mathcal{D}_k = \{x_{k,i}, y_{k,i}, a_{k,i}\}_{i=1}^{N_k}$ from client $k \in [N]$; Communication round $T$;
  Local round $R$; Initial parameters $\lambda^0, \mu^0, k \in [N]$; Learning rate $\eta_d$; Pre-trained $\hat{\eta}$
**for** $t = 0, 1, \ldots, T$ **do**
  | Each client $k \in [N]$ **in parallel do:**
  | Perform $R$ local-batch update of $\mu_k^t, i = 1, 2$ with $\widehat{H}'_k(\lambda, \mu_k)$:
  |   $\mu_k^{t,r+1} = \Pi_{\mathcal{M}_k}(\mu_k^{t,r} - \eta_d \nabla_\mu \widehat{H}'_k(\lambda, \mu_k)), r = 0, \cdots, R-1.$
  | Set $\mu_k^{t+1} = \mu_k^{t,R}$, and calculate update of $\lambda^t$: $\Delta \lambda_k^{t+1} = \nabla_\lambda \widehat{H}'_k(\lambda^t, \mu_k^{t+1})$.
  | **Server do:**
  | Server aggregates $\{\Delta \lambda_k^{t+1}\}$: $\lambda^{t+1} = \Pi_\Lambda(\lambda^t - \eta_d \sum_{k=1}^{N} \hat{p}_k \Delta \lambda_k^{t+1})$; Send $\lambda^{t+1}$ to clients;
**end**
**Return** Classifiers $\{h_1, \ldots, h_N\}$, $h_k(x, a) := \arg\max_{j \in [m]} \left( \left[ \widehat{\mathbf{M}}^{\lambda^*, \mu^*}(a, k) \right]^\top \hat{\eta}(x, a, k) \right)_j$;

---

**Proposition 6.** *The bi-level objectives $\widehat{H}'_k(\lambda, \mu_k), k \in [N]$ are convex and L-smooth.*

Proof of Proposition 6 is given in Appendix B.7. Existing research in FL [74] shows that the $L$-smoothness of the local objective suffice for Algorithm 2 to achieve an $\mathcal{O}(1/\sqrt{T})$ convergence rate. Moreover, owing to the equivalence of nested and joint minimization under convexity [66, 36], the corresponding bi-level optimization can approach the optimal solution of the empirical primal problem. Consequently, the remaining error arises from the **generalization risk** induced by finite sampling, which is explored in **Appendix B.8**.

### 4.4 Discussion

**In- versus post-processing.** In- and post-processing interventions play complementary roles: the former removes bias in representations during training (incurring higher computational cost), and the latter adjusts fairness on model outputs with low overhead, unable to debias learned representations. Both of them support adaptable fairness calibration in resource-limited, heterogeneous FL. Note that combining the in-processing and post-processing methods is theoretically unjustifiable, as the in-processing classifier is not designed to approximate the Bayes score function.

**Efficiency & Privacy.** Each iteration of our in- and post-processing methods requires only a single client-server interaction and is supported by convergence guarantees that demonstrate our algorithms' efficiency. This will be empirically validated in our experiments; FedFACT is also privacy-friendly. In-processing requires sharing $\lambda$ alongside the unified model $\theta$, while post-processing involves sharing only $\lambda$. These exchanges conform to standard FL [55] and preserve data confidentiality. Furthermore, differential privacy [28, 30] or encryption schemes [7] can be applied to further reinforce privacy.

## 5 Experiments

To comprehensively assess the proposed FedFACT framework, we conduct extensive experiments on four publicly available real-world datasets to answer the following Research Questions (RQ): **RQ1**:

Table 2: Overall Experimental Results.

| Partition | Method | Compas | | | Adult | | | CelebA | | | ENEM | | |
|---|---|---|---|---|---|---|---|---|---|---|---|---|---|
| | | Acc | $\mathscr{D}^{global}$ | $\mathscr{D}^{local}$ | Acc | $\mathscr{D}^{global}$ | $\mathscr{D}^{local}$ | Acc | $\mathscr{D}^{global}$ | $\mathscr{D}^{local}$ | Acc | $\mathscr{D}^{global}$ | $\mathscr{D}^{local}$ |
| | FedAvg | 69.73 | 0.2766 | 0.3590 | 84.52 | 0.1765 | 0.2310 | 89.14 | 0.1435 | 0.1308 | 67.56 | 0.2620 | 0.2462 |
| $\gamma = 0.5$ | FairFed | 59.39 | 0.1008 | 0.1022 | 80.73 | 0.0983 | 0.1434 | 81.85 | 0.0704 | 0.1058 | 60.99 | 0.1165 | 0.1733 |
| | FedFB | 58.09 | 0.0879 | 0.0983 | 81.85 | 0.0751 | 0.1165 | 85.32 | 0.1188 | 0.0949 | 64.35 | 0.0814 | 0.1326 |
| | FCFL | 56.53 | 0.0646 | 0.0614 | 81.91 | 0.0845 | 0.1455 | 83.83 | 0.0704 | 0.1090 | 61.07 | 0.1260 | 0.1189 |
| | praFFed | 59.93 | 0.0824 | 0.0968 | 80.96 | 0.0591 | 0.0763 | 85.45 | 0.0731 | 0.0862 | 62.60 | 0.0736 | 0.0806 |
| | Cost | 64.51 | 0.0585 | 0.0941 | 81.09 | 0.0262 | 0.0590 | 85.60 | 0.0314 | 0.0577 | 65.79 | 0.0487 | 0.0674 |
| | FedFACT$_g$ (In) | 61.19 | 0.0344 | 0.0761 | 82.05 | 0.0015 | 0.0408 | 86.49 | 0.0205 | 0.0544 | 63.79 | 0.0493 | 0.0568 |
| | FedFACT$_l$ (In) | 61.81 | 0.0600 | 0.0636 | 82.44 | 0.0140 | 0.0508 | 85.90 | 0.0461 | 0.0312 | 63.93 | 0.0434 | 0.0487 |
| | FedFACT$_{g\&l}$ (In) | 61.17 | 0.0407 | 0.0732 | 82.04 | 0.0014* | 0.0401 | 86.15 | 0.0382 | 0.0473 | 62.51 | 0.0366 | 0.0413 |
| | FedFACT$_g$ (Post) | 67.27 | 0.0128* | 0.0660 | 82.83* | 0.0173 | 0.0276 | 87.25 | 0.0089* | 0.0253 | 66.15 | 0.0175 | 0.0181* |
| | FedFACT$_l$ (Post) | 67.49* | 0.0315 | 0.0552* | 82.79 | 0.0154 | 0.0267* | 87.36* | 0.0127 | 0.0163* | 66.54* | 0.0197 | 0.0240 |
| | FedFACT$_{g\&l}$ (Post) | 67.33 | 0.0139 | 0.0641 | 82.74 | 0.0134 | 0.0274 | 87.06 | 0.0093 | 0.0172 | 66.52 | 0.0162* | 0.0184 |
| | FedAvg | 69.12 | 0.2513 | 0.4044 | 85.34 | 0.1596 | 0.2358 | 89.85 | 0.1360 | 0.1742 | 67.45 | 0.2037 | 0.3068 |
| Hetero | FairFed | 61.87 | 0.1825 | 0.2448 | 81.85 | 0.1074 | 0.1283 | 82.50 | 0.0672 | 0.1415 | 59.81 | 0.0949 | 0.1624 |
| | FedFB | 60.16 | 0.1284 | 0.1332 | 81.14 | 0.0949 | 0.1005 | 86.00 | 0.1121 | 0.1284 | 63.68 | 0.0780 | 0.2039 |
| | FCFL | 59.96 | 0.1498 | 0.1507 | 79.10 | 0.0528 | 0.0596 | 84.50 | 0.0657 | 0.1458 | 61.05 | 0.1134 | 0.1425 |
| | praFFed | 60.42 | 0.0902 | 0.1062 | 80.12 | 0.0523 | 0.0606 | 85.13 | 0.0592 | 0.1152 | 60.84 | 0.0932 | 0.1246 |
| | Cost | 63.01 | 0.0773 | 0.1044 | 81.04 | 0.0286 | 0.0567 | 86.28 | 0.0495 | 0.0761 | 62.11 | 0.0315 | 0.0730 |
| | FedFACT$_g$ (In) | 60.33 | 0.0665 | 0.0841 | 82.09* | 0.0122 | 0.0962 | 86.18 | 0.0188 | 0.0731 | 62.05 | 0.0380 | 0.0577 |
| | FedFACT$_l$ (In) | 60.22 | 0.0730 | 0.0753 | 81.19 | 0.0208 | 0.0250 | 86.58 | 0.0424 | 0.0426 | 61.96 | 0.0471 | 0.0473 |
| | FedFACT$_{g\&l}$ (In) | 61.44 | 0.0676 | 0.0789 | 81.19 | 0.0055 | 0.0239* | 86.38 | 0.0355 | 0.0634 | 61.48 | 0.0322 | 0.0364 |
| | FedFACT$_g$ (Post) | 64.36 | 0.0398* | 0.0699 | 81.09 | 0.0047* | 0.0257 | 87.95 | 0.0094 | 0.0344 | 65.32 | 0.0199* | 0.0343 |
| | FedFACT$_l$ (Post) | 64.38 | 0.0479 | 0.0740 | 81.27 | 0.0049 | 0.0306 | 88.05* | 0.0112 | 0.0217* | 65.68* | 0.0246 | 0.0120* |
| | FedFACT$_{g\&l}$ (Post) | 64.41* | 0.0408 | 0.0680* | 81.31 | 0.0053 | 0.0293 | 87.75 | 0.0088* | 0.0247 | 65.19 | 0.0201 | 0.0131 |

\* The best results are marked with \*. The second-best results are underlined.

\* All results are the average of five repeated experiments.

\* We use **FedAvg** as the baseline for optimal accuracy, without comparing its accuracy-fairness trade-off.

Does FedFACT outperform the existing methods in effectively achieving a global-local accuracy-fairness balance? **RQ2**: Is FedFACT capable of adjusting the trade-off between accuracy and global-local fairness (sensitivity analysis)? **RQ3**: How do important hyper-parameters influence the performance of FedFACT? **RQ4**: How about the communication efficiency and scalability of FedFACT?

## 5.1 Datasets and Experimental Settings

Due to space limitations, the detailed information in this section is provided in **Appendix C**.

**Datasets.** Experiments are conducted on four real-world datasets: **Compas**[22], **Adult** [4], **CelebA** [96], and **ENEM** [40], which are well established for assessing fairness in FL [31, 12, 24, 95].

**Baselines.** For binary classification, experiments are conducted on all four datasets. We compare our method with traditional federated baselines **FedAvg** [55] and five state-of-the-art methods tailored for addressing global and local fairness within FL, namely **FairFed** [31], **FedFB** [93], **FCFL** [18], **praFFed** [85] and the method in [25], denoted as **Cost** in our experiments. The reason for we did not include the experiments with [95] is explained in Appendix C.2. For multi-group or multi-class classification, the experiments are implemented on CelebA and ENEM datasets due to label limitations.

**Data distribution.** To model the statistical heterogeneity in the FL context, we investigate two data partitioning strategies: *(i) Dirichlet partition*: we control the distribution of the sensitive attribute at each client using a Dirichlet distribution $Dir(\gamma)$ as proposed by [31]. A smaller $\gamma$ indicates greater heterogeneity across clients. *(ii) Heterogeneous split*: Inspired by [33], we propose a partitioning method that introduces heterogeneous correlations between the sensitive attribute $A$ and label $Y$. The correlation between $A$ and $Y$ for each client is controlled by a parameter randomly sampled from $[0, 1]$, as detailed in Appendix C.3.

**Evaluation.** *(i) Firstly*, we partition each dataset into a 60% training set and the remaining 40% for test set. *(ii) Secondly*, the number of clients is set to 2 in Compas, and 5 in other datasets to ensure sufficient samples for local fairness estimation. *(iii) Thirdly*, we evaluate the FL model with Accuracy (Acc), global fairness metric ($\mathscr{D}^{global}$), and maximal local fairness metric among clients ($\mathscr{D}^{local}$), with smaller values of fairness metrics indicating a fairer FL model.

## 5.2 Overall Comparison (RQ1)

We perform extensive experiments comparing FedFACT against existing fair FL baselines under varying statistical heterogeneity. We set $\xi^g = \xi^k = 0.01$ for FedFACT. The subscript $g$ and $l$

Table 3: Accuracy-Fairness Balance (Sensitivity Analysis).

| Dataset | Compas (In-) | | | Adult (In-) | | | Compas (Post-) | | | Adult (Post-) | | |
|---|---|---|---|---|---|---|---|---|---|---|---|---|
| $(\xi^g, \xi^l)$ | Acc | $\mathscr{D}^{global}$ | $\mathscr{D}^{local}$ | Acc | $\mathscr{D}^{global}$ | $\mathscr{D}^{local}$ | Acc | $\mathscr{D}^{global}$ | $\mathscr{D}^{local}$ | Acc | $\mathscr{D}^{global}$ | $\mathscr{D}^{local}$ |
| (0.00,0.00) | 61.17 | 0.0407 | 0.0732 | 82.04 | 0.0014 | 0.0401 | 67.33 | 0.0139 | 0.0641 | 82.74 | 0.0134 | 0.0274 |
| (0.02,0.00) | 61.39 | 0.0548 | 0.0848 | 82.37 | 0.0028 | 0.0458 | 67.49 | 0.0315 | 0.0552 | 82.75 | 0.0154 | 0.0255 |
| (0.04,0.00) | 61.81 | 0.0600 | 0.0836 | 82.44 | 0.0140 | 0.0508 | 67.92 | 0.0557 | 0.0692 | 82.83 | 0.0173 | 0.0276 |
| (0.00,0.02) | 61.23 | 0.0418 | 0.0732 | 82.04 | 0.0018 | 0.0409 | 67.40 | 0.0134 | 0.0658 | 82.76 | 0.0139 | 0.0278 |
| (0.02,0.02) | 61.50 | 0.0569 | 0.0895 | 82.41 | 0.0056 | 0.0450 | 67.93 | 0.0558 | 0.0624 | 82.77 | 0.0166 | 0.0262 |
| (0.04,0.02) | 61.63 | 0.0665 | 0.0933 | 82.52 | 0.0080 | 0.0479 | 67.95 | 0.0623 | 0.0598 | 82.81 | 0.0150 | 0.0256 |
| (0.00,0.04) | 61.23 | 0.0418 | 0.0732 | 82.04 | 0.0014 | 0.0408 | 67.27 | 0.0128 | 0.0660 | 82.79 | 0.0154 | 0.0267 |
| (0.02,0.04) | 61.66 | 0.0556 | 0.0919 | 82.57 | 0.0089 | 0.0442 | 67.95 | 0.0536 | 0.0644 | 82.81 | 0.0174 | 0.0283 |
| (0.04,0.04) | 62.39 | 0.0720 | 0.1105 | 82.66 | 0.0223 | 0.0449 | 68.03 | 0.0645 | 0.0598 | 82.81 | 0.0185 | 0.0278 |

represent the presence of global and local fairness constraints, respectively. It is essential to note that there is an inherent trade-off between accuracy and global-local fairness.

**Binary classification results.** We compare FedFACT with benchmarks tailored to binary classification in terms of the binary DP and EOP criteria on the four datasets. The results of DP are presented in Table 2. We also report the Pareto frontier in **Appendix D.1** to evaluate the ability of FedFACT to strike a accuracy-fairness balance, along with the Pareto results of binary EOP criterion. Overall, when compared to existing SOTA methods, FedFACT demonstrates superior performance in achieving a balanced trade-off between accuracy and fairness.

**Multi-Class results.** We illustrate how FedFACT performs on multi-class prediction using CelebA and ENEM with DP and EOP constraints. As there are no established methods addressing multi-class fairness in federated learning, we conduct comparisons only between FedAvg and our proposed in- and post-processing approaches, as shown in Figure 1. More experimental details are in Appendix D.2.

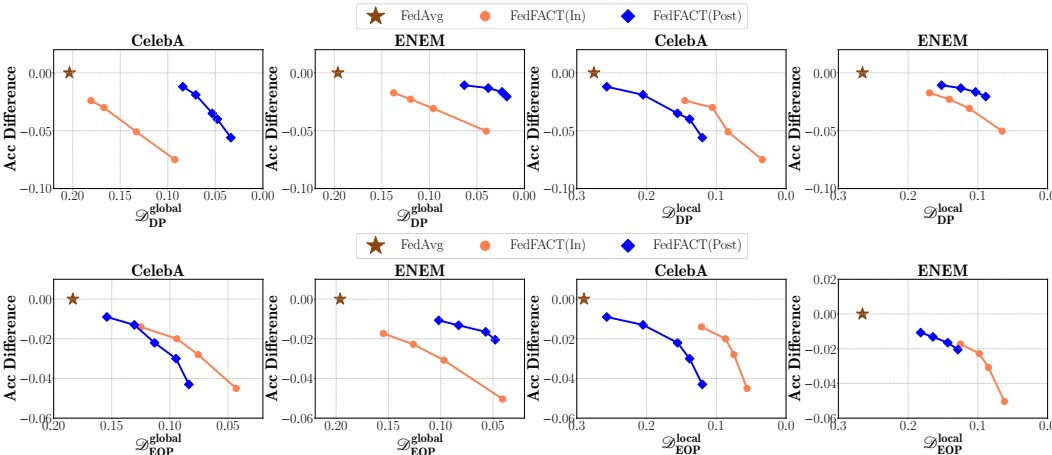

Figure 1: Multi-Class Fair Classification Results. The top line depict global and local multiclass Demographic Parity (DP) results, while the bottom line show global and local multiclass Equal Opportunity (EOP) outcomes.

The outcomes of the multiclass experiments do not parallel those in the binary setting, where post-processing methods vastly outperform alternatives; instead, performance is comparatively lower. This can be attributed to the paucity of local samples for fairness evaluation at individual clients, which causes post-processing under joint global and local fairness constraints to incur significant generalization error and thus fail to precisely enforce local fairness. Under these conditions, the in-training approach, leveraging globally aggregated data, offers superior fairness calibration, thereby underscoring the complementarity of the two methods we introduce.

## 5.3 Flexibility of Adjusting Accuracy-Fairness Trade-Off (RQ2)

To investigate the capability of FedFACT in adjusting accuracy-fairness trade-off, we examine the Acc, $\mathscr{D}^{global}$ and $\mathscr{D}^{local}$ under different fairness relaxation of $(\xi^g, \xi^l)$ with $\gamma = 0.5$ on Adult and Compas in Table 3. Here we set the local fairness levels $\xi^k$ for each client to the same value, denoted as $\xi^l$. More experimental results are presented in Appendix D.3.

**Sensitivity Analysis.** Table 3 shows that, for a fixed global constraint $\xi^g$, reducing $\xi^l$ diminishes both accuracy and local fairness—implying that stricter local fairness comes at the cost of overall performance. Conversely, by keeping $\xi^l$ constant, one can modulate global fairness via adjustments to $\xi^g$. Note that the difference between the constraints and the fairness metrics arises due to the unavoidable generalization error with finite samples. In general, these findings substantiate our claim that FedFACT enables flexible control over the accuracy-fairness trade-off in FL.

## 5.4 Hyper-parameter Experiments (RQ3)

There is no tunable hyper-parameter in our proposed method except for **the number of deterministic classifiers** utilized to construct the weight classifiers. We gradually raise the number of classifiers forming the weighted classifier, starting with the most recent one and extending to the previous 10 classifiers. The detailed experimental results are provided in **Appendix D.4**.

## 5.5 Efficiency and Scalability Study (RQ4)

In **Appendix D.5**, we undertake extensive experiments to empirically demonstrate the communication efficiency and scalability to client number of the proposed method FedFACT.

## 6 Conclusion

This paper introduces a novel controllable **Fed**erated group-**FA**irness **C**alibra**T**ion framework called FedFACT, to ensure both group and local fairness within FL. FedFACT is proposed to learn the federated Bayes-optimal fair classifier in both in- and post-processing stages, which achieves a theoretically minimal accuracy loss with both fairness constraints. We developed efficient algorithms—with convergence and consistency guarantees—that reduce fair classification to personalized cost-sensitive learning for in-pocessing and bi-level optimization for post-processing. Extensive experiments on four publicly available real-world datasets demonstrate that FedFACT outperforms SOTA methods, exhibiting a remarkable ability to harmonious balance between accuracy and global-local fairness.

## Reproducibility Statement

Details for the experimental setting are provided in the beginning of Section 5 and Appendix C, and the code can be found at https://github.com/liizhang/FedFACT.

## Acknowledgments

This work was supported in part by the Hangzhou Key Scientific Research Plan (No. 2024SZD1A28), and National Natural Science Foundation of China (No. 62402148).

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

# A Fairness Criteria in Centralized and Federated Learning Setting

In this section, we provide supplementary discussion of the fairness criteria and their corresponding confusion-matrix formulations under both centralized and federated learning settings. First, in addition to the demographic parity (DP) and equal opportunity (EOP) notions introduced above, we here present the definitions of equality of odds (EO) along with their confusion-matrix representations. Next, we clarify how these fairness notions are formalized within FL, specifying the distinct fairness metrics employed at both the global and the local levels. Note that this paper adopts a subgroup-like fairness metric [81, 14, 44] to reduce the number of constraints, while our confusion-matrix representation is also applicable to the group-wise definitions of these fairness metrics [78, 79].

## A.1 Group Fairness Criteria

**Probabilistic notations.** We elucidate some probability notations in the Preliminaries 3 and Table 1. Here, we use $p_\delta$ to denote the probability of event $\delta$ occurring. For example, $p_a := \mathbb{P}(A = a)$, $p_y = \mathbb{P}(Y = y)$, $p_{a,k} := \mathbb{P}(A = a, K = k)$, $p_{k|a} := \mathbb{P}(K = k \mid A = a)$, $p_{a|k} := \mathbb{P}(A = a \mid K = k)$, $p_{a,y} := \mathbb{P}(A = a, Y = y)$, $p_{y,k} := \mathbb{P}(Y = y, K = k)$, and $p_{a,y,k} := \mathbb{P}(A = a, Y = y, K = k)$.

**Confusion-matrix-based fairness notations.** For random tuple $(X, Y, A)$, the prediction of the (attribute-aware) classifier is defined as $\widehat{Y} = h(X, A)$. One may simply choose $\widehat{Y} = h(X)$ to consider the attribute-blind setting. To represent group fairness constraints, previous works [81, 60] introduce the group-specific confusion matrices $\mathbf{C}^a, a \in \mathcal{A}$ to characterize the fairness constraints, where $\mathbf{C}^a_{i,j} := \mathbb{P}(Y = i, \widehat{Y} = j \mid A = a)$.

**Example 1.** *For DP criterion,*

$$\mathscr{D}_{DP} = \max_{y \in [m]} \max_{a' \in \mathcal{A}} \left| \mathbb{P}(\widehat{Y} = y \mid A = a') - \mathbb{P}(\widehat{Y} = y) \right|,$$

*where* $\mathbb{P}(Y = y \mid A = a') = \sum_{i \in [m]} \mathbb{P}(\widehat{Y} = y, Y = i \mid A = a') = \sum_{i \in [m]} \mathbf{C}^{a'}_{i,y}$ *and* $\mathbb{P}(\widehat{Y} = y) = \sum_{a \in \mathcal{A}} \mathbb{P}(A = a) \sum_{i \in [m]} \mathbb{P}(\widehat{Y} = y, Y = i \mid A = a) = \sum_{a \in \mathcal{A}} \sum_{i \in [m]} \mathbb{P}(A = a) \mathbf{C}^a_{i,y}$.
*Hence, we have*

$$\mathscr{D}_{DP} = \max_{y \in [m]} \max_{a' \in \mathcal{A}} \left| \sum_{a \in \mathcal{A}} \sum_{i \in [m]} \left( \mathbb{I}[a = a'] - \mathbb{P}(A = a) \right) \mathbf{C}^a_{i,y} \right| = \max_{y \in [m]} \max_{a' \in \mathcal{A}} \left| \sum_{a \in \mathcal{A}} \langle \mathbf{D}^a_{a',y}, \mathbf{C}^a \rangle \right|,$$

*where* $\mathbf{D}^a_{a',y} \in \mathbb{R}^{m \times m}$, *and the $y$-th column elements of* $\mathbf{D}^a_{a',y}$ *are* $\mathbb{I}[a = a'] - \mathbb{P}(A = a)$ *with all other elements set to* $0$.

**Example 2.** *For EOP criterion,*

$$\mathscr{D}_{EOP} = \max_{y \in [m]} \max_{a' \in \mathcal{A}} \left| \mathbb{P}(\widehat{Y} = y \mid A = a', Y = y) - \mathbb{P}(\widehat{Y} = y \mid Y = y) \right|,$$

*where* $\mathbb{P}(Y = y \mid A = a', Y = y) = \frac{p_{a'}}{p_{a',y}} \mathbf{C}^{a'}_{y,y}$ *and* $\mathbb{P}(\widehat{Y} = y \mid Y = y) = \sum_{a \in \mathcal{A}} \frac{p_a}{p_y} \mathbf{C}^a_{y,y}$.
*Hence, we have*

$$\mathscr{D}_{EOP} = \max_{y \in [m]} \max_{a' \in \mathcal{A}} \left| \sum_{a \in \mathcal{A}} \left( \frac{p_{a'}}{p_{a',y}} \mathbb{I}[a = a'] - \frac{p_a}{p_y} \right) \mathbf{C}^a_{y,y} \right| = \max_{y \in [m]} \max_{a' \in \mathcal{A}} \left| \sum_{a \in \mathcal{A}} \langle \mathbf{D}^a_{a',y}, \mathbf{C}^a \rangle \right|,$$

*where* $\mathbf{D}^a_{a',y} \in \mathbb{R}^{m \times m}$, *and the entry in the $y$-th row and $y$-th column is* $\frac{p_{a'}}{p_{a',y}} \mathbb{I}[a = a'] - \frac{p_a}{p_y}$ *with all other elements set to* $0$.

**Example 3.** *For EOP criterion, we follow [3] to introduce the mean equalized odds (MEO) constraint, and consider its subgroup-like representation:*

$$\mathscr{D}_{EO} = \max_{y \in [m]} \max_{a' \in \mathcal{A}} \frac{1}{2} (|\text{TPR}_y(a) - \text{TPR}_y| + |\text{FPR}_y(a) - \text{FPR}_y|),$$

*where* $\text{TPR}_y(a) = \mathbb{P}(\widehat{Y} = y \mid Y = y, A = a), \text{TPR}_y = \mathbb{P}(\widehat{Y} = y \mid Y = y)$ *and* $\text{FPR}_y(a) = \mathbb{P}(\widehat{Y} = y \mid Y \neq y, A = a), \text{FPR}_y = \mathbb{P}(\widehat{Y} = y \mid Y \neq y)$.

*It shows that*

$$\frac{1}{2}\left(|\mathrm{TPR}_y(a) - \mathrm{TPR}_y| + |\mathrm{FPR}_y(a) - \mathrm{FPR}_y|\right)$$

$$= \frac{1}{2}\left(\left|\sum_{a\in\mathcal{A}}\left(\frac{p_{a'}}{p_{a',y}}\mathbb{I}[a=a'] - \frac{p_a}{p_y}\right)\mathbf{C}^a_{y,y}\right| + \left|\sum_{a\in\mathcal{A}}\sum_{y_i\neq y}\left(\frac{p_{a'}}{\sum_{y_j\neq y}p_{a',y_j}}\mathbb{I}[a=a'] - \frac{p_a}{\sum_{y_j\neq y}p_{y_j}}\right)\mathbf{C}^a_{y_i,y}\right|\right),$$

$$= \frac{1}{2}\left(\left|\sum_{a\in\mathcal{A}}\langle\mathbf{D}^{a,0}_{a',y},\mathbf{C}^a\rangle\right| + \left|\sum_{a\in\mathcal{A}}\langle\mathbf{D}^{a,1}_{a',y},\mathbf{C}^a\rangle\right|\right),$$

*where the entry in the y-th row and y-th column is $\frac{p_{a'}}{p_{a',y}}\mathbb{I}[a=a'] - \frac{p_a}{p_y}$ with all other elements set to $0$ for $\mathbf{D}^{a,0}_{a',y}\in\mathbb{R}^{m\times m}$, and the entry in the y-th column is $\frac{p_{a'}}{\sum_{y_j\neq y}p_{a',y_j}}\mathbb{I}[a=a'] - \frac{p_a}{\sum_{y_j\neq y}p_{y_j}}$ except for the y-th row with all other elements set to $0$ for $\mathbf{D}^{a,1}_{a',y}\in\mathbb{R}^{m\times m}$.*

## A.2 Group Fairness notations in FL

As noted in the main text, fairness at the level of each client's dataset (*local fairness*) differs from fairness across the aggregate dataset of all clients (*global fairness*). Local fairness is defined with respect to each client's individual data distribution $\mathbb{P}(X,Y,A\mid K)$, whereas global fairness is defined over the overall (aggregate) distribution $\mathbb{P}(X,Y,A)$. Motivated by approaches that employ group-specific confusion matrices for fairness [81, 60], we propose the **decentralized group-specific confusion matrices** $\mathbf{C}^{a,k}, a\in\mathcal{A}, k\in[N]$ to capture both global and local fairness across multiple data distributions within FL, with elements defined for $i,j\in[m]$ as $\mathbf{C}^{a,k}_{i,j}(h) := \mathbb{P}(Y=i,\widehat{Y}=j\mid A=a, K=k)$.

**Example 4.** *For DP criterion, the global DP fairness metric is defined as*

$$\mathscr{D}^g_{DP} = \max_{y\in[m]}\max_{a'\in\mathcal{A}}\left|\mathbb{P}(\widehat{Y}=y\mid A=a') - \mathbb{P}(\widehat{Y}=y)\right|,$$

*where* $\mathbb{P}(Y=y\mid A=a') = \sum_{k\in[N]}\sum_{i\in[m]}p_{k|a'}\mathbf{C}^{a',k}_{i,y}(h_k)$, *and* $\mathbb{P}(\widehat{Y}=y) = \sum_{a\in\mathcal{A}}\sum_{k\in[N]}\sum_{i\in[m]}p_{a,k}\mathbf{C}^{a,k}_{i,y}(h_k)$. *Hence, we have*

$$\mathscr{D}^g_{DP} = \max_{y\in[m]}\max_{a'\in\mathcal{A}}\left|\sum_{a\in\mathcal{A}}\sum_{k\in[N]}\sum_{i\in[m]}\left(p_{k|a'}\mathbb{I}[a=a'] - p_{a,k}\right)\mathbf{C}^{a,k}_{i,y}(h_k)\right|$$

$$= \max_{y\in[m]}\max_{a'\in\mathcal{A}}\left|\sum_{a\in\mathcal{A}}\sum_{k\in[N]}\langle\mathbf{D}^{a,k}_{a',y},\mathbf{C}^{a,k}(h_k)\rangle\right|,$$

*where* $\mathbf{D}^{a,k}_{a',y}\in\mathbb{R}^{m\times m}$, *and the y-th column elements of* $\mathbf{D}^{a,k}_{a',y}$ *are* $\mathbb{P}(K=k\mid A=a')\mathbb{I}[a=a'] - \mathbb{P}(A=a, K=k)$ *with all other elements set to* $0$.

*The local DP fairness metric for k-th client is defined as*

$$\mathscr{D}^k_{DP} = \max_{y\in[m]}\max_{a'\in\mathcal{A}}\left|\mathbb{P}(\widehat{Y}=y\mid A=a', K=k) - \mathbb{P}(\widehat{Y}=y\mid K=k)\right|,$$

*where* $\mathbb{P}(Y=y\mid A=a', K=k) = \sum_{i\in[m]}\mathbf{C}^{a',k}_{i,y}$, *and* $\mathbb{P}(\widehat{Y}=y\mid K=k) = \sum_{a\in\mathcal{A}}\sum_{i\in[m]}p_{a|k}\mathbb{P}(\widehat{Y}=y,Y=i\mid A=a, K=k)$. *Hence, we have*

$$\mathscr{D}^k_{DP} = \max_{y\in[m]}\max_{a'\in\mathcal{A}}\left|\sum_{a\in\mathcal{A}}\sum_{i\in[m]}\left(\mathbb{I}[a=a'] - p_{a|k}\right)\mathbf{C}^{a,k}_{i,y}(h_k)\right| = \max_{y\in[m]}\max_{a'\in\mathcal{A}}\left|\sum_{a\in\mathcal{A}}\langle\mathbf{D}^{a,k}_{a',y},\mathbf{C}^{a,k}(h_k)\rangle\right|,$$

*where* $\mathbf{D}^{a,k}_{a',y}\in\mathbb{R}^{m\times m}$, *and the y-th column elements of* $\mathbf{D}^{a,k}_{a',y}$ *are* $\mathbb{I}[a=a'] - \mathbb{P}(A=a\mid K=k)$ *with all other elements set to* $0$.

**Example 5.** *For EOP criterion, the global EOP fairness metric is defined as*

$$\mathscr{D}^g_{EOP} = \max_{y\in[m]} \max_{a'\in\mathcal{A}} \left| \mathbb{P}(\widehat{Y}=y \mid Y=y, A=a') - \mathbb{P}(\widehat{Y}=y \mid Y=y) \right|,$$

*where* $\mathbb{P}(Y=y \mid Y=y, A=a') = \sum_{k\in[N]} \frac{p_{a',k}}{p_{a',y}} \mathbf{C}^{a',k}_{i,y}(h_k)$, *and* $\mathbb{P}(\widehat{Y}=y \mid Y=y) = \sum_{a\in\mathcal{A}} \sum_{k\in[N]} \frac{p_{a,k}}{p_y} \mathbf{C}^{a,k}_{i,y}(h_k)$. *Hence, we have*

$$\mathscr{D}^g_{DP} = \max_{y\in[m]} \max_{a'\in\mathcal{A}} \left| \sum_{a\in\mathcal{A}} \sum_{k\in[N]} \left( \frac{p_{a',k}}{p_{a',y}}\mathbb{I}[a=a'] - \frac{p_{a,k}}{p_y} \right) \mathbf{C}^{a,k}_{i,y}(h_k) \right|$$

$$= \max_{y\in[m]} \max_{a'\in\mathcal{A}} \left| \sum_{a\in\mathcal{A}} \sum_{k\in[N]} \langle \mathbf{D}^{a,k}_{a',y}, \mathbf{C}^{a,k}(h_k) \rangle \right|,$$

*where* $\mathbf{D}^{a,k}_{a',y} \in \mathbb{R}^{m\times m}$, *and the entry in the y-th row and y-th column is* $\frac{p_{a',k}}{p_{a',y}}\mathbb{I}[a=a'] - \frac{p_{a,k}}{p_y}$ *with all other elements set to* $0$.

*The local EOP fairness metric for k-th client is defined as*

$$\mathscr{D}^k_{EOP} = \max_{y\in[m]} \max_{a'\in\mathcal{A}} \left| \mathbb{P}(\widehat{Y}=y \mid A=a', Y=y, K=k) - \mathbb{P}(\widehat{Y}=y \mid Y=y, K=k) \right|,$$

*where* $\mathbb{P}(\widehat{Y}=y \mid A=a', Y=y, K=k) = \frac{p_{a',k}}{p_{a',y,k}} \mathbf{C}^{p_{a',k}}_{i,y}$, *and* $\mathbb{P}(\widehat{Y}=y \mid Y=y, K=k) = \sum_{a\in\mathcal{A}} \frac{p_{a,k}}{p_{y,k}} \mathbb{P}(\widehat{Y}=y, Y=i \mid A=a, K=k)$. *Hence, we have*

$$\mathscr{D}^k_{EOP} = \max_{y\in[m]} \max_{a'\in\mathcal{A}} \left| \sum_{a\in\mathcal{A}} \left( \frac{p_{a',k}}{p_{a',y,k}}\mathbb{I}[a=a'] - \frac{p_{a,k}}{p_{y,k}} \right) \mathbf{C}^{a,k}_{i,y}(h_k) \right| = \max_{y\in[m]} \max_{a'\in\mathcal{A}} \left| \sum_{a\in\mathcal{A}} \langle \mathbf{D}^{a,k}_{a',y}, \mathbf{C}^{a,k}(h_k) \rangle \right|,$$

*where* $\mathbf{D}^{a,k}_{a',y} \in \mathbb{R}^{m\times m}$, *and the entry in the y-th row and y-th column is* $\frac{p_{a',k}}{p_{a',y,k}}\mathbb{I}[a=a'] - \frac{p_{a,k}}{p_{y,k}}$ *with all other elements set to* $0$.

**Example 6.** *For EOP criterion, the global EOP fairness metric is defined as*

$$\mathscr{D}^g_{EO} = \max_{y\in[m]} \max_{a'\in\mathcal{A}} \frac{1}{2} \left( \left| \mathbb{P}(\widehat{Y}=y \mid Y=y, A=a') - \mathbb{P}(\widehat{Y}=y \mid Y=y) \right| \right.$$

$$\left. + \left| \mathbb{P}(\widehat{Y}=y \mid Y\neq y, A=a') - \mathbb{P}(\widehat{Y}=y \mid Y\neq y) \right| \right),$$

*where* $\mathbb{P}(Y=y \mid Y\neq y, A=a') = \sum_{k\in[N]} \sum_{y_i\neq y} \frac{p_{a',k}}{\sum_{y_j\neq y} p_{a',y_j}} \mathbf{C}^{a',k}_{y_i,y}(h_k)$, *and* $\mathbb{P}(\widehat{Y}=y \mid Y\neq y) = \sum_{a\in\mathcal{A}} \sum_{k\in[N]} \sum_{y_i\neq y} \frac{p_{a,k}}{\sum_{y_j\neq y} p_{y_j}} \mathbf{C}^{a,k}_{y_i,y}(h_k)$. *Hence, we have*

$$\mathscr{D}^g_{EO} = \max_{y\in[m]} \max_{a'\in\mathcal{A}} \frac{1}{2} \left( \left| \sum_{a\in\mathcal{A}} \sum_{k\in[N]} \langle \mathbf{D}^{a,k,0}_{a',y}, \mathbf{C}^{a,k}(h_k) \rangle \right| + \left| \sum_{a\in\mathcal{A}} \sum_{k\in[N]} \langle \mathbf{D}^{a,k,1}_{a',y}, \mathbf{C}^{a,k}(h_k) \rangle \right| \right),$$

*where the entry in the y-th row and y-th column is* $\frac{a',k}{p_{a',y}}\mathbb{I}[a=a'] - \frac{p_{a,k}}{p_y}$ *with all other elements set to* $0$ *for* $\mathbf{D}^{a,k,0}_{a',y} \in \mathbb{R}^{m\times m}$, *and the entry in the y-th column is* $\frac{p_{a',k}}{\sum_{y_j\neq y} p_{a',y_j}}\mathbb{I}[a=a'] - \frac{p_{a,k}}{\sum_{y_j\neq y} p_{y_j}}$ *except for the y-th row with all other elements set to* $0$ *for* $\mathbf{D}^{a,k,1}_{a',y} \in \mathbb{R}^{m\times m}$.

*The local EOP fairness metric for k-th client is defined as*

$$\mathscr{D}^k_{EO} = \max_{y\in[m]} \max_{a'\in\mathcal{A}} \frac{1}{2} \left( \left| \mathbb{P}(\widehat{Y}=y \mid Y=y, A=a', K=k) - \mathbb{P}(\widehat{Y}=y \mid Y=y, K=k) \right| \right.$$

$$\left. + \left| \mathbb{P}(\widehat{Y}=y \mid Y\neq y, A=a', K=k) - \mathbb{P}(\widehat{Y}=y \mid Y\neq y, K=k) \right| \right),$$

*where* $\mathbb{P}(\widehat{Y} = y \mid A = a', Y \neq y, K = k) = \sum_{y_i \neq y} \frac{p_{a',k}}{\sum_{y_j \neq y} p_{a',y_j,k}} \mathbf{C}^{p_{a',k}}_{y_i,y}$, *and* $\mathbb{P}(\widehat{Y} = y \mid Y \neq$
$y, K = k) = \sum_{a \in \mathcal{A}} \frac{p_{a,k}}{\sum_{y_j \neq y} p_{y_j,k}} \mathbb{P}(\widehat{Y} = y, Y = i \mid A = a, K = k)$. *Hence, we have*

$$\mathscr{D}^g_{EO} = \max_{y \in [m]} \max_{a' \in \mathcal{A}} \frac{1}{2} \left( \left| \sum_{a \in \mathcal{A}} \sum_{k \in [N]} \langle \mathbf{D}^{a,k,0}_{a',y}, \mathbf{C}^{a,k}(h_k) \rangle \right| + \left| \sum_{a \in \mathcal{A}} \sum_{k \in [N]} \langle \mathbf{D}^{a,k,1}_{a',y}, \mathbf{C}^{a,k}(h_k) \rangle \right| \right),$$

*where the entry in the y-th row and y-th column is* $\frac{a',k}{p_{a',y,k}} \mathbb{I}[a = a'] - \frac{p_{a,k}}{p_{y,k}}$ *with all other elements set to 0 for* $\mathbf{D}^{a,k,0}_{a',y} \in \mathbb{R}^{m \times m}$, *and the entry in the y-th column is* $\frac{p_{a',k}}{\sum_{y_j \neq y} p_{a',y_j,k}} \mathbb{I}[a = a'] - \frac{p_{a,k}}{\sum_{y_j \neq y} p_{y_j,k}}$
*except for the y-th row with all other elements set to 0 for* $\mathbf{D}^{a,k,1}_{a',y} \in \mathbb{R}^{m \times m}$.

### A.3 Other Fairness Notations

Fairness notions formulated as ratios metrics can be converted into linear constraints under certain conditions, and our framework is well suited to enforce these fairness constraints in federated learning environments. Specifically, the ratios metric or constraints, which are formulated as $\left| \frac{\sum_a \langle \mathbf{D}^a, \mathbf{C}^a(h) \rangle}{\sum_a \langle \mathbf{G}^a, \mathbf{C}^a(h) \rangle} \right| \leq \xi$ with constant metrix $\mathbf{D}^a, \mathbf{G}^a$ and group-specific confusion matrix $\mathbf{C}^a(h)$ depend on classifier $h$, can certainly be transformed into multiple linear constraints if the sign of $\sum_a \langle \mathbf{G}^a, \mathbf{C}^a(h) \rangle$ is unchanged for any $h$. When the denominator's sign is uncertain, the feasible domain of $\mathbf{C}^a(h)$ is non-convex, precluding its expression via linear constraints. In fact, since each entry of $\mathbf{C}^a(h)$ lies in [0,1], whenever the entries of $\mathbf{G}^a$ are sign-consistent, the corresponding ratio constraint admits a linear-constraint representation. For example, the Calibration within Groups (CG) which was proposed in [5] and further explored in [46], is a fairness metric in binary classification and can be formulated as $\frac{FN^a}{FN^a + TN^a} = v_0$; $\frac{TP^a}{TP^a + FP^a} = v_1$, where $TP^a, FP^a, FN^a, TN^a$ are derived from binary group-specific confusion matrix $\mathbf{C}^a$ and $0 \leq v_0 < v_1 \leq 1$ and have no implicit dependence on any entries of the fairness-confusion tensor. Because this fairness criterion appears as a ratio metric and every element of corresponding $\mathbf{G}^a$ is non-negative, it admits a linear-constraint representation and can be realized in our proposed distributed framework. Moreover, the ratio metrics presented in [3] also can be formulated into multiple linear constraints based on the above analysis.

# B Proofs and Discussion in Section 4

## B.1 Proof of Proposition 1

This section provides the proof of Proposition 1. The proof is primarily inspired by the characterization of the Bayes-optimal fair classifier in the centralized fair machine learning literature (e.g. Theorem 3.1 of [81], Proposition 10 of [60]).

*Proof.* We begin by casting the primal problem (1) into an optimization problem defined on the Cartesian product of confusion matrices. Consider the the set of achievable confusion matrices:

$$\mathcal{C}^{|\mathcal{A}| \times N} := \{\mathbf{C}^{|\mathcal{A}| \times N}(\mathbf{h}) := \{\mathbf{C}^{a,k}(h_k)\}_{a \in \mathcal{A}, k \in [N]} : \mathbf{h} \in \mathcal{H}^N\},$$

where $\mathcal{C}^{|\mathcal{A}| \times N}$ be the product space of all confusion matrices $\mathbf{C}^{a,k}$ corresponding to sensitive group $a \in \mathcal{A}$ and $k \in [N]$ associated with a given instance $\mathbf{h} \in \mathcal{H}^N$ of the problem. It is clear that the performance metric $\mathcal{R}$ and fairness metrics $\mathscr{D}^g, \mathscr{D}^k, k \in [N]$ are continuous and bounded to $\mathbf{C}^{|\mathcal{A}| \times N}(\mathbf{h}) := \{\mathbf{C}^{a,k}(h_k)\}_{a \in \mathcal{A}, k \in [N]}$.

**Convexity of $\mathcal{C}^{|\mathcal{A}| \times N}$.** Let $\forall \mathbf{C}_1, \mathbf{C}_2 \in \mathcal{C}^{|\mathcal{A}| \times N}$ be realized by classifier tuples $\mathbf{h}_1, \mathbf{h}_2$. For any $\omega \in [0, 1]$, define the mixed classifier $\mathbf{h}' = \omega \mathbf{h}_1 + (1 - \omega) \mathbf{h}_2$,. By linearity of performance and fairness metrics, its confusion matrix satisfies

$$\mathbf{C}(\mathbf{h}') = \omega \mathbf{C}(\mathbf{h}_1) + (1 - \omega) \mathbf{C}(\mathbf{h}_2) = \omega \mathbf{C}_1 + (1 - \omega) \mathbf{C}_2 = \mathbf{C}_\omega.$$

Thus every convex combination of $\mathbf{C}_1$ and $\mathbf{C}_2$ lies in $\mathcal{C}^{|\mathcal{A}| \times N}$, establishing convexity.

**Deterministic classifiers.** It can be seen that, for any linear objective $\phi_{\mathbf{L}}(\mathbf{C}^{|\mathcal{A}| \times N}(\mathbf{h})) = \sum_{a \in \mathcal{A}} \sum_{k \in [N]} \langle \mathbf{L}^{a,k}, \mathbf{C}^{a,k}(h_k) \rangle$, there is a deterministic classifiers $\mathbf{h}^* = (h_1^*, \cdots, h_N^*)$ that is optimal for $\phi_{\mathbf{L}}$ (see proof in B.2). By the supporting-hyperplane theorem [9] for compact convex sets, for each point $\mathbf{C}_b = \{\mathbf{C}_b^{a,k}\}_{a \in \mathcal{A}, k \in [N]} \in \partial \mathcal{C}^{|\mathcal{A}| \times N}$, there exists a nonzero collection of matrices $\mathbf{L}_b = \{\mathbf{L}_b^{a,k}\}_{a \in \mathcal{A}, k \in [N]}$ constitutes a hyperplane, such that for every $\mathbf{C} = \{\mathbf{C}^{a,k}\} \in \mathcal{C}^{|\mathcal{A}| \times N}$ we have $\sum_{a \in \mathcal{A}} \sum_{k=1}^N \langle \mathbf{L}_b^{a,k}, \mathbf{C}_b^{a,k} \rangle \leq \sum_{a \in \mathcal{A}} \sum_{k=1}^N \langle \mathbf{L}_b^{a,k}, \mathbf{C}^{a,k} \rangle$ which is precisely the desired supporting-hyperplane condition at $\mathbf{C}_b$. In other words, we arrive at the conclusion that each boundary point of $\mathcal{C}^{|\mathcal{A}| \times N}$ can be achieved by deterministic classifiers $\mathbf{h}' = (h_1', \cdots, h_N')$.

**Combination of deterministic classifiers.** Since $\mathcal{C}^{|\mathcal{A}| \times N}$ is compact and convex, we know that its extreme points fall in its boundary. By the Krein-Milman theorem [61], we have that $\mathcal{C}^{|\mathcal{A}| \times N}$ is equal to the convex hull of its extreme points. We further have from Caratheodory's theorem [9] that any $\mathbf{C} \in \mathcal{C}^{|\mathcal{A}| \times N}$ can be expressed as a convex combination of $d_k = |\mathcal{A}| N m^2$ points in the extreme point set, where each extreme point can be characterized by deterministic classifiers. Hence, we have proved that the optimal solution $\mathbf{h}$ can be represented by the convex combination of deterministic classifiers. $\square$

**Discussion on feasibility.** The only condition for the above theorem to hold is that the feasible set is non-empty, which is clearly satisfied by the mentioned fairness constraints, DP, EOP, and EO. For these fairness criteria, the classifier that always predicts a single, fixed label $y'$ trivially meets $\xi^g = 0, \xi^k = 0, k \in [N]$, and hence satisfies the fairness constraints.

**The number of deterministic classifiers.** As for the number of deterministic classifiers required, the parameter $d^k$ in the proof scales with the number of nonzero entries in the linear performance and fairness constraints [60]. Since each matrix $\mathbf{D}^{a,k}$ in our fairness formulation is zero except for one column, we in fact need far fewer than $|\mathcal{A}| N m^2$ classifiers. Moreover, under the continuity assumption 1, this number can be reduced even further [81].

## B.2 Proof of Proposition 2

*Proof.* We denote $p_a := \mathbb{P}(A = a)$, $p_k := \mathbb{P}(K = k)$, $p_{a,k} := \mathbb{P}(A = a, K = k)$, and $\mathcal{P}_k^X := \mathbb{P}(X | K = k)$. Consider the form Lagrangian function of federated Bayes-optimal fair classification

problem (1),

$\mathcal{L}(\mathbf{h}, \lambda, \mu)$

$$
\begin{aligned}
&= \sum_{k=1}^{N} \sum_{a \in \mathcal{A}} p_{a,k} \langle \mathbf{1} - \mathbf{I}, \mathbf{C}^{a,k}(h_k) \rangle + \sum_{u_g \in \mathcal{U}_g} (\lambda_{u_g}^{(1)} - \lambda_{u_g}^{(2)}) \sum_{k=1}^{N} \sum_{a \in \mathcal{A}} \langle \mathbf{D}_{u_g}^{a,k}, \mathbf{C}^{a,k}(h_k) \rangle \\
&\quad + \sum_{k=1}^{N} \sum_{u_k \in \mathcal{U}_k} (\mu_{k,u_k}^{(1)} - \mu_{k,u_k}^{(2)}) \sum_{a \in \mathcal{A}} \langle \mathbf{D}_{u_k}^{a,k}, \mathbf{C}^{a,k}(h_k) \rangle - \sum_{u_g \in \mathcal{U}_g} (\lambda_{u_g}^{(1)} + \lambda_{u_g}^{(2)}) \xi^g \\
&\quad - \sum_{k \in [N]} \sum_{u_k \in \mathcal{U}_k} (\mu_{k,u_k}^{(1)} + \mu_{k,u_k}^{(2)}) \xi^k \\
&= \sum_{k=1}^{N} \sum_{a \in \mathcal{A}} \left\langle p_{a,k}(\mathbf{1} - \mathbf{I}) + \sum_{u_g \in \mathcal{U}_g} (\lambda_{u_g}^{(1)} - \lambda_{u_g}^{(2)}) \mathbf{D}_{u_g}^{a,k} + \sum_{u_k \in \mathcal{U}_k} (\mu_{k,u_k}^{(1)} - \mu_{k,u_k}^{(2)}) \mathbf{D}_{u_k}^{a,k}, \mathbf{C}^{a,k}(h_k) \right\rangle \\
&\quad - \sum_{u_g \in \mathcal{U}_g} (\lambda_{u_g}^{(1)} + \lambda_{u_g}^{(2)}) \xi^g - \sum_{k \in [N]} \sum_{u_k \in \mathcal{U}_k} (\mu_{k,u_k}^{(1)} + \mu_{k,u_k}^{(2)}) \xi^k.
\end{aligned}
$$

The inner problem of Lagrangian dual ask we to solve $\min_{\mathbf{h} \in \mathcal{H}} \mathcal{L}(\mathbf{h}, \lambda, \mu)$ given element-wise non-negative dual parameter $\lambda$ and $\mu$, which can be formulated as

$$
\max_{\mathbf{h} \in \mathcal{H}^N} V(\mathbf{h}, \lambda, \mu) = \sum_{k=1}^{N} \sum_{a \in \mathcal{A}} p_{a,k} \left\langle \mathbf{M}^{\lambda,\mu}(a, k), \mathbf{C}^{a,k}(h_k) \right\rangle,
$$

where $\mathbf{M}^{\lambda,\mu}(a, k) := \mathbf{I} - \frac{1}{p_{a,k}} \left[ \sum_{u_g \in \mathcal{U}_g} (\lambda_{u_g}^{(1)} - \lambda_{u_g}^{(2)}) \mathbf{D}_{u_g}^{a,k} - \sum_{u_k \in \mathcal{U}_k} (\mu_{k,u_k}^{(1)} - \mu_{k,u_k}^{(2)}) \mathbf{D}_{u_k}^{a,k} \right].$

The next step is to derive the optimal solution of $\max_{\mathbf{h} \in \mathcal{H}^N} V(\mathbf{h}, \lambda, \mu)$. For this purpose, we perform manipulations of $H$ to reveal its clear relationship with the personalized classifier $\mathbf{h} = (h_1, \ldots, h_N)$. Denote the condition distribution of $X$ given sensitive attribute $A = a$ on client $K = k$ as $\mathcal{P}_{a,k}^X$, i.e., $\mathcal{P}_{a,k}^X := \mathbb{P}(X \mid A = a, K = k)$, we have

$$
\begin{aligned}
V(\mathbf{h}, \lambda, \mu) &= \sum_{k=1}^{N} \sum_{a \in \mathcal{A}} p_{a,k} \left\langle \mathbf{M}^{\lambda,\mu}(a, k), \mathbf{C}^{a,k}(h_k) \right\rangle \\
&= \sum_{k=1}^{N} \sum_{a \in \mathcal{A}} p_{a,k} \int_{\mathcal{X}} [\eta(X, a, k)]^\top \mathbf{M}^{\lambda,\mu}(a, k) h_k(x) d\mathcal{P}_{a,k}^X(x) \\
&= \sum_{k=1}^{N} \sum_{a \in \mathcal{A}} p_{a,k} \mathbb{E}_{X|A=a,K=k} \left[ [\eta(X, a, k)]^\top \mathbf{M}^{\lambda,\mu}(a, k) h_k(x) \right] \\
&= \mathbb{E}_{A,K} \left[ \mathbb{E}_{X|A,K} \left[ [\eta(X, A, K)]^\top \mathbf{M}^{\lambda,\mu}(A, K) h_K(X) \right] \right] \\
&= \mathbb{E}_{X,A,K} \left[ [\eta(X, A, K)]^\top \mathbf{M}^{\lambda,\mu}(A, K) h_K(X) \right] \\
&= \mathbb{E}_{X,K} \left[ \mathbb{E}_{A|X,K} \left[ [\eta(X, A, K)]^\top \mathbf{M}^{\lambda,\mu}(A, K) h_K(X) \right] \right] \\
&= \mathbb{E}_{X,K} \left[ \sum_{a \in \mathcal{A}} \mathbb{P}(A = a \mid X, K) [\eta(X, a, K)]^\top \mathbf{M}^{\lambda,\mu}(A, K) h_K(X) \right] \\
&= \sum_{k=1}^{N} p_k \mathbb{E}_{x \sim \mathcal{P}_k^X} \left[ \sum_{a \in \mathcal{A}} \mathbb{P}(A = a \mid x, k) [\eta(x, a, k)]^\top \mathbf{M}^{\lambda,\mu}(a, k) h_k(x) \right].
\end{aligned}
$$

To derive the optimal solution of the inner optimization problem, it suffices to perform a pointwise maximization of the above objective: for fixed $x, k$, the classifier $h_k(x)$ selects the label that maximizes the term inside the expectation, i.e.,

$$
h_k^*(x) = e_y, \ y \in \arg\max_{j \in [m]} \left( \sum_{a \in \mathcal{A}} \mathbb{P}(A = a|x, k) \left[ \mathbf{M}^{\mu,\lambda}(a, k) \right]^\top \eta(x, a, k) \right)_j.
$$

Thus, we have finished the proof of Proposition 2. $\qquad\square$

## B.3 Proof of Proposition 3 and Further Exploration

In this section, we prove that the representation in (4) is calibrated for both unified and personalized inner optimization problem. We begin by presenting the following lemma.

**Lemma B.1.** *For any categorical distribution characterized by $\mathbf{p} \in \Delta_m$, the minimizer of the expected risk*

$$\mathbb{E}_{y \sim \mathbf{p}} \left[ -\log(\mathbf{q}_y) \right] = -\sum_{i=1}^{m} \mathbf{p}_i \log(\mathbf{q}_i)$$

*over all $\mathbf{q} \in \Delta_m$ is unique and achieved at $\mathbf{p} = \mathbf{q}$.*

This lemma is commonly used in the design of multiclass loss functions [77, 57, 59].

### B.3.1 Proof of Proposition 3

*Proof.* We aim to prove that for any fixed $x \in \mathcal{X}, k \in [N]$, the optimal personalized scoring function $\mathbf{s}_k^* : \mathcal{X} \to \mathbb{R}^m$ that minimizes the expected loss $\ell_k(y, \mathbf{s}(x), a)$ over the local data distribution $\mathbb{P}(X, A, Y \mid K = k)$ recovers the personalized federated Bayes-optimal classifier $h_k^*(x)$ in Proposition 2.

It is equivalent to show that, for any $x$:

$$\arg\max_{j \in [m]} [\mathbf{s}_k^*(x)]_j \subseteq \arg\max_{j \in [m]} \left( \sum_{a \in \mathcal{A}} P(A = a|x, k) \left[ \mathbf{M}^{\mu,\lambda}(a, k) \right]^\top \eta(x, a, k) \right)_j.$$

To this end, by leveraging the properties of conditional expectation, the cost-sensitive loss is reformulated as a function of the marginal distribution $(X, K)$:

$$\mathbb{E}_{(x,y,a,k) \sim (X,Y,A,K)}[\ell_k(y, \mathbf{s}(x), a)] = -\mathbb{E}_{X,Y,A,K} \left[ \sum_{i=1}^{m} \overline{\mathbf{M}}_{Y,i}^{\lambda,\mu}(A, K) \log \frac{\exp([\mathbf{s}(X)]_i)}{\sum_{j=1}^{m} \exp([\mathbf{s}(X)]_j)} \right]$$

$$= -\mathbb{E}_{X,A,K} \left[ \mathbb{E}_{Y|X,A,K} \left[ \sum_{i=1}^{m} \overline{\mathbf{M}}_{Y,i}^{\lambda,\mu}(A, K) \log \frac{\exp([\mathbf{s}(X)]_i)}{\sum_{j=1}^{m} \exp([\mathbf{s}(X)]_j)} \right] \right]$$

$$= -\mathbb{E}_{X,A,K} \left[ \sum_{y \in [m]} \mathbb{P}(Y = y \mid X, A, K) \sum_{i=1}^{m} \overline{\mathbf{M}}_{y,i}^{\lambda,\mu}(A, K) \log \frac{\exp([\mathbf{s}(X)]_i)}{\sum_{j=1}^{m} \exp([\mathbf{s}(X)]_j)} \right]$$

$$= -\mathbb{E}_{X,K} \left[ \mathbb{E}_{A|X,K} \left[ \sum_{y \in [m]} \eta_y(X, A, K) \sum_{i=1}^{m} \overline{\mathbf{M}}_{y,i}^{\mu,\lambda}(A, K) \log \frac{\exp([\mathbf{s}(X)]_i)}{\sum_{j=1}^{m} \exp([\mathbf{s}(X)]_j)} \right] \right]$$

$$= \sum_{k=1}^{N} p_k \mathbb{E}_{x \sim \mathcal{P}_k^X} \left[ -\sum_{a \in \mathcal{A}} \mathbb{P}(A = a|x, k) \sum_{i=1}^{m} \left( \left[ \overline{\mathbf{M}}^{\mu,\lambda}(a, k) \right]^\top \eta(x, a, k) \right)_i \log \frac{\exp([\mathbf{s}(x)]_i)}{\sum_{j=1}^{m} \exp([\mathbf{s}(x)]_j)} \right]$$

Denoting $\mathbf{v}_i(x, k) := \sum_{a \in \mathcal{A}} \mathbb{P}(A = a|x, k) \left( \left[ \overline{\mathbf{M}}^{\mu,\lambda}(a, k) \right]^\top \eta(x, a, k) \right)_i$, we have

$$\mathbb{E}_{X,Y,A,K}[\ell_k(y, \mathbf{s}(x), a)] = \mathbb{E}_{X,K} \left[ -c_{X,K} \sum_{i=1}^{m} \frac{\mathbf{v}_i(X, K)}{\sum_{j \in [m]} \mathbf{v}_j(X, K)} \log \frac{\exp([\mathbf{s}(X)]_i)}{\sum_{j=1}^{m} \exp([\mathbf{s}(X)]_j)} \right]$$

where $c_{x,k} = \sum_{j \in [m]} \mathbf{v}_j(x, k)$ can be treated as a constant for fixed $x, k$. According to Lemma B.1, given fixed $x, k$, an optimal personalized classifier $\mathbf{s}_k^*(x)$ minimizing the cost-sensitive loss point-wise satisfies

$$\frac{\mathbf{v}_i(x, k)}{\sum_{j \in [m]} \mathbf{v}_j(x, k)} = \frac{\exp([\mathbf{s}_k^*(x)]_i)}{\sum_{j=1}^{m} \exp([\mathbf{s}_k^*(x)]_j)}, \quad \forall i \in [m].$$

It presents that, for all $i \in [m]$, since $\sum_{i \in [m]} \eta_i(x, a, k) = 1$,

$$
\begin{aligned}
[\mathbf{s}_k^*(x)]_i = \mathbf{v}_i(x, k) &= \sum_{a \in \mathcal{A}} \mathbb{P}(A = a | x, k) \left( \left[ \overline{\mathbf{M}}^{\mu, \lambda}(a, k) \right]^\top \eta(x, a, k) \right)_i \\
&= \sum_{a \in \mathcal{A}} \mathbb{P}(A = a | x, k) \left( \left[ \mathbf{M}^{\mu, \lambda}(a, k) + \alpha \mathbf{1}_{m \times m} \right]^\top \eta(x, a, k) \right)_i \\
&= \sum_{a \in \mathcal{A}} \mathbb{P}(A = a | x, k) \left( \left[ \mathbf{M}^{\mu, \lambda}(a, k) \right]^\top \eta(x, a, k) + \alpha \mathbf{1}_m \right)_i \\
&= \sum_{a \in \mathcal{A}} \mathbb{P}(A = a | x, k) \left( \left[ \mathbf{M}^{\mu, \lambda}(a, k) \right]^\top \eta(x, a, k) \right)_i + \alpha.
\end{aligned}
$$

Hence,

$$
\arg \max_{y \in [m]} [\mathbf{s}_k^*(x)]_y \subseteq \arg \max_{y \in [m]} \left( \sum_{a \in \mathcal{A}} \mathbb{P}(A = a | x, k) \left[ \mathbf{M}^{\mu, \lambda}(a, k) \right]^\top \eta(x, a, k) \right)_y.
$$

The personalized classifier $h_k^*(x) \in \arg \min_{y \in [m]} [\mathbf{s}_k^*(x)]_y$ recovers that in Proposition 2. We finish the proof. $\qquad \square$

### B.3.2 Exploration of Calibrated Loss for Unified Bayes-Optimal Classifier

We start from the inner optimization objective $V(\mathbf{h}, \lambda, \mu)$,

$$
\begin{aligned}
V(\mathbf{h}, \lambda, \mu) &= \sum_{k=1}^N \sum_{a \in \mathcal{A}} p_{a,k} \left\langle \mathbf{M}^{\lambda, \mu}(a, k), \mathbf{C}^{a,k}(h_k) \right\rangle \\
&= \mathbb{E}_{X, A, K} \left[ [\eta(X, A, K)]^\top \mathbf{M}^{\lambda, \mu}(A, K) h(X) \right] \\
&= \mathbb{E}_X \left[ \mathbb{E}_{A, K | X} \left[ [\eta(X, A, K)]^\top \mathbf{M}^{\lambda, \mu}(A, K) h(X) \right] \right] \\
&= \mathbb{E}_X \left[ \sum_{a \in \mathcal{A}} \sum_{k=1}^N \mathbb{P}(A = a, K = k \mid X) [\eta(X, a, k)]^\top \mathbf{M}^{\lambda, \mu}(a, k) h(X) \right]
\end{aligned}
$$

To derive the optimal solution of the inner optimization problem, it suffices to perform a point-wise maximization of the above objective: for fixed $x$, the classifier $h(x)$ selects the label that maximizes the term inside the expectation, i.e.,

$$
h^*(x) = e_y, \ y \in \arg \max_{j \in [m]} \left( \sum_{a \in \mathcal{A}} \sum_{k=1}^N \mathbb{P}(A = a, K = k \mid X) [\eta(X, a, k)]^\top \mathbf{M}^{\lambda, \mu}(a, k) h(X) \right)_j.
$$

Consider the calibrated loss function in (4),

$$
\mathbb{E}_{(x,y,a,k)\sim(X,Y,A,K)}[\ell_k(y,\mathbf{s}(x),a)] = -\mathbb{E}_{X,Y,A,K}\left[\sum_{i=1}^{m}\overline{\mathbf{M}}_{Y,i}^{\lambda,\mu}(A,K)\log\frac{\exp([\mathbf{s}(X)]_i)}{\sum_{j=1}^{m}\exp([\mathbf{s}(X)]_j)}\right]
$$

$$
= -\mathbb{E}_{X,A,K}\left[\mathbb{E}_{Y|X,A,K}\left[\sum_{i=1}^{m}\overline{\mathbf{M}}_{Y,i}^{\lambda,\mu}(A,K)\log\frac{\exp([\mathbf{s}(X)]_i)}{\sum_{j=1}^{m}\exp([\mathbf{s}(X)]_j)}\right]\right]
$$

$$
= -\mathbb{E}_{X,A,K}\left[\sum_{y\in[m]}\mathbb{P}(Y=y\mid X,A,K)\sum_{i=1}^{m}\overline{\mathbf{M}}_{y,i}^{\lambda,\mu}(A,K)\log\frac{\exp([\mathbf{s}(X)]_i)}{\sum_{j=1}^{m}\exp([\mathbf{s}(X)]_j)}\right]
$$

$$
= -\mathbb{E}_{X}\left[\mathbb{E}_{A,K|X}\left[\sum_{y\in[m]}\eta_y(X,A,K)\sum_{i=1}^{m}\overline{\mathbf{M}}_{y,i}^{\mu,\lambda}(A,K)\log\frac{\exp([\mathbf{s}(X)]_i)}{\sum_{j=1}^{m}\exp([\mathbf{s}(X)]_j)}\right]\right]
$$

$$
= \mathbb{E}_{x\sim\mathbb{P}(X)}\left[-\sum_{a\in\mathcal{A}}\sum_{k=1}^{N}\mathbb{P}(A=a,K=k|x)\sum_{i=1}^{m}\left(\left[\overline{\mathbf{M}}^{\mu,\lambda}(a,k)\right]^{\top}\eta(x,a,k)\right)_i\log\frac{\exp([\mathbf{s}(x)]_i)}{\sum_{j=1}^{m}\exp([\mathbf{s}(x)]_j)}\right]
$$

By leveraging Lemma B.1, and employing an approach analogous to that used in the proof of Proposition 3, it is clear that we can obtain

$$
[\mathbf{s}^*(x)]_i = \sum_{a\in\mathcal{A}}\sum_{k=1}^{N}\mathbb{P}(A=a,K=k|x)\left(\left[\mathbf{M}^{\mu,\lambda}(a,k)\right]^{\top}\eta(x,a,k)\right)_i + \alpha.
$$

Hence,

$$
\arg\max_{y\in[m]}[\mathbf{s}^*(x)]_y \subseteq \arg\max_{y\in[m]}\left(\sum_{a\in\mathcal{A}}\sum_{k=1}^{N}\mathbb{P}(A=a,K=k|x)\left[\mathbf{M}^{\mu,\lambda}(a,k)\right]^{\top}\eta(x,a,k)\right)_y.
$$

The unified classifier $h^*(x) \in \arg\min_{y\in[m]}[\mathbf{s}^*(x)]_y$ recovers that in Proposition 2. We have shown that the loss $\ell_k$ in (4) is also calibrated for the unified federated Bayes-optimal fair classifier. $\qquad\square$

### B.4  The Complete Formulation of Theorem 4 with Its Proof.

In this subsection, we fully articulate Theorem 4 through Theorem 7 and Theorem 8, which together form an extended version of the result in Theorem 4. Before proceeding, we first clarify some notations and assumptions.

With a little abuse of notation, let $f_{k,ens}^t(x) := f(x;\phi_k^t) + f(x;\theta^t)$, $f(x;\phi_k^t) = \text{softmax}(\mathbf{s}_k(x;\phi_k^t))$ and $f(x;\theta^t) = \text{softmax}(\mathbf{s}_k(x;\theta^t))$. The local objective for

$$
L_k^t(f(x;\theta^t)) := -\sum_{i=1}^{n_k}\sum_{y'=1}^{m}\overline{\mathbf{M}}_{y_i,y'}^{\lambda^t,\mu^t}(a_i,k)\log[f(x_i;\theta^t)], \ k\in[N],
$$

which is similar to $L_k^t(f(x;\phi_k^t))$ and $L_k^t(f_{k,ens}^t(x))$.

**Assumption 2.** The local loss function $L_1^t,\cdots,L_N^t$ are convex, $\beta$-smooth and bounded by $B_L$ to model parameters $\phi_k$ and $\theta$, $t\in[T]$.

**Assumption 3.** Let $\mathcal{B}_k^{t,r}$ be sampled from the $k$-th device's local data uniformly at random. The variance of stochastic gradients in each client is bounded:

$$
\mathbb{E}\left\|\nabla_\theta L_k^t(f(x;\theta);\mathcal{B}_k^{t,r}) - \nabla L_k^t(f(x;\theta))\right\|^2 \leq \sigma^2
$$

for $k\in[N], t\in[T]$.

Assumption 2 and 3 are standard in the convergence analysis of federated model [51, 49, 74]. Now we present Theorem 7 and Theorem 8, which together constitute an extended form of Theorem 4.

**Theorem 7.** *Under assumptions 2 and 3, for the ensemble personalized models, denoting $\theta^* :=$ $\arg\min_\theta \sum_{t=1}^T \sum_{k=1}^N p_k L_k^t(f(x;\theta))$ and $B_k = R\beta B_L + \frac{3}{2}\beta B_L + \frac{3}{4}\sigma^2$, the following cumulative global regret upper bound of all clients is guaranteed:*

$$\frac{1}{T}\sum_{t=1}^T \sum_{k=1}^N \hat{p}_k \mathbb{E}[L_k^t(f_{k,ens}^t) - L_k^t(f(x;\theta^*))] \leq \frac{\|\theta^*\|^2}{2\eta RT} + \eta B_k + \frac{\log(2)}{\eta_w T} + \eta_w B_L$$

*while denoting $\phi_k^* := \arg\min_{\phi_k} \sum_{t=1}^T L_k^t(f(x;\phi_k))$, the k-th client achieves the following personalized regret upper bound:*

$$\frac{1}{T}\sum_{t=1}^T \mathbb{E}[L_k^t(f_{k,ens}^t) - L_k^t(f(x;\phi_k^*))] \leq \frac{\|\phi_k^*\|^2}{2\eta RT} + \eta B_k + \frac{\log(2)}{\eta_w T} + \eta_w B_L.$$

**Theorem 8.** *Suppose that personalized models achieve a $\rho_t$-approximate optimal response at iteration $t$, namely $\widehat{\mathcal{L}}(\mathbf{h}^t, \lambda^t, \mu^t) \leq \min_{\mathbf{h}} \widehat{\mathcal{L}}(\mathbf{h}, \lambda^t, \mu^t) + \rho_t$, denoting $\bar{\rho} = \sum_{t=1}^T \rho_t/T$, then the sequences of model and bounded dual parameters comprise an approximate mixed Nash equilibrium:*

$$\max_{\lambda^*,\mu^*} \frac{1}{T}\sum_{t=1}^T \widehat{\mathcal{L}}\left(\mathbf{h}^t, \lambda^*, \mu^*\right) - \inf_{\mathbf{h}^*} \frac{1}{T}\sum_{t=1}^T \widehat{\mathcal{L}}\left(\mathbf{h}^*, \lambda^t, \mu^t\right) \leq \epsilon = \bar{\rho} + 16B_d^2\sqrt{\frac{1}{T}}. \tag{9}$$

### B.4.1 Proof of Theorem 7

The proof of Theorem 7 comprises proofs of the global regret bound, and the local regret bound.

**(1) Global regret upper bound**. In Algorithm 1, the model parameter is updated for $R$ iterations locally. Therefore, for any $\theta \in \Theta$,

$$\mathbb{E}\|\theta^{t+1} - \theta\|^2 = \mathbb{E}\left\|\sum_{k=1}^N \hat{p}_k \theta_k^{t,R} - \theta\right\|^2 \leq \sum_{k=1}^N \hat{p}_k \mathbb{E}\left\|\theta_k^{t,R} - \theta\right\|^2. \tag{10}$$

Denoting $g_k^{t,r} = \nabla L_k^t(f(x;\theta_k^{t,r}))$ and $G_k^{t,r} = \nabla L_k^t(f(x;\theta_k^{t,r}); \mathcal{B}_k^{t,r})$, the local update can be written as

$$\mathbb{E}\left\|\theta_k^{t,r+1} - \theta\right\|^2 = \mathbb{E}\left\|\theta_k^{t,r} - \eta G_k^{t,r} - \theta\right\|^2$$

$$= \mathbb{E}\left\|\theta_k^{t,r} - \theta\right\|^2 - 2\eta\mathbb{E}[\mathbb{E}[\langle G_k^{t,r}, \theta_k^{t,r} - \theta\rangle \mid \theta_k^{t,r}]] + \eta^2\mathbb{E}\|G_k^{t,r}\|^2$$

$$\leq \mathbb{E}\left\|\theta_k^{t,r} - \theta\right\|^2 - 2\eta\mathbb{E}[\langle g_k^{t,r}, \theta_k^{t,r} - \theta\rangle] + \eta^2(\mathbb{E}\|g_k^{t,r}\|^2 + \sigma^2).$$

Summarizing the inequality for $r = 0, \cdots, R-1$, it shows that

$$\mathbb{E}\left\|\theta_k^{t,R} - \theta\right\|^2 = \mathbb{E}\left\|\theta^t - \theta\right\|^2 - 2\eta\sum_{r=0}^{R-1}\mathbb{E}[\langle g_k^{t,r}, \theta_k^{t,r} - \theta\rangle] + \eta^2\sum_{r=0}^{R-1}(\mathbb{E}\|g_k^{t,r}\|^2 + \sigma^2). \tag{11}$$

By convexity, we have

$$\sum_{r=0}^{R-1}\langle g_k^{t,r}, \theta_k^{t,r} - \theta\rangle \geq \sum_{r=0}^{R-1} L_k^t(f(x;\theta_k^{t,r})) - L_k^t(f(x;\theta))$$

$$= \sum_{r=0}^{R-1} L_k^t(f(x;\theta_k^{t,r})) - L_k^t(f(x;\theta^t)) + L_k^t(f(x;\theta^t)) - L_k^t(f(x;\theta)) \tag{12}$$

By the $\beta$-smoothness, it indicates that $\|g_k^{t,r}\|^2 \leq 2\beta B_L$, and then

$$\mathbb{E}[L_k^t(f(x;\theta_k^{t,r+1}))] \geq \mathbb{E}[L_k^t(f(x;\theta_k^{t,r}))] - \eta\mathbb{E}[\langle g_k^{t,r}, \theta_k^{t,r+1} - \theta_k^{t,r}\rangle] - \frac{\beta}{2}\|\theta_k^{t,r+1} - \theta_k^{t,r}\|^2$$

$$= \mathbb{E}[L_k^t(f(x;\theta_k^{t,r}))] - \eta\mathbb{E}[\langle g_k^{t,r}, G_k^{t,r}\rangle] - \frac{\beta\eta^2}{2}\|G_k^{t,r}\|^2$$

$$= \mathbb{E}[L_k^t(f(x;\theta_k^{t,r}))] - \eta\mathbb{E}\|g_k^{t,r}\|^2 - \frac{\beta\eta^2}{2}(\|g_k^{t,r}\|^2 + \sigma^2)$$

$$\geq \mathbb{E}[L_k^t(f(x;\theta_k^{t,r}))] - \left(\eta + \frac{\beta\eta^2}{2}\right)2\beta B_L - \frac{\beta\eta^2}{2}\sigma^2$$

Summing up over $r = 0, \cdots, r'$, it presents that

$$\mathbb{E}[L_k^t(f(x; \theta_k^{t,r+1}))] - L_k^t(f(x; \theta)) \geq -(2 + \eta\beta)\beta\eta B_L r' - \frac{1}{2}\beta\eta^2\sigma^2 r'$$

Hence, summing up over $r' = 0, \cdots, R-1$ again, we have

$$\sum_{r=0}^{R-1} \mathbb{E}[L_k^t(f(x; \theta_k^{t,r}))] - L_k^t(f(x; \theta^t)) \geq -((2 + \eta\beta)B_L - \frac{1}{2}\eta\sigma^2)\frac{\beta\eta R(R-1)}{2} \qquad (13)$$

Combining (11), (12) and (13), and let $\eta \leq \frac{1}{\beta R}$, we obtain

$$\mathbb{E}\left\|\theta_k^{t,R} - \theta\right\|^2 \leq \mathbb{E}\left\|\theta^t - \theta\right\|^2 - 2\eta R[L_k^t(f(x; \theta^t)) - L_k^t(f(x; \theta))]$$
$$+ 2\eta^2 R^2 \beta B_L + 3\eta^2 R\beta B_L + \frac{3}{2}\eta^2 R\sigma^2. \qquad (14)$$

From (10), we know that

$$\mathbb{E}\left\|\theta^{t+1} - \theta\right\|^2 \leq \mathbb{E}\left\|\theta^t - \theta\right\|^2 - 2\eta R\sum_{k=1}^{N} \hat{p}_{a,k}[L_k^t(f(x; \theta^t)) - L_k^t(f(x; \theta))]$$
$$+ 2\eta^2 R^2 \beta B_L + 3\eta^2 R\beta B_L + \frac{3}{2}\eta^2 R\sigma^2$$

Summing over time and dividing both sides by $\frac{1}{2\eta RT}$, we obtain

$$\frac{1}{T}\sum_{t=1}^{T}\sum_{k=1}^{N} \hat{p}_{a,k}\mathbb{E}[L_k^t(f(x; \theta^t)) - L_k^t(f(x; \theta^*))]$$
$$\leq \frac{\mathbb{E}\left\|\theta^0 - \theta\right\|^2 - \mathbb{E}\left\|\theta^{T+1} - \theta\right\|^2}{2\eta RT} + \eta(R\beta B_L + \frac{3}{2}\beta B_L + \frac{3}{4}\sigma^2).$$

Plugging in $\theta = \theta^*$ and $\theta^0 = 0$ and considering the fact that $\theta^{T+1} - \theta \geq 0$, the result turns to

$$\frac{1}{T}\sum_{t=1}^{T}\sum_{k=1}^{N} \hat{p}_{a,k}\mathbb{E}[L_k^t(f(x; \theta^t)) - L_k^t(f(x; \theta^*))] \leq \frac{\|\theta^*\|^2}{2\eta RT} + \eta(R\beta B_L + \frac{3}{2}\beta B_L + \frac{3}{4}\sigma^2). \qquad (15)$$

Consider the update rule of ensemble weight $w_k^t$ in Algorithm 1,

$$w_k^{t+1} = \frac{1}{1 + \mathcal{W}_k^t(w_k^t)} = \frac{w_k^t \exp(-\eta_w L_k^t(\theta_k))}{w_k^t \exp(-\eta_w L_k^t(\theta^t)) + (1 - w_k^t)\exp(-\eta_w L_k^t(\phi_k^t))}.$$

Here, the update can be viewed as exponentiated gradient descent on the normalized weight vector $\mathbf{w_k^t} = (w_{k,1}^t, w_{k,2}^t) \in \Delta_2$, and $w_{k,i}^t \propto \exp(-\eta_w z_{t,i}^k), i = 1, 2$, where $z_{t,1}^k = L_k^t(f(x; \theta^t)), z_{t,2}^k = L_k^t(f(x; \phi_k^t))$. A well-known regret bound in online learning [70, 71] shows that, for any $\mathbf{u} = (u_1, u_2) \in \Delta_2$,

$$\sum_{t=1}^{T} w_k^t \mathbb{E}[L_k^t(f(x; \theta^t))] + (1 - w_k^t)\mathbb{E}[L_k^t(f(x; \phi_k^t))] - \sum_{t=1}^{T}\left(u_1\mathbb{E}[L_k^t(f(x; \theta^t))] + u_2\mathbb{E}[L_k^t(f(x; \phi_k^t))]\right)$$
$$\leq \frac{\log(2)}{\eta_w} + \eta_w TB_L. \qquad (16)$$

By the convexity of $L_k^t$, we have $\sum_{t=1}^{T} L_k^t(f_{k,ens}^t) \leq \sum_{t=1}^{T} w_k^t L_k^t(f(x; \theta^t)) + (1 - w_k^t)L_k^t(f(x; \phi_k^t))$. Plugging in $u_1 = 1, u_2 = 0$, it presents that

$$\sum_{t=1}^{T} \mathbb{E}[L_k^t(f_{k,ens}^t) - L_k^t(f(x; \theta^t))] \leq \frac{\log(2)}{\eta_w} + \eta_w TB_L. \qquad (17)$$

Weighted summing (17) over all clients and dividing both sides by $T$, we obtain

$$\frac{1}{T}\sum_{t=1}^{T}\sum_{k=1}^{N}\hat{p}_k\mathbb{E}[L_k^t(f_{k,ens}^t) - L_k^t(f(x;\theta^t))] \leq \frac{\log(2)}{\eta_w T} + \eta_w B_L. \tag{18}$$

Combining (18) and (15), and denoting $B_k = R\beta B_L + \frac{3}{2}\beta B_L + \frac{3}{4}\sigma^2$, we obtain

$$\frac{1}{T}\sum_{t=1}^{T}\sum_{k=1}^{N}\hat{p}_k\mathbb{E}[L_k^t(f_{k,ens}^t) - L_k^t(f(x;\theta^*))] \leq \frac{\|\theta^*\|^2}{2\eta RT} + \eta B_k + \frac{\log(2)}{\eta_w T} + \eta_w B_L. \tag{19}$$

Thus, we finish the proof of the global regret upper bound.

**(2) Local regret upper bound.** Plugging in $u_1 = 0, u_2 = 1$ in (16), it presents that

$$\sum_{t=1}^{T}\mathbb{E}[L_k^t(f_{k,ens}^t) - L_k^t(f(x;\phi_k^t))] \leq \frac{\log(2)}{\eta_w} + \eta_w T B_L \tag{20}$$

Following the proof technique of global regret upper bound, from (14), since $\phi_k^{t,R} = \phi_k^{t+1}$, and making $\eta \leq \frac{1}{\beta R}$, we have for any $\phi_k$,

$$\mathbb{E}\left\|\phi_k^{t+1} - \phi_k\right\|^2 \leq \mathbb{E}\left\|\phi_k^t - \phi_k\right\|^2 - 2\eta R[L_k^t(f(x;\phi_k^t)) - L_k^t(f(x;\phi_k))]$$
$$+ 2\eta^2 R^2\beta B_L + 3\eta^2 R\beta B_L + \frac{3}{2}\eta^2 R\sigma^2. \tag{21}$$

Combining (20) and (21), and plugging in $\phi_k = \phi_k^*$ and $\phi_k^0 = 0$ denoting $B_k = R\beta B_L + \frac{3}{2}\beta B_L + \frac{3}{4}\sigma^2$, the result turns to

$$\frac{1}{T}\sum_{t=1}^{T}\mathbb{E}[L_k^t(f_{k,ens}^t) - L_k^t(f(x;\phi_k^*))] \leq \frac{\|\phi_k^*\|^2}{2\eta RT} + \eta B_k + \frac{\log(2)}{\eta_w T} + \eta_w B_L. \tag{22}$$

Thus, we finish the proof of the local regret upper bound. $\qquad\square$

### B.4.2 Proof of Theorem 8

The proof of Theorem 8 relies on Lemma B.2.

**Lemma B.2.** *[70] Let $f^1, f^2, \ldots : \Lambda \rightarrow \mathbb{R}$ be a sequence of convex functions that we wish to minimize on a compact convex set $\Lambda$. Define the bound of the convex set $B_d \geq \max_{\lambda \in \Lambda}\|\Lambda\|_2$, and $B_G \geq \|\nabla f^t(\lambda^t)\|_2$ is a uniform upper bound on the norms of the subgradients. Suppose that we perform $T$ iterations of the following update, starting from $\lambda^{(1)} = \operatorname{argmin}_{\lambda \in \Lambda}\|\lambda\|_1$:*

$$\Lambda^t = \Pi_\Lambda\left(\lambda^{(t)} - \eta\nabla f^t(\lambda^t)\right)$$

*where $\nabla f^t(\lambda^t) \in \partial f^t(\lambda^t)$ is a subgradient of $f^t$ at $(\lambda$, and $\Pi_\Lambda$ projects its argument onto $\Lambda$ w.r.t. the Euclidean norm. Then:*

$$\frac{1}{T}\sum_{t=1}^{T}f^t(\lambda^t) - \frac{1}{T}\sum_{t=1}^{T}f^t(\lambda^*) \leq \frac{B_d^2}{2\eta} + \eta T B_G^2$$

*where $\lambda^* \in \Lambda$ is an arbitrary reference vector.*

*Proof of Theorem 8.* Consider the empirical form of the Lagrangian function $\widehat{\mathcal{L}}(\mathbf{h}, \lambda, \mu)$,

$$\widehat{\mathcal{L}}(\mathbf{h}, \lambda, \mu) = \widehat{\mathcal{R}}(\mathbf{h}) + (\lambda^{(1)} - \lambda^{(2)})^\top(\widehat{\mathscr{D}}^g(\mathbf{h}) - \xi^g) + \sum_{k=1}^{N}(\mu^{1,k} - \mu^{2,k})^\top(\widehat{\mathscr{D}}^k(\mathbf{h}) - \xi^k),$$

$$= \sum_{k=1}^{N}\hat{p}_k\frac{1}{n_k}\sum_{i=1}^{n_k}e_{y_{k,i}}^\top\left[\mathbf{1} - \widehat{\mathbf{M}}^{\lambda,\mu}(a_{k,i}, k)\right]h_k(x_{k,i}).$$

where $\widehat{\mathbf{M}}^{\lambda,\mu}(a,k) := \mathbf{I} - \frac{1}{\hat{p}_{a,k}}\left[\sum_{u_g \in \mathcal{U}_g}(\lambda_{u_g}^{(1)} - \lambda_{u_g}^{(2)})\widehat{\mathbf{D}}_{u_g}^{a,k} - \sum_{u_k \in \mathcal{U}_k}(\mu_{k,u_k}^{(1)} - \mu_{k,u_k}^{(2)})\widehat{\mathbf{D}}_{u_k}^{a,k}\right]$. It is clear that the inner problem is linear to classifiers in the empirical case.

From the definition in Section 4, we have $\|\lambda\|_1 \le B_d, \|\mu\|_1 \le B_d$. Since the norm of fairness metrics is less than 2, setting the step size $\eta_d = B_d/\sqrt{T}$, by Lemma B.2,

$$\max_{\lambda^*,\mu^*} \frac{1}{T}\sum_{t=1}^{T}\widehat{\mathcal{L}}\left(\mathbf{h}^t, \lambda^*, \mu^*\right) - \frac{1}{T}\sum_{t=1}^{T}\widehat{\mathcal{L}}\left(\mathbf{h}^t, \lambda^t, \mu^t\right) \le \frac{2B_d^2}{\eta_d} + 4\eta_d T = 16B_d^2\sqrt{\frac{1}{T}}, \qquad (23)$$

where $\lambda^*, \mu^*$ are the optimal dual parameters satisfying $\|\lambda\|_1 \le B_d, \|\mu\|_1 \le B_d$.

On the other hand, according to the sub-optimal assumption on the classifier $\mathbf{h}$, we have

$$\frac{1}{T}\sum_{t=1}^{T}\widehat{\mathcal{L}}\left(\mathbf{h}^t, \lambda^t, \mu^t\right) - \inf_{\mathbf{h}^*}\frac{1}{T}\sum_{t=1}^{T}\widehat{\mathcal{L}}\left(\mathbf{h}^*, \lambda^t, \mu^t\right) \le \frac{1}{T}\sum_{t=1}^{T}\widehat{\mathcal{L}}\left(\mathbf{h}^t, \lambda^t, \mu^t\right) - \frac{1}{T}\sum_{t=1}^{T}\inf_{\mathbf{h}^*}\widehat{\mathcal{L}}\left(\mathbf{h}^*, \lambda^t, \mu^t\right) \le \bar{\rho},$$

$$(24)$$

where $\bar{\rho} := \sum_{t=1}^{T}\rho_t/T$. Combining (23) and (24), the result shows that

$$\max_{\lambda^*,\mu^*} \frac{1}{T}\sum_{t=1}^{T}\widehat{\mathcal{L}}\left(\mathbf{h}^t, \lambda^*, \mu^*\right) - \inf_{\mathbf{h}^*}\frac{1}{T}\sum_{t=1}^{T}\widehat{\mathcal{L}}\left(\mathbf{h}^*, \lambda^t, \mu^t\right) \le \bar{\rho} + 16B_d^2\sqrt{\frac{1}{T}}. \qquad (25)$$

Let $\overline{\mathbf{h}} := \frac{1}{T}\sum_{t=1}^{T}\mathbf{h}^t$ with $\overline{h}_k := \frac{1}{T}\sum_{t=1}^{T}h_k^t, k \in [N]$, and let $\bar{\lambda} := \frac{1}{T}\sum_{t=1}^{T}\lambda^t, \bar{\mu} := \frac{1}{T}\sum_{t=1}^{T}\mu^t$ denote the point-wise average of dual parameters. Therefore, due to the linearity of the empirical Lagrange function to classifiers and dual parameters, (25) can be formulated as

$$\max_{\lambda^*,\mu^*} \widehat{\mathcal{L}}\left(\overline{\mathbf{h}}, \lambda^*, \mu^*\right) - \inf_{\mathbf{h}^*}\widehat{\mathcal{L}}\left(\mathbf{h}^*, \bar{\lambda}, \bar{\mu}\right) \le \bar{\rho} + 16B_d^2\sqrt{\frac{1}{T}}, \qquad (26)$$

which presents the approximate mixed Nash equilibrium of the stochastic saddle-point problem. $\square$

### B.5 Generalization Error For In-processing Algorithm

We begin by introducing some notations and simplifications, which are commonly employed in generalization analyses of FL [39, 68]. Without loss of generalization, let $n = n_1 = \cdots = n_k$ present the sample number in local datasets. For any class $\mathcal{H} = \{h : \mathcal{X} \to [m]\}$, denote $\mathcal{H}_y = \{\mathbb{I}\{h(x) = y\} : h \in \mathcal{H}\}$ and the maximal Vapnik-Chervonenkis dimension [69], $VC(\mathcal{H}) := \max_{y \in [m]} VC(\mathcal{H}_y)$.

**Theorem 9.** *If classifiers* $\overline{\mathbf{h}} = (\overline{h}_1, \ldots, \overline{h}_k)$ *with dual parameters* $(\bar{\lambda}, \bar{\mu})$ *form a* $\epsilon$*-saddle point of empirical Lagrangian* $\widehat{\mathcal{L}}(\mathbf{h}, \lambda, \mu)$*, and an optimal solution* $\mathbf{h}^* \in \mathcal{H}$ *satisfies both global and local fairness constraints, denoting* $\nu(n, \mathcal{H}, \delta) = 2\sqrt{\frac{2VC(\mathcal{H})\log(n+1)}{n}} + \sqrt{\frac{2\log(m^2 N/\delta))}{n}}$, $B_g = \max_{a \in \mathcal{A}, k \in [N]} \|\mathbf{D}_{u_g}^{a,k}\|_1, \Omega_n^g = \max_{a \in \mathcal{A}, k \in [N]} \|\mathbf{D}_{u_g}^{a,k} - \widehat{\mathbf{D}}_{u_g}^{a,k}\|_\infty, \Omega_n^p := \sum_{k=1}^{N}|p_k - \hat{p}_k|$, *and* $B_k = \max_{a \in \mathcal{A}} \|\mathbf{D}_{u_k}^{a,k}\|_1, \Omega_n^k = \max_{a \in \mathcal{A}} \|\mathbf{D}_{u_k}^{a,k} - \widehat{\mathbf{D}}_{u_k}^{a,k}\|_\infty, k \in [N]$, *then with probability at least* $1 - \delta$,

$$|\mathscr{D}^g(\overline{\mathbf{h}})| \le \xi^g + \nu(n, \mathcal{H}, \delta/|\mathcal{A}||\mathcal{U}_g|)|\mathcal{A}|NB_g + \Omega_n^g + \frac{1 + 2\epsilon}{B_d},$$

$$|\mathscr{D}^k(\overline{\mathbf{h}})| \le \xi^k + \nu(n, \mathcal{H}, N\delta/|\mathcal{A}||\mathcal{U}_k|)|\mathcal{A}|B_k + \Omega_n^k + \frac{1 + 2\epsilon}{B_d}$$

$$\mathcal{R}(\overline{\mathbf{h}}) \le \mathcal{R}(\mathbf{h}^*) + 2m\Omega_n^p + 2m\nu(n, \mathcal{H}, \delta/2) + 2\epsilon.$$

The proof of Theorem 9 relies on the following lemma.

**Lemma B.3.** *Let* $\mathcal{H} : \mathcal{X} \to [m], \mathcal{D}$ *a distribution over* $\mathcal{X} \times \Delta_m$*, of which* $\{x_i, y_i\}_{i=1}^{n}$ *are i.i.d samples. Denoting* $\mathcal{H}_y = \{\mathbb{I}\{h(x) = y\} : h \in \mathcal{H}\}$ *and* $VC(\mathcal{H}) = \max_{y \in [m]} VC(\mathcal{H}_y)$*, then with probability at least* $1 - \delta$*, for* $\forall i, j \in [m]$,

$$\sup_{h \in \mathcal{F}_{\mathcal{H}}} \left|\mathbf{C}_{i,j}(h) - \widehat{\mathbf{C}}_{i,j}(h)\right| \le 2\sqrt{\frac{2VC(\mathcal{H})\log(n+1)}{n}} + \sqrt{\frac{2\log\frac{1}{\delta}}{n}}.$$

*where* $\mathcal{F}_{\mathcal{H}} := \{f(x) = \sum_{j=1}^{N}\alpha_j h_j(x) : \alpha \in \Delta_N, h_j \in \mathcal{H}, j \in [m]\}$.

*Proof of Lemma B.3.* Let $\ell_{i,j}(x, y; h) = \mathbb{I}(y = i \wedge h(x) = j)$. Then we have $\mathbf{C}_{i,j}(h) = \mathbb{E}[\ell_{i,j}(x, y; h)]$ and $\widehat{\mathbf{C}}_{i,j}(h) = \frac{1}{n}\sum_{i=1}^{n}\ell_{i,j}(x_i, y_i; h)$. Hence, according to the classical result with respect to cost sensitive binary classification [8], with probability at least $1 - \delta$,

$$\sup_{h \in \mathcal{F}_\mathcal{H}} \left|\mathbf{C}_{i,j}(h) - \widehat{\mathbf{C}}_{i,j}(h)\right| \leq 2\sqrt{\frac{2VC(\mathcal{H}_j)\log(n+1)}{n}} + \sqrt{\frac{2\log\frac{1}{\delta}}{n}}.$$

By the definition of $VC(\mathcal{H})$, it achieves the generalization bound.

### B.5.1 Proof of Theorem 9

Let the optimal solution $\mathbf{h}^*$ minimize the risk $\mathcal{R}(\mathbf{h})$ subjected to global and local fairness constraints $|\mathscr{D}^g(\mathbf{h}^*)| \leq \xi^g$, $|\mathscr{D}^{k,l}(\mathbf{h}^*)| \leq \xi^{k,l}$. With the properties of the saddle point, it is clear that

$$\widehat{\mathcal{L}}\left(\bar{\mathbf{h}}, \bar{\lambda}, \bar{\mu}\right) \leq \widehat{\mathcal{L}}\left(\mathbf{h}, \bar{\lambda}, \bar{\mu}\right) + \epsilon, \qquad \forall\, \mathbf{h} \in \mathcal{H}, \tag{27}$$

$$\widehat{\mathcal{L}}\left(\bar{\mathbf{h}}, \bar{\lambda}, \bar{\mu}\right) \geq \widehat{\mathcal{L}}\left(\bar{\mathbf{h}}, \lambda, \mu\right) - \epsilon, \qquad \forall\, \|\lambda\|_1, \|\mu\|_1 \leq B_d. \tag{28}$$

Considering the global fairness constraints, we first explore its concentration, for any $h \in \mathcal{H}$,

$$\mathscr{D}_{u_g}^g(\mathbf{h}) - \widehat{\mathscr{D}}_{u_g}^g(\mathbf{h}) = \sum_{k=1}^{N}\sum_{a\in\mathcal{A}}\langle\mathbf{D}_{u_g}^{a,k}, \mathbf{C}^{a,k}(h_k)\rangle - \langle\widehat{\mathbf{D}}_{u_g}^{a,k}, \widehat{\mathbf{C}}^{a,k}(h_k)\rangle$$

$$= \sum_{k=1}^{N}\sum_{a\in\mathcal{A}}\langle\mathbf{D}_{u_g}^{a,k}, \mathbf{C}^{a,k}(h_k) - \widehat{\mathbf{C}}^{a,k}(h_k)\rangle + \langle\mathbf{D}_{u_g}^{a,k} - \widehat{\mathbf{D}}_{u_g}^{a,k}, \widehat{\mathbf{C}}^{a,k}(h_k)\rangle$$

$$\leq \sum_{k=1}^{N}\sum_{a\in\mathcal{A}}\|\mathbf{D}_{u_g}^{a,k}\|_1\|\mathbf{C}^{a,k}(h_k) - \widehat{\mathbf{C}}^{a,k}(h_k)\|_\infty + \|\mathbf{D}_{u_g}^{a,k} - \widehat{\mathbf{D}}_{u_g}^{a,k}\|_\infty\|\widehat{\mathbf{C}}^{a,k}(h_k)\|_1.$$

The last inequality is by the Holder's inequality. Let $\ell_{i,j}^{a',k}(x, y, a; h) = \mathbb{I}(y = i \wedge h(x) = j \wedge a = a')$. Then we have $\mathbf{C}_{i,j}^{a',k}(h) = \mathbb{E}[\ell_{i,j}^{a',k}(x, y, a; h)]$ and $\widehat{\mathbf{C}}_{i,j}^{a'}(h) = \frac{1}{n}\sum_{i=1}^{n}\ell_{i,j}(x_i, y_i; h)$. By taking a union bound in Lemma B.3, we have that with probability at least $1 - \delta$,

$$\sup_{h \in \mathcal{F}_\mathcal{H}}\max_{k\in[N]}\max_{a\in\mathcal{A}}\left\|\mathbf{C}^{a,k}(h) - \widehat{\mathbf{C}}^{a,k}(h)\right\|_\infty \leq 2\sqrt{\frac{2VC(\mathcal{H})\log(n+1)}{n}} + \sqrt{\frac{2\log(m^2|\mathcal{A}|N/\delta)}{n}}.$$

Since $\|\widehat{\mathbf{C}}^{a,k}(h_k)\|_1 = 1$, taking the union bound again, denoting $\nu(n, \mathcal{H}, \delta) = 2\sqrt{\frac{2VC(\mathcal{H})\log(n+1)}{n}} + \sqrt{\frac{2\log(m^2N/\delta))}{n}}$, it turns out that with probability at least $1 - \delta$,

$$\mathscr{D}_{u_g}^g(\mathbf{h}) - \widehat{\mathscr{D}}_{u_g}^g(\mathbf{h}) \leq \sum_{k=1}^{N}\sum_{a\in\mathcal{A}}\nu(n, \mathcal{H}, \delta/|\mathcal{A}|)\|\mathbf{D}_{u_g}^{a,k}\|_1 + \|\mathbf{D}_{u_g}^{a,k} - \widehat{\mathbf{D}}_{u_g}^{a,k}\|_\infty \tag{29}$$

$$\leq \nu(n, \mathcal{H}, \delta/|\mathcal{A}|)|\mathcal{A}|NB_g + \Omega_n^g. \tag{30}$$

Next, we consider the optimality. Denoting $u_g^* := \arg\max_{u_g\in\mathcal{U}_g}|\widehat{\mathscr{D}}_{u_g}^g(\bar{\mathbf{h}})|$, then we have

$$B_d(\widehat{\mathscr{D}}_{u_g^*}^g(\bar{\mathbf{h}}) - \xi^g) = \widehat{\mathcal{L}}(\bar{\mathbf{h}}, Be_{u_g^*}^1, 0) - \widehat{R}(\bar{\mathbf{h}}) \leq \widehat{\mathcal{L}}(\bar{\mathbf{h}}, \bar{\lambda}, \bar{\mu}) - \widehat{R}(\bar{\mathbf{h}}) + \epsilon, \tag{31}$$

where $e_{u_g^*}^1$ defines as the basis vector with 1 at the position of $\lambda_{u_g^*}^1$. Let $\mathbf{h}$ satisfy the fairness constraints. With (27), we obtain

$$\widehat{\mathcal{L}}(\bar{\mathbf{h}}, \bar{\lambda}, \bar{\mu}) - \widehat{R}(\bar{\mathbf{h}}) \leq \widehat{\mathcal{L}}(\mathbf{h}, \bar{\lambda}, \bar{\mu}) - \widehat{R}(\bar{\mathbf{h}}) + \epsilon \leq \widehat{R}(\mathbf{h}) - \widehat{R}(\bar{\mathbf{h}}) + \epsilon. \tag{32}$$

Combining (31) and (32), it shows that

$$\widehat{\mathscr{D}}_{u_g^*}^g(\bar{\mathbf{h}}) - \xi^g \leq \frac{\widehat{R}(\mathbf{h}) - \widehat{R}(\bar{\mathbf{h}}) + 2\epsilon}{B_d} \leq \frac{1 + 2\epsilon}{B_d}. \tag{33}$$

Therefore, the result shows that $\max_{u_g \in \mathcal{U}_g} |\widehat{\mathscr{D}}^g_{u^*_g}(\overline{\mathbf{h}})| - \xi^g \le \frac{1+2\epsilon}{B_d}$.

Now we consider the generalization error for the empirically optimal classifier $\overline{\mathbf{h}}$, with probability at least $1 - \delta$,

$$|\mathscr{D}^g_{u_g}(\overline{\mathbf{h}})| - \xi^g \le |\mathscr{D}^g_{u_g}(\overline{\mathbf{h}}) - \widehat{\mathscr{D}}^g_{u_g}(\overline{\mathbf{h}})| + |\widehat{\mathscr{D}}^g_{u_g}(\overline{\mathbf{h}})| - \xi^g \tag{34}$$

$$\le \nu(n, \mathcal{H}, \delta/|\mathcal{A}|)|\mathcal{A}|NB_g + \Omega^g_n + \frac{1+2\epsilon}{B_d}. \tag{35}$$

Taking the union bound over $u_g \in \mathcal{U}_g$, we have that with probability at least $1 - \delta$,

$$|\mathscr{D}^g(\overline{\mathbf{h}})| - \xi^g \le \nu(n, \mathcal{H}, \delta/|\mathcal{A}||\mathcal{U}_g|)|\mathcal{A}|NB_g + \Omega^g_n + \frac{1+2\epsilon}{B_d}. \tag{36}$$

For local fairness constraints, $|\mathscr{D}_k| \le \xi^k$, following the similar proof procedures as local fairness constraints, we have that

$$|\mathscr{D}^k(\overline{\mathbf{h}})| - \xi^k \le \nu(n, \mathcal{H}, N\delta/|\mathcal{A}||\mathcal{U}_k|)|\mathcal{A}|B_k + \Omega^k_n + \frac{1+2\epsilon}{B_d}. \tag{37}$$

For risk metric $\mathcal{R}(\mathbf{h})$, it presents that

$$\mathcal{R}(\overline{\mathbf{h}}) - \mathcal{R}(\mathbf{h}^*) = \mathcal{R}(\overline{\mathbf{h}}) - \widehat{\mathcal{R}}(\overline{\mathbf{h}}) + \widehat{\mathcal{R}}(\overline{\mathbf{h}}) - \widehat{\mathcal{R}}(\mathbf{h}^*) + \widehat{\mathcal{R}}(\mathbf{h}^*) - \mathcal{R}(\mathbf{h}^*). \tag{38}$$

By (27) and (28),

$$\widehat{\mathcal{R}}(\overline{\mathbf{h}}) - \widehat{\mathcal{R}}(\mathbf{h}^*) \le \widehat{\mathcal{L}}(\overline{\mathbf{h}}, \mathbf{0}, \mathbf{0}) - \widehat{\mathcal{L}}(\mathbf{h}^*, \bar{\lambda}, \bar{\mu}) \le \widehat{\mathcal{L}}\left(\overline{\mathbf{h}}, \bar{\lambda}, \bar{\mu}\right) + \epsilon - \widehat{\mathcal{L}}\left(\overline{\mathbf{h}}, \bar{\lambda}, \bar{\mu}\right) + \epsilon = 2\epsilon. \tag{39}$$

Since we have $\widehat{\mathcal{R}}(\mathbf{h}) = 1 - \sum_{k=1}^N \hat{p}_k \langle \mathbf{I}, \mathbf{C}^k(h_k) \rangle$, it presents that

$$\begin{aligned}
\widehat{\mathcal{R}}(h) - \mathcal{R}(h) &= \sum_{k=1}^N \langle \mathbf{I}, p_k \mathbf{C}^k(h_k) - \hat{p}_k \widehat{\mathbf{C}}^k(h_k) \rangle \\
&= \sum_{k=1}^N (p_k - \hat{p}_k)\langle \mathbf{I}, \mathbf{C}^k(h_k) \rangle + \sum_{k=1}^N p_k \langle \mathbf{I}, \mathbf{C}^k(h_k) - \widehat{\mathbf{C}}^k(h_k) \rangle \\
&\le m \sum_{k=1}^N |p_k - \hat{p}_k| + \sum_{k=1}^N p_k m \|\mathbf{C}^k(h_k) - \widehat{\mathbf{C}}^k(h_k)\|_\infty
\end{aligned}$$

By taking a union bound in Lemma B.3, we have that with probability at least $1 - \delta$,

$$\sup_{h \in \mathcal{F}_\mathcal{H}} \max_{k \in [N]} \left\| \mathbf{C}^k(h) - \widehat{\mathbf{C}}^k(h) \right\|_\infty \le 2\sqrt{\frac{2VC(\mathcal{H})\log(n+1)}{n}} + \sqrt{\frac{2\log(m^2 N/\delta)}{n}}.$$

Hence, denoting $\Omega^p_n := \sum_{k=1}^N |p_k - \hat{p}_k|$, we arrive that, for any $h \in \mathcal{F}_\mathcal{H}$,

$$\begin{aligned}
\widehat{\mathcal{R}}(h) - \mathcal{R}(h) &\le N \max_{k \in [N]} |p_k - \hat{p}_k| + \sum_{k=1}^N p_k m \|\mathbf{C}^k(h_k) - \widehat{\mathbf{C}}^k(h_k)\|_\infty \\
&\le m\Omega^p_n + m\nu(n, \mathcal{H}, \delta). \tag{40}
\end{aligned}$$

Therefore, combining (38), (39) and (40), we obtain

$$\mathcal{R}(\overline{\mathbf{h}}) - \mathcal{R}(\mathbf{h}^*) \le 2m\Omega^p_n + 2m\nu(n, \mathcal{H}, \delta/2) + 2\epsilon. \tag{41}$$

This completes the proof. $\qquad\square$

## B.6  Proof of Theorem 5

We begin by introducing some definitions and lemmas, which are useful in the proof of Theorem 5.

**Definition 3.** Let $V$ be a real vector space and let $A, B \subseteq V$. The sum of $A$ and $B$ is defined by

$$A + B := \{ a + b \mid a \in A, \ b \in B \}.$$

**Lemma B.4.** *[6] The subdifferential of the function $F(x) = \mathbb{E}\{f(x, \omega)\}$ at a point $x$ is given by*

$$\partial F(x) = \mathbb{E}\{\partial f(x, \omega)\}$$

*where $f(\cdot, \omega)$ is a real-value convex function and the set $\mathbb{E}\{\partial f(x, \omega)\}$ is defined as*

$$\mathbb{E}\{\partial f(x, \omega)\} := \int_\Omega \partial f(x, \omega) d\mathbb{P}(\omega)$$

$$= \left\{ x^* \in \mathbb{R}^n \mid x^* = \int_\Omega x^*(\omega) d\mathbb{P}(\omega), x^*(\cdot) : measurable, x^*(\omega) \in \partial f(x, \omega) \ a.e. \right\}.$$

**Lemma B.5.** *[66] Let $f_1, \ldots, f_m : \mathbb{R}^n \to (-\infty, +\infty]$ be convex functions. Define $f(x) = \max\{f_1(x), \ldots, f_m(x)\}, \quad \forall x \in \mathbb{R}^n$. For $x_0 \in \bigcap_{i=1}^m \text{dom} f_i$, define $I(x_0) = \{i \mid f_i(x_0) = f(x_0)\}$. Then $\partial f(x_0) = \text{conv} \bigcup_{i \in I(x_0)} \partial f_i(x_0)$.*

**Lemma B.6.** *[66] Let $f : \mathbb{R}^d \to \mathbb{R}$ be a convex continuous function. We consider the minimizer $x^*$ of the function $f$ over the set $B$. Then for $x^*$ to be locally optimal it is necessary that*

$$\partial f(x^*) + \mathcal{N}_B(x^*) \ni 0,$$

*where $\mathcal{N}_B$ denotes the normal cone of set $B$. If $B = \mathbb{R}_+^d$, let $\mathcal{K} := \{k \in [d], x_k^* \neq 0\}$. Then there exists a subgradient $\xi \in \partial f(x^*)$, such that for all $k \in [d]$ we have $\xi_k \geq 0$ and $\forall k \in \mathcal{K}, \xi_k = 0$.*

*Proof of Theorem 5.* From the above analysis, it follows that the Lagrange function can be written as

$$\mathcal{L}(\mathbf{h}, \lambda, \mu) = 1 - \sum_{k=1}^N \sum_{a \in \mathcal{A}} p_{a,k} \left\langle \mathbf{M}^{\lambda,\mu}(a, k), \mathbf{C}^{a,k}(h_k) \right\rangle - \sum_{u_g \in \mathcal{U}_g} (\lambda_{u_g}^{(1)} + \lambda_{u_g}^{(2)}) \xi^g$$

$$- \sum_{k \in [N]} \sum_{u_k \in \mathcal{U}_k} (\mu_{k,u_k}^{(1)} + \mu_{k,u_k}^{(2)}) \xi^k.$$

We first consider the inner optimization problem $\min_{\mathbf{h} \in \mathcal{H}^N} \mathcal{L}(\mathbf{h}, \lambda, \mu)$, which is equivalent to optimize

$$\max_{\mathbf{h} \in \mathcal{H}^N} V(\mathbf{h}, \lambda, \mu) = \sum_{k=1}^N \sum_{a \in \mathcal{A}} p_{a,k} \left\langle \mathbf{M}^{\lambda,\mu}(a, k), \mathbf{C}^{a,k}(h_k) \right\rangle.$$

where $\mathbf{M}^{\lambda,\mu}(a, k) := \mathbf{I} - \frac{1}{p_{a,k}} \left[ \sum_{u_g \in \mathcal{U}_g} (\lambda_{u_g}^{(1)} - \lambda_{u_g}^{(2)}) \mathbf{D}_{u_g}^{a,k} - \sum_{u_k \in \mathcal{U}_k} (\mu_{k,u_k}^{(1)} - \mu_{k,u_k}^{(2)}) \mathbf{D}_{u_k}^{a,k} \right]$. Considering the personalized attribute-aware classifier $h_k(x, a), k \in [N]$ in post-processing, the inner function turns to

$$V(\mathbf{h}, \lambda, \mu) = \sum_{k=1}^N \sum_{a \in \mathcal{A}} p_{a,k} \left\langle \mathbf{M}^{\lambda,\mu}(a, k), \mathbf{C}^{a,k}(h_k) \right\rangle$$

$$= \sum_{k=1}^N \sum_{a \in \mathcal{A}} p_{a,k} \int_{\mathcal{X}} [\eta(x, a, k)]^\top \mathbf{M}^{\lambda,\mu}(a, k) h_k(x, a) d\mathcal{P}_{a,k}^X$$

An explicit optimal solution of personalized classifier is that

$$h_k^{\lambda,\mu}(x, a) := \arg\max_{y \in [m]} \left( \left[ \mathbf{M}^{\lambda,\mu}(a, k) \right]^\top \eta(x, a, k) \right)_j.$$

If the maximum entry of the output vector occurs at multiple indices, one of them is randomly selected as the predicted class. Thus, the dual problem can be formulated as

$$\min_{\lambda,\mu} H(\lambda, \mu) := \sum_{k \in [N]} \sum_{a \in \mathcal{A}} p_{a,k} \mathbb{E}_{X \sim \mathcal{P}_{a,k}^X} \left[ \max_{y \in [m]} \left( \left[ \mathbf{M}^{\lambda,\mu}(a, k) \right]^\top \eta(X, a, k) \right)_y \right] + \xi^g \sum_{u_g \in \mathcal{U}_g} (\lambda_{u_g}^{(1)} + \lambda_{u_g}^{(2)})$$

$$+ \sum_{k \in [N]} \xi^k \sum_{u_k \in \mathcal{U}_k} (\mu_{k,u_k}^{(1)} + \mu_{k,u_k}^{(2)}).$$

(42)

Before exploring the optimal solution of outer optimization, we first prove that the optimal dual parameter $\lambda^* \in \mathbb{R}_{\geq 0}^{2|\mathcal{U}_g|}, \mu^* \in \mathbb{R}_{\geq 0}^{2\sum_{k=1}^{N}|\mathcal{U}_k|}$ is bounded. Define the Hilbert space on $\mathcal{F} := \{f : \mathcal{X} \to \mathbb{R}^m\}$ with inner product $\langle f, g \rangle = \int_{\mathcal{X}} f^\top g d\mathcal{P}(x)$. Then the classifier space $\mathcal{H} : \mathcal{X} \to \Delta_m$ is a convex subset of $\mathcal{F}$. Therefore, we can also consider the topology structure on $\mathcal{H}$ or $\mathcal{H}^{|\mathcal{A}|}$. Since we assume that $\forall \xi^g, \xi^k > 0$, the feasible set of the primal problem is non-empty, it indicates that the feasible set of the primal problem has non-empty interior for any positive $\xi^g, \xi^k$. It is clear that for $\forall \xi^g, \xi^k > 0$, the dual problem

$$\min_{\mathbf{h}} \mathcal{L}(\mathbf{h}, \lambda, \mu) = 1 - H(\lambda, \mu) \leq \mathcal{R}(\mathbf{h}) \leq \mathcal{R}_{\max}(\mathbf{h}^{fair})$$

where $\mathbf{h}^{fair}$ denotes a classifier that satisfies fairness constraints for given $\xi^g, \xi^k > 0$. Hence, we arrive at

$$H(\lambda, \mu) \geq 1 - \mathcal{R}_{\max}(\mathbf{h}^{fair}) > 0 \tag{43}$$

holds for all $\lambda, \mu \geq 0$. Notice that given $\lambda, \mu \geq 0$, this inequality holds for any $\xi^g, \xi^k > 0$. Let $\xi^g \to 0, \xi^k \to 0$, combining (42) and (43) gives that

$$\sum_{k \in [N]} \sum_{a \in \mathcal{A}} p_{a,k} \mathbb{E}_{X \sim \mathcal{P}_{a,k}^X} \left[ \max_{y \in [m]} \left( [\mathbf{M}^{\lambda,\mu}(a,k)]^\top \eta(X,a,k) \right)_y \right] > 0 \tag{44}$$

Therefore, the dual problem has a lower bound

$$H(\lambda, \mu) \geq \xi^g \sum_{u_g \in \mathcal{U}_g} (\lambda_{u_g}^{(1)} + \lambda_{u_g}^{(2)}) + \sum_{k \in [N]} \xi^k \sum_{u_k \in \mathcal{U}_k} (\mu_{k,u_k}^{(1)} + \mu_{k,u_k}^{(2)}) \tag{45}$$

It presents that, as $\|\lambda\|_1 \to \infty$ or $\|\mu\|_1 \to \infty$, there must be $H(\lambda, \mu) \to \infty$, which conflicts with the dual problem $\min_{\lambda,\mu} H(\lambda, \mu)$. Hence, the optimal $\lambda^*, \mu^*$ of dual problem $\min_{\lambda,\mu} H(\lambda, \mu)$ must have bounded norms, denoting as $\|\lambda\|_1 \leq B_d, \|\mu\|_1 \leq B_d$.

Now we consider the differential of $H(\lambda, \mu)$. It is clear that $\{S_y = \{x \in \mathcal{X} : h_k(x, a) = y\}, y \in [m]\}$ constructs a partition of the feature space $\mathcal{X}$. Hence, for dual parameter $\lambda_{u_g}^{(1)}$, since the outer objective $H$ is convex to $\lambda$ and $\mu$, by the additivity subgradients and Lemma B.4, the differential $\frac{\partial}{\partial \lambda_{u_g}^{(1)}} H(\lambda, \mu)$ can be formulated as

$$\frac{\partial}{\partial \lambda_{u_g}^{(1)}} H(\lambda, \mu) = \sum_{k \in [N]} \sum_{a \in \mathcal{A}} p_{a,k} \mathbb{E}_{X \sim \mathcal{P}_{a,k}^X} \left[ \frac{\partial}{\partial \lambda_{u_g}^{(1)}} \max_{y \in [m]} \left( [\mathbf{M}^{\lambda,\mu}(a,k)]^\top \eta(X,a,k) \right)_y \right] + \xi^g. \tag{46}$$

With a slight abuse of notation, let score function $f(x, a, k) = [\mathbf{M}^{\lambda,\mu}(a,k)]^\top \eta(x,a,k)$, by Lemma B.5, we have

$$\mathbb{E}_{X \sim \mathcal{P}_{a,k}^X} \left[ \frac{\partial}{\partial \lambda_{u_g}^{(1)}} \max_{y \in [m]} \left( [\mathbf{M}^{\lambda,\mu}(a,k)]^\top \eta(X,a,k) \right)_y \right] \tag{47}$$

$$= \sum_{y \in [m]} \int_{\{x : h_k^{\lambda,\mu}(x,a) = y\}} \left[ \frac{\partial}{\partial \lambda_{u_g}^{(1)}} \max_{y \in [m]} \left( [\mathbf{M}^{\lambda,\mu}(a,k)]^\top \eta(x,a,k) \right)_y \right] d\mathcal{P}_{a,k}^X(x) \tag{48}$$

$$= \frac{1}{p_{a,k}} \sum_{y \in [m]} \int_{\{x : h_k^{\lambda,\mu}(x,a) = y\}} \left[ \text{conv} \left( \bigcup_{i \in \arg\max_i(f_i(x,a,k))} -[\eta(x,a,k)]^T \mathbf{D}_{u_g}^{a,k} e_y \right) \right] d\mathcal{P}_{a,k}^X(x) \tag{49}$$

$$= \frac{1}{p_{a,k}} \sum_{y \in [m]} \left\{ \int_{\{x : f_y(x,a,k) \geq f_i(x,a,k), \forall i \neq y, i \in [m]\}} \left[ -[\eta(x,a,k)]^\top \mathbf{D}_{u_g}^{a,k} e_y \right] d\mathcal{P}_{a,k}^X(x) \right.$$

$$\left. + \int_{B_y^t} \left[ -b_t [\eta(x,a,k)]^\top \mathbf{D}_{u_g}^{a,k} (e_t - e_y) \right] d\mathcal{P}_{a,k}^X(x) \right\}, \tag{50}$$

where $B_y^t := \{\exists t \neq y, f_t(x,a,k) \geq f_i(x,a,k), \forall i \in [m]; f_t(x,a,k) = f_y(x,a,k)\}$ with $b_t \in [0,1]$. Since the convex hull is a interval here, by Caratheodory's theorem, it can be characterized by

two point here (the initial point $e_y$ and another point $e_t$ in the convex hull). Without loss of generality, we assume the existence of one $e_t$ such that $f_t(x, a, k) = f_y(x, a, k)$ here. We know that $f_t(x, a, k) - f_y(x, a, k) = [\eta(x, a, k)]^\top \mathbf{M}^{\lambda, \mu}(a, k)(e^t - e^y)$. With Assumption 1, we obtain that the measure of $B_y^t$ is 0, unless the $t$-th and $y$-th column of $\mathbf{M}^{\lambda, \mu}(a, k)$ are equal. An effective simplification is to exclude all $\lambda', \mu'$ that cause $\mathbf{M}^{\lambda', \mu'}(a, k)(e^t - e^y) = 0$. Since we suppose that the non-zero columns of each $\mathbf{D}_u^{a,k}$ are distinct, the dual parameter $\lambda', \mu' \in S_{t,y}$, such that $\mathbf{M}^{\lambda', \mu'}(a, k)(e^t - e^y) = 0$, constructs the empty relative interior in the dual parameter space. By the convexity of the objective function, we have $\inf_{\lambda, \mu \notin S_{t,y}} H(\lambda, \mu) = \min_{\lambda, \mu} H(\lambda, \mu)$, due to the density of $(\lambda, \mu) \notin S_{t,y}$.

Overall, under the assumptions of the theorem, we have that $B_y^t$ has a measure of zero. It follows that

$$
\frac{\partial}{\partial \lambda_{u_g}^{(1)}} H(\lambda, \mu) = \sum_{k \in [N]} \sum_{a \in \mathcal{A}} \sum_{y \in [m]} \int_{\{x : h_k^{\lambda, \mu}(x, a) = y\}} \left[ -\left( \left[ \mathbf{D}_{u_g}^{a,k} \right]^\top \eta(x, a, k) \right)_y \right] d\mathcal{P}_{a,k}^X(x) + \xi^g
$$

$$
= \sum_{k \in [N]} \sum_{a \in \mathcal{A}} \int_{\mathcal{X}} -\left[ \eta(x, a, k) \right]^\top \mathbf{D}_{u_g}^{a,k} h_k^{\lambda, \mu}(x, a) d\mathcal{P}_{a,k}^X(x) + \xi^g
$$

$$
= -\mathscr{D}_{u_g}^g(\mathbf{h}^{\lambda, \mu}) + \xi^g \tag{51}
$$

In a similar manner, we can derive

$$
\frac{\partial}{\partial \lambda_{u_g}^{(2)}} H(\lambda, \mu) = \sum_{k \in [N]} \sum_{a \in \mathcal{A}} \int_{\mathcal{X}} \left[ \eta(x, a, k) \right]^\top \mathbf{D}_{u_g}^{a,k} h_k^{\lambda, \mu}(x, a) d\mathcal{P}_{a,k}^X(x) + \xi^g
$$

$$
= \mathscr{D}_{u_g}^g(\mathbf{h}^{\lambda, \mu}) + \xi^g \tag{52}
$$

Considering paired optimal dual parameter $\lambda_{u_g}^{(i)*}, i = 1, 2$, by Lemma B.6, if $\lambda_{u_g}^{(1)*}, \lambda_{u_g}^{(2)*} > 0$, we have

$$
\mathscr{D}_{u_g}^g(\mathbf{h}^{\lambda^*, \mu}) = -\xi^g, \mathscr{D}_{u_g}^g(\mathbf{h}^{\lambda^*, \mu}) = \xi^g,
$$

which leads to a contradiction. If $\lambda_{u_g}^{(1)*} = 0, \lambda_{u_g}^{(2)*} = 0$, we have

$$
\mathscr{D}_{u_g}^g(\mathbf{h}^{\lambda^*, \mu}) \geq -\xi^g, \mathscr{D}_{u_g}^g(\mathbf{h}^{\lambda^*, \mu}) \leq \xi^g.
$$

If $\lambda_{u_g}^{(1)*} = 0, \lambda_{u_g}^{(2)*} > 0$, we have

$$
\mathscr{D}_{u_g}^g(\mathbf{h}^{\lambda^*, \mu}) \geq -\xi^g, \mathscr{D}_{u_g}^g(\mathbf{h}^{\lambda^*, \mu}) = \xi^g.
$$

If $\lambda_{u_g}^{(1)*} > 0, \lambda_{u_g}^{(2)*} = 0$, we have

$$
\mathscr{D}_{u_g}^g(\mathbf{h}^{\lambda^*, \mu}) = -\xi^g, \mathscr{D}_{u_g}^g(\mathbf{h}^{\lambda^*, \mu}) \leq \xi^g.
$$

Overall, we have shown that for all $u_g \in \mathcal{U}_g$, $|\mathscr{D}_{u_g}^g(\mathbf{h}^{\lambda^*, \mu})| \leq \xi^g$.

The local fairness guarantee also can be derived from the optimality of $\mu^*$. The proof techniques are extremely similar to our proof with respect to $\lambda^*$. Hence, we omit the proof of the local fairness guarantee here. The result turns out that $|\mathscr{D}_{u_k}^k(\mathbf{h}^{\lambda, \mu^*})| \leq \xi^k, k \in [N]$.

The next step is to prove that the classifier $\mathbf{h}^{\lambda^*, \mu^*}$ is the optimal solution of the primal problem (1). From the proof above, we can obtain that, for $\forall u_g \in \mathcal{U}_g$,

$$
(\lambda_{u_g}^{(1)} - \lambda_{u_g}^{(2)}) \mathscr{D}^g(\mathbf{h}^{\lambda^*, \mu^*}) - (\lambda_{u_g}^{(1)} + \lambda_{u_g}^{(2)}) \xi^g = 0,
$$

which satisfies the optimality conditions for the dual solution of the constrained optimization problem. The same holds for the local fairness constraints $\mathscr{D}^k(\mathbf{h}^{\lambda^*, \mu^*})$. Consequently, the Lagrangian function equals to risk function when plugging in optimal classifier, $\mathcal{L}(h^{\lambda^*, \mu^*}, \lambda^*, \mu^*) = \mathcal{R}(h^{\lambda^*, \mu^*})$. For any other classifiers $\mathbf{h}'$ that satisfies the global and local fairness constraints, denoting its corresponding dual parameter to maximize the outer problem as $\lambda', \mu'$, it can be deduced that

$$
\mathcal{L}(h^{\lambda^*, \mu^*}, \lambda^*, \mu^*) \leq \mathcal{L}(\mathbf{h}', \lambda', \mu') \leq \mathcal{R}(\mathbf{h}').
$$

Therefore, we arrive at

$$
\mathcal{R}(\mathbf{h}^{\lambda^*, \mu^*}) = \mathcal{L}(h^{\lambda^*, \mu^*}, \lambda^*, \mu^*) \leq \mathcal{R}(\mathbf{h}').
$$

This completes the proof. □

## B.7 Proof of Prosition 6

**Note that** $\lambda \in \mathbb{R}_{\geq 0}^{2|\mathcal{U}_g|}$ and $\mu_k \in \mathbb{R}_{\geq 0}^{2|\mathcal{U}_k|}$, the operator $\|\cdot\|_1$ is linear in dual parameters' domain. We can just write

$$\widehat{H}_k'(\lambda,\mu_k) := \frac{1}{n_k}\sum_{i=1}^{n_k}\sigma_\beta\big(\big[\widehat{\mathbf{M}}^{\lambda,\mu}(a_i,k)\big]^\top \eta(x_i,a_i,k)\big) + \xi^g \sum_{u_g \in \mathcal{U}_g}(\lambda_{u_g}^{(1)} + \lambda_{u_g}^{(2)})$$

$$+ \frac{\xi^k}{\hat{p}_k}\sum_{u_k \in \mathcal{U}_k}(\mu_{k,u_k}^{(1)} + \mu_{k,u_k}^{(2)}),$$

where $\widehat{\mathbf{M}}^{\lambda,\mu}(a,k) := \mathbf{I} - \frac{1}{\hat{p}_{a,k}}\Big[\sum_{u_g \in \mathcal{U}_g}(\lambda_{u_g}^{(1)} - \lambda_{u_g}^{(2)})\widehat{\mathbf{D}}_{u_g}^{a,k} - \sum_{u_k \in \mathcal{U}_k}(\mu_{k,u_k}^{(1)} - \mu_{k,u_k}^{(2)})\widehat{\mathbf{D}}_{u_k}^{a,k}\Big]$ and $\sigma_\beta(x) = \sum_{i=1}^m \frac{\exp(x_i/\beta)}{\sum_{j=1}^m \exp(x_j/\beta)}x_i$.

**Convexity.** The $\widehat{\mathbf{M}}^{\lambda,\mu}(a,k)$ is linear to $\lambda$ and $\mu_k$, and the soft-max operator is convex. Since the composition of an affine mapping and a convex function preserves convexity, $\widehat{H}_k'(\lambda,\mu_k)$ is convex to $\lambda$ and $\mu_k$.

**Smoothness.** Consider the soft-max weighted sum $\sigma_\beta(x) := \sum_{j=1}^m \frac{\exp(x_j/\beta)}{\sum_{\ell=1}^m \exp(x_\ell/\beta)}x_j$, and its Hessian matrix is given by $H_\sigma(x) := \nabla^2 \sigma_\beta(x)$, $[H_\sigma(x)]_{i,j} = \frac{p_i}{\beta}\Big[\big(2 + \frac{x_i - \bar{x}}{\beta}\big)\mathbb{I}(i=j) - p_j\big(2 + \frac{x_i + x_j - \bar{x}}{\beta}\big)\Big]$. For $\forall i,j \in m$, if $\|x\|_1 \leq R$,

$$|[H_\sigma(x)]_{ij}| \leq \frac{1}{\beta}\Big(\big(2 + \tfrac{2R}{\beta}\big) + \big(2 + \tfrac{4R}{\beta}\big)\Big) = \frac{4\beta + 6R}{\beta^2}.$$

Hence, its spectral norm is bounded,

$$\|H_\sigma(x)\|_2 \leq \|H_\sigma(x)\|_F \leq \Big(\sum_{i,j \in [m]}[H_\sigma]_{ij}^2\Big)^{\frac{1}{2}} \leq m\frac{4\beta + 6R}{\beta^2}.$$

Then, there exists a finite constant $L_\sigma := m\frac{4\beta+6R}{\beta^2}$, such that $\|\nabla^2 \sigma_\beta(x)\|_2 \leq L_\sigma$.

For each sample $i = 1,\dots,n_k$, define the affine map

$$z_i(\lambda) := A_i\lambda + b_i, \quad [A_i]_{u_g}^{(j)} = \frac{3j-2}{\hat{p}_{a,k}}[\eta(x_i,a_i,k)]^\top \widehat{\mathbf{D}}_{u_g}^{a_i,k} \quad \text{for } \lambda_{u_g}^{(j)},\ j = 1,2.$$

Set $f_i(\lambda) := \sigma_\beta(z_i(\lambda))$. and let $f_i(\lambda) = \sigma_\beta(z_i(\lambda))$, $\sigma_\beta(x) = \sum_{j=1}^m \frac{e^{x_j/\beta}}{\sum_{\ell=1}^m e^{x_\ell/\beta}}x_j$, By the chain rule and second-order derivatives, $\nabla_\lambda f_i(\lambda) = A_i^\top \nabla_x \sigma_\beta(z_i(\lambda))$, $\nabla_\lambda^2 f_i(\lambda) = A_i^\top[\nabla^2 \sigma_\beta(z_i(\lambda))]A_i$. Hence, due to the boundedness of $\|\lambda\|_1$, the inside $z_i(\lambda)$ is bounded, setting the upper bound as $R$ for simplification here, $\|\nabla_\lambda^2 f_i(\lambda)\|_2 \leq \|A_i\|_2^2 \sup_x \|\nabla^2 \sigma_\beta(x)\|_2 = \|A_i\|_2^2 L_\sigma$, showing $f_i$ is $\|A_i\|_2^2 L_\sigma$-smooth. The linear term in $\lambda$ has zero Hessian. Therefore, since the average of smooth functions is smooth with averaged constants, the function $\widehat{H}_k'(\lambda,\mu_k)$ is $L$-smooth in $\lambda$ with $L = \frac{1}{n_k}\sum_{i=1}^{n_k}\|A_i\|_2^2 L_\sigma$. Following the similar proof procedure, we can obtain the smoothness of $\widehat{H}_k'(\lambda,\mu_k)$ to $\mu_k$. $\qquad\square$

## B.8 Generalization Error For Post-Processing Algorithm

We begin by introducing some notations and simplifications, same as the proof of Theorem 9. Without loss of generality, let $n = n_1 = \cdots = n_k$ present the sample number in local datasets. Denote $p_{a|k} := \mathbb{P}(A = a|K = k)$, $p_{\min} := \min_{a \in \mathcal{A}, k \in [N]} p_{a|k}$. Assume $n_{min} \geq 1$ denotes the sample size of the sensitive group with the fewest observations across all clients.

**Theorem 10.** *If classifiers $\widehat{\mathbf{h}}^* = (\widehat{h}_1^*,\dots,\widehat{h}_k^*)$ with dual parameters $(\widehat{\lambda}^*, \widehat{\mu}^*)$ form an optimal solution of the empirical plug-in estimation of* (7), *denoting* $\rho(n,\delta) = \sqrt{\frac{8|\mathcal{A}|m^2 \log(n+1)}{n}} + \sqrt{\frac{2\log(m^2|\mathcal{A}|N/\delta)}{n}}$, $B_g = \max_{a \in \mathcal{A}, k \in [N]} \|\mathbf{D}_{u_g}^{a,k}\|_1$, $\widehat{B}_g = \max_{a \in \mathcal{A}, k \in [N]} \|\widehat{\mathbf{D}}_{u_g}^{a,k}\|_1$, $\Omega_n^g = \max_{a \in \mathcal{A}, k \in [N]} \|\mathbf{D}_{u_g}^{a,k} -$

$\widehat{\mathbf{D}}_{u_g}^{a,k}\|_\infty$, and $B_k = \max_{a\in\mathcal{A}}\|\mathbf{D}_{u_k}^{a,k}\|_1, \widehat{B}_k = \max_{a\in\mathcal{A}}\|\widehat{\mathbf{D}}_{u_k}^{a,k}\|_1, \Omega_n^k = \max_{a\in\mathcal{A}}\|\mathbf{D}_{u_k}^{a,k} - \widehat{\mathbf{D}}_{u_k}^{a,k}\|_\infty, k\in[N]$.

(1) Let $0 < \delta < 1$, suppose that $n > \frac{2|\mathcal{A}|N\widehat{B}_k}{p_{\min}\xi^g} + \frac{1}{2p_{\min}^2}\log\frac{1}{\delta}$, then with probability at least $1 - 2|\mathcal{A}|\delta$,

$$|\mathscr{D}^g(\widehat{\mathbf{h}}^*)| \leq \xi^g + \mathcal{O}(|\mathcal{A}|NB_g\rho(n,\delta/|\mathcal{A}||\mathcal{U}_g|)) + \Omega_n^g + \frac{|\mathcal{A}|N\widehat{B}_g}{n_{\min}}.$$

(2) Let $0 < \delta < 1$, suppose that $n > \frac{2|\mathcal{A}|\widehat{B}_g}{p_{\min}\xi^k} + \frac{1}{2p_{\min}^2}\log\frac{1}{\delta}$, then with probability at least $1 - 2|\mathcal{A}|\delta$,

$$|\mathscr{D}^k(\widehat{\mathbf{h}}^*)| \leq \xi^k + \mathcal{O}(|\mathcal{A}|B_k\rho(n,N\delta/|\mathcal{A}||\mathcal{U}_g|)) + \Omega_n^k + \frac{|\mathcal{A}|\widehat{B}_k}{n_{\min}}, \; k\in[N].$$

The proof of Theorem 10 needs the following lemma.

**Lemma B.7.** *Let* $X_1,\ldots,X_n$ *be independent* Bernoulli$(p)$ *random variables and define* $S_n = \sum_{i=1}^n X_i$. *Fix any* $M \in (0, np)$ *and confidence level* $\delta \in (0, 1)$. *If the sample size satisfies* $n \geq \frac{2M}{p} + \frac{1}{2p^2}\log\frac{1}{\delta}$, *then we have* $\mathbb{P}(S_n > M) \geq 1 - \delta$.

*Proof of Lemma B.7.* By Hoeffding's inequality, for any $t > 0$, $\mathbb{P}(S_n - \mathbb{E}[S_n] \leq -t) \leq \exp\left(-\frac{2t^2}{n}\right)$. Since $\mathbb{E}[S_n] = np$, set $t = np - M$. Then

$$\mathbb{P}(S_n \leq M) = \mathbb{P}(S_n - np \leq -(np - M)) \leq \exp\left(-\frac{2(np - M)^2}{n}\right).$$

To guarantee $\mathbb{P}(S_n \leq M) \leq \delta$, it suffices that $\frac{2(np-M)^2}{n} \geq \log\frac{1}{\delta}$. Substitute $n = \frac{2M}{p} + \frac{1}{2p^2}\log\frac{1}{\delta}$. Then $np - M = \left(\frac{2M}{p} + \frac{1}{2p^2}\log\frac{1}{\delta}\right)p - M = M + \frac{1}{2p}\log\frac{1}{\delta}$, and one can check

$$\frac{2(np - M)^2}{n} = \frac{2\left(M + \frac{1}{2p}\log\frac{1}{\delta}\right)^2}{\frac{2M}{p} + \frac{1}{2p^2}\log\frac{1}{\delta}} \geq \log\frac{1}{\delta}.$$

Hence $\mathbb{P}(S_n \leq M) \leq \delta$, i.e. $\mathbb{P}(S_n > M) \geq 1 - \delta$. $\qquad\square$

### B.8.1  Proof of Theorem 10

We first consider the generalization error of the fairness constraints. Without loss of generalization, here we only prove the generalization error for global fairness constraints and corresponding parameter $\lambda$. The proof technique for local fairness constraints and corresponding parameter $\mu$ is extremely similar to that for global fairness constraints.

We know that the personalized attribute-aware empirical classifier can be written as

$$\widehat{h}_k^{\widehat{\lambda}^*,\widehat{\mu}^*}(x,a) := \arg\max_{y\in[m]} \left[\widehat{\mathbf{M}}^{\widehat{\lambda}^*,\widehat{\mu}^*}(a,k)\right]^\top \widehat{\eta}(x,a,k) \tag{53}$$

As $h$ depends on the Bayes score function $\eta$, we can consider the input as $(\eta_{k,i} := \eta(x_{k,i}, a_{k,i}, k), a_{k,i}, y_{k,i})$. Let $\ell_{i,j}^{a',k}(\eta, a, y; h) = \mathbb{I}(y = i \wedge h(\eta) = j \wedge a = a')$. Then we have $\mathbf{C}_{i,j}^{a',k}(h) = \mathbb{E}[\ell_{i,j}^{a',k}(\eta, y, a; h)]$ and $\widehat{\mathbf{C}}_{i,j}^{a'}(h) = \frac{1}{n}\sum_{z=1}^n \ell_{i,j}^{a',k}(\eta_{k,z}, a_{k,z}, y_{k,z})$. Then we turn to consider the VC dimension of the function class $\mathcal{H}_{i,j,a'} := \{h : (x, a, y) \to \mathbb{I}(y = i \wedge h(\eta) = j \wedge a = a')\}$. Thanks to the classifier's specific structural form (53), we can directly state an explicit upper bound on its VC dimension: for given class $j$,

$$\widehat{h}_k(x,a) = j \Leftrightarrow [\eta(x,a,k)]^\top \left(\left[\widehat{\mathbf{M}}^{\widehat{\lambda}^*,\widehat{\mu}^*}(a,k)\right]_{:,j} - \left[\widehat{\mathbf{M}}^{\widehat{\lambda}^*,\widehat{\mu}^*}(a,k)\right]_{:,i}\right) \geq 0, \; \forall i \neq j \in [m],$$

which can be regarded as the intersection of $m - 1$ half-spaces given $\eta_{k,i}, a_{k,i}$. A single halfspace function class can be viewed as the class of linear classifiers, possessing a VC dimension of $m$. By the additive property of VC dimension, for function classes $\{\mathcal{G}_i\}_{i=1}^m$, $VC\left(\bigwedge_{i=1}^m \mathcal{G}_i\right) \leq \sum_{i=1}^m VC(\mathcal{G}_i)$,

the function class $\mathcal{H}_{i,j,a'}$ has VC dimension at most $\mathcal{O}(|\mathcal{A}|m^2)$. By taking a union bound in the Lemma B.3, we have that with probability at least $1-\delta$,

$$\sup_{h\in\mathcal{F}_{\mathcal{H}}} \max_{k\in[N]} \max_{a\in\mathcal{A}} \left\|\mathbf{C}^{a,k}(h) - \widehat{\mathbf{C}}^{a,k}(h)\right\|_\infty \leq \mathcal{O}\left(\sqrt{\frac{8|\mathcal{A}|m^2\log(n+1)}{n}} + \sqrt{\frac{2\log(m^2|\mathcal{A}|N/\delta)}{n}}\right)$$

$$:= \mathcal{O}(\rho(n,\delta)).$$

Hence, for the global fairness constraints $\mathscr{D}_{u_g}^g$ with the empirical optimal solution $\widehat{\mathbf{h}}^*$, by the generalization bound in (29), we have that,

$$\mathscr{D}_{u_g}^g(\widehat{\mathbf{h}}) - \widehat{\mathscr{D}}_{u_g}^g(\widehat{\mathbf{h}}) \leq \mathcal{O}(\rho(n,\delta/|\mathcal{A}|))|\mathcal{A}|NB_g + \Omega_n^g.$$

Now we consider the bound on empirical $\widehat{\mathscr{D}}_{u_g}^g(\widehat{\mathbf{h}}^*)$. The empirical optimal dual parameter $\widehat{\lambda}^*$ and $\widehat{\mu}^*$ are obtained by the empirical dual function:

$$\widehat{H}(\lambda,\mu) = \sum_{k\in[N]}\sum_{a\in\mathcal{A}} \hat{p}_{a,k} \sum_{i=1}^{n_{a,k}} \left[\max_{y\in[m]}\left(\left[\widehat{\mathbf{M}}^{\lambda,\mu}(a,k)\right]^\top \widehat{\eta}(x_i,a,k)\right)_y\right] + \xi^g \sum_{u_g\in\mathcal{U}_g}(\lambda_{u_g}^{(1)}+\lambda_{u_g}^{(2)})$$

$$+ \sum_{k\in[N]}\xi^k \sum_{u_k\in\mathcal{U}_k}(\mu_{k,u_k}^{(1)}+\mu_{k,u_k}^{(2)}). \tag{54}$$

This representation is fully consistent with that given in (8) restricting to group-$a$ observations within the $k$-th client's data, where $n_{a,k}$ denotes the sample number of group $a$ in client $k$. Considering the subgradient of the empirical dual function w.r.t. $\lambda_{u_g}^{(1)}$, by the additivity of subgradient,

$$\frac{\partial}{\partial\lambda_{u_g}^{(1)}}\widehat{H}(\lambda,\mu) = \sum_{k\in[N]}\sum_{a\in\mathcal{A}} \hat{p}_{a,k}\frac{1}{n_{a,k}}\sum_{i=1}^{n_{a,k}}\left[\frac{\partial}{\partial\lambda_{u_g}^{(1)}}\max_{y\in[m]}\left(\left[\widehat{\mathbf{M}}^{\lambda,\mu}(a,k)\right]^\top\widehat{\eta}(x_i,a,k)\right)_y\right] + \xi^g. \tag{55}$$

Denoting empirical score function $\widehat{f}(x,a,k) = \left[\widehat{\mathbf{M}}^{\lambda,\mu}(a,k)\right]^\top\widehat{\eta}(x,a,k)$, by Lemma B.5, we have

$$\sum_{i=1}^{n_{a,k}}\left[\frac{\partial}{\partial\lambda_{u_g}^{(1)}}\max_{y\in[m]}\left(\left[\widehat{\mathbf{M}}^{\lambda,\mu}(a,k)\right]^\top\widehat{\eta}(x_i,a,k)\right)_y\right]$$

$$= \sum_{i=1}^{n_{a,k}}\sum_{y\in[m]}\mathbb{I}(\widehat{h}_k(x_i,a)=y)\left[\frac{\partial}{\partial\lambda_{u_g}^{(1)}}\max_{y\in[m]}\left(\left[\widehat{\mathbf{M}}^{\lambda,\mu}(a,k)\right]^\top\widehat{\eta}(x_i,a,k)\right)_y\right] + \xi^g$$

$$= \frac{1}{\hat{p}_{a,k}}\sum_{i=1}^{n_{a,k}}\sum_{y\in[m]}\mathbb{I}(\widehat{h}_k(x_i,a)=y)\left[\text{conv}\left(\bigcup_{i\in\arg\max_{j\in[m]}\widehat{f}_j(x,a,k)} -[\widehat{\eta}(x_i,a,k)]^\top\widehat{\mathbf{D}}_{u_g}^{a,k}e_i\right)\right]$$

$$= \frac{1}{\hat{p}_{a,k}}\sum_{i=1}^{n_{a,k}}\sum_{y\in[m]}\left\{-\mathbb{I}\left(\widehat{f}_y(x_i,a,k)>\widehat{f}_j(x_i,a,k),\forall j\neq y, j\in[m]\right)[\widehat{\eta}(x_i,a,k)]^\top\widehat{\mathbf{D}}_{u_g}^{a,k}e_y\right.$$

$$\left. + \mathbb{I}\left(x_i\in B_y^t\right)\left[-b_t[\widehat{\eta}(x,a,k)]^\top\widehat{\mathbf{D}}_{u_g}^{a,k}(e_t-e_y)\right]\right\}$$

where $B_y^t := \{x : \exists t\neq y, \widehat{f}_t(x,a,k)\geq\widehat{f}_i(x,a,k),\forall i\in[m]; \widehat{f}_t(x,a,k)=\widehat{f}_y(x,a,k)\}$ and $b_t\in[0,1]$. According to Carathéodory's theorem, the subgradient interval can still be represented by two points. According to our assumption, the plug-in estimator $\widehat{\eta}$ still meet the continuity assumption and we exclude singular $\lambda',\mu'$. Therefore, we know that

$$\mathbb{P}\left(\sum_{i=1}^{n_{a,k}}\mathbb{I}\left(\exists t\neq y, \widehat{f}_t(x_i,a,k)\geq\widehat{f}_i(x_i,a,k),\forall i\in[m]; \widehat{f}_t(x_i,a,k)=\widehat{f}_y(x_i,a,k)\right)\leq 1\right)=1$$

$$\tag{56}$$

Hence, the subgradient falls into an interval. Since $[\widehat{\eta}(x_i, a, k)]^\top \widehat{\mathbf{D}}_{u_g}^{a,k} e_t \leq \|[\widehat{\eta}(x_i, a, k)]^\top \widehat{\mathbf{D}}_{u_g}^{a,k}\|_1 \leq B_g$, denoting $n_{\min} := \min_{a \in \mathcal{A}, k \in [N]} n_{a,k}$ and $\widehat{B}_g$, we have that

$$\frac{\partial}{\partial \lambda_{u_g}^{(1)}} \widehat{H}(\lambda, \mu) \leq \sum_{k \in [N]} \sum_{a \in \mathcal{A}} \frac{1}{n_{a,k}} \sum_{i=1}^{n_{a,k}} \left[ -[\widehat{\eta}(x_i, a, k)]^\top \widehat{\mathbf{D}}_{u_g}^{a,k} \widehat{h}_k(x_i, a) \right] + \xi^g \tag{57}$$

$$+ \sum_{k \in [N]} \sum_{a \in \mathcal{A}} \frac{1}{n_{a,k}} \widehat{B}_g \tag{58}$$

$$\leq -\widehat{\mathscr{D}}_{u_g}^g(\widehat{\mathbf{h}}) + \xi^g + \frac{|\mathcal{A}| N \widehat{B}_g}{n_{\min}} \tag{59}$$

On the other hand,

$$\frac{\partial}{\partial \lambda_{u_g}^{(1)}} \widehat{H}(\lambda, \mu) \geq \sum_{k \in [N]} \sum_{a \in \mathcal{A}} \frac{1}{n_{a,k}} \sum_{i=1}^{n_{a,k}} \left[ -[\widehat{\eta}(x_i, a, k)]^\top \widehat{\mathbf{D}}_{u_g}^{a,k} \widehat{h}_k(x_i, a) \right] + \xi^g \tag{60}$$

$$- \sum_{k \in [N]} \sum_{a \in \mathcal{A}} \frac{1}{n_{a,k}} \widehat{B}_g \tag{61}$$

$$= -\widehat{\mathscr{D}}_{u_g}^g(\widehat{\mathbf{h}}) + \xi^g - \frac{|\mathcal{A}| N \widehat{B}_g}{n_{\min}} \tag{62}$$

Hence, we obtain that

$$\frac{\partial}{\partial \lambda_{u_g}^{(1)}} \widehat{H}(\lambda, \mu) - \xi^g \subset \left[ -\widehat{\mathscr{D}}_{u_g}^g(\widehat{\mathbf{h}}^*) - \frac{|\mathcal{A}| N \widehat{B}_g}{n_{\min}}, -\widehat{\mathscr{D}}_{u_g}^g(\widehat{\mathbf{h}}^*) + \frac{|\mathcal{A}| N \widehat{B}_g}{n_{\min}} \right].$$

In a similar manner, we can derive the range of subgradient for $\lambda_{u_g}^{(2)}$,

$$\frac{\partial}{\partial \lambda_{u_g}^{(2)}} \widehat{H}(\lambda, \mu) - \xi^g \subset \left[ \widehat{\mathscr{D}}_{u_g}^g(\widehat{\mathbf{h}}^*) - \frac{|\mathcal{A}| N \widehat{B}_g}{n_{\min}}, \widehat{\mathscr{D}}_{u_g}^g(\widehat{\mathbf{h}}^*) + \frac{|\mathcal{A}| N \widehat{B}_g}{n_{\min}} \right].$$

Since we assume that $n > \frac{2|\mathcal{A}| N \widehat{B}_k}{p_{\min} \xi^g} + \frac{1}{2p_{\min}^2} \log \frac{1}{\delta}$, by Lemma B.7, we have that with probability at last $1 - |\mathcal{A}|\delta$,

$$n_{\min} \geq \frac{|\mathcal{A}| N \widehat{B}_g}{\xi^g} \Leftrightarrow \xi^g \geq \frac{|\mathcal{A}| N \widehat{B}_g}{n_{\min}}.$$

Consider the optimality of $\widehat{\lambda}_{u_g}^{(1)}, \widehat{\lambda}_{u_g}^{(2)}$, by Lemma B.6, if $\widehat{\lambda}_{u_g}^{(1)} > 0, \widehat{\lambda}_{u_g}^{(2)} > 0$, we have $0 \in \frac{\partial}{\partial \lambda_{u_g}^{(1)}} \widehat{H}(\widehat{\lambda}^*, \mu), 0 \in \frac{\partial}{\partial \lambda_{u_g}^{(2)}} \widehat{H}(\widehat{\lambda}^*, \mu)$. Thus,

$$|\widehat{\mathscr{D}}_{u_g}^g(\widehat{\mathbf{h}}^*) - \xi^g| \leq \frac{|\mathcal{A}| N \widehat{B}_g}{n_{\min}}$$

$$|\widehat{\mathscr{D}}_{u_g}^g(\widehat{\mathbf{h}}^*) + \xi^g| \leq \frac{|\mathcal{A}| N \widehat{B}_g}{n_{\min}},$$

which leads to a contradiction. For other cases, such as $\widehat{\lambda}_{u_g}^{(1)} = \widehat{\lambda}_{u_g}^{(2)} = 0$; $\widehat{\lambda}_{u_g}^{(1)} > 0, \widehat{\lambda}_{u_g}^{(2)} = 0$, and $\widehat{\lambda}_{u_g}^{(1)} = 0, \widehat{\lambda}_{u_g}^{(2)} > 0$, as discussed in the proof of Theorem 5, it turns out that

$$|\widehat{\mathscr{D}}_{u_g}^g(\widehat{\mathbf{h}}^*)| \leq \xi^g + \frac{|\mathcal{A}| N \widehat{B}_g}{n_{\min}}.$$

By taking a union bound, we obtain that with probability at least $1 - 2|\mathcal{A}|\delta$,

$$|\mathscr{D}^g(\widehat{\mathbf{h}}^*)| \leq \xi^g + \mathcal{O}(\rho(n, \delta/|\mathcal{A}||\mathcal{U}_g|))|\mathcal{A}|NB_g + \Omega_n^g + \frac{|\mathcal{A}|N\widehat{B}_g}{n_{\min}}.$$

For local fairness constraints $\mathscr{D}^k(\widehat{\mathbf{h}}^*)$, following the same proof procedures, we arrive at that with probability at least $1 - 2|\mathcal{A}|\delta$,

$$|\mathscr{D}^k(\widehat{\mathbf{h}}^*)| \leq \xi^k + \mathcal{O}(\rho(n, N\delta/|\mathcal{A}||\mathcal{U}_g|))|\mathcal{A}|B_k + \Omega_n^k + \frac{|\mathcal{A}|\widehat{B}_k}{n_{\min}}.$$

$\square$

# C  Additional Datasets and Experimental Setting

## C.1  Datasets and Experimental Details

### C.1.1  Datasets

- The **Compas** dataset [22] comprises 6,172 criminal defendants from Broward County, Florida, between 2013 and 2014, with the task of predicting whether a defendant will recidivate within two years of their initial risk assessment. We consider the race of each individual as the sensitive attribute and train a logistic classifier as our prediction model.

- The **Adult** dataset [4] comprises more than 45000 samples based on 1994 U.S. census data, where the task is to predict whether the annual income of an individual is above \$50,000. We consider the gender of each individual as the sensitive attribute and train the logistic regression as the classification model.

- The **ENEM** dataset [40] contains about 1.4 million samples from Brazilian college entrance exam scores along with student demographic information. We follow [3] to quantized the exam score into 2 or 5 classes as label, and consider race as sensitive attribute. As [3] used a random subset of 50K samples, we instead sample 100K data points to construct our federated dataset. We train multilayer perceptron (MLP) as the classification model.

- The **CelebA** dataset [96] is a facial image dataset consists of about 200k instances with 40 binary attribute annotations. We identify the binary feature *smile* as target attributes which aims to predict whether the individuals in the images exhibit a smiling expression. The *race* of individuals is chosen as sensitive attribute. We train Resnet18 [38] on CelebA as the classification model.

The determination of sensitive attributes and labels on three datasets has been verified significant in previous research [3, 34].

### C.1.2  Baselines

We compare the performance of FedFACT with traditional **FedAvg** [55] and five SOTA methods tailored for calibrating global and local fairness in FL, namely **FairFed** [31], **FedFB** [93], **FCFL** [18], **praFFL** [85], and the method in [25], denoted as **Cost** in our experiments.

- **FedAvg** serves as a core Federated Learning model and provides the baseline for our experiments. It works by computing updates on each client's local dataset and subsequently aggregating these updates on a central server via averaging.

- **FairFed** introduces an approach to adaptively adjust the aggregation weights of different clients based on their local fairness metric to train federated model with global fairness guarantee.

- **FedFB** presents a FairBatch-based approach [67] to compute the coefficients of FairBatch parameters on the server. This method integrates global reweighting for each client into the FedAvg framework to fulfill fairness objectives.

- **FCFL** proposed a two-stage optimization to solve a multi-objective optimization with fairness constraints. The prediction loss at each local client is treated as an objective, and FCFL maximize the worst-performing client while considering fairness constraints by optimizing a surrogate maximum function involving all objectives.

- **praFFL** proposed a preference-aware federated learning scheme that integrates client-specific preference vectors into both the shared and personalized model components via a hypernetwork. It is theoretically proven to linearly converge to Pareto-optimal personalized models for each client's preference.

- **[25]** proposed a convex-programming-based post-processing framework that characterizes and enforces the minimum accuracy loss required to satisfy specified levels of both local and global fairness constraints in multi-class federated learning by approximating the region under the ROC hypersurface with a simplex and solving a linear program, denoted as **Cost** in our experiments.

Meanwhile, we adapt FedFACT to focus solely on global or local fairness in FL, denoted as FedFACT$_g$ and FedFACT$_l$. FedFACT$_{g\&l}$ indicates the algorithm simultaneously achieving global and local fairness. The FedFACT (In) presents the in-processing method and FedFACT (Post) presents the post-processing method.

### C.1.3 Parameter Settings

We provide hyperparameter selection ranges for each model in Table 4. For all other hyperparameters, we follow the codes provided by authors and retain their default parameter settings.

Table 4: Hyperparameter Selection Ranges

| Model | Hyperparameter | Ranges |
|---|---|---|
| **General** | Learning rate | {0.0001, 0.001, 0.003, 0.005, 0.01, 0.03 ,0.05} |
| | Global round | {20, 30, 50, 80} |
| | Local round | {10, 20, 30, 50} |
| | Local batch size | {128, 256, 512} |
| | Hidden layer | {16, 32, 64} |
| | Optimizer | {Adam, SGD} |
| **FedFB** | Step size ($\alpha$) | {0.005, 0.01, 0.05, 0.3} |
| **FairFed** | Fairness budget ($\beta$) | {0.01, 0.05, 0.5, 1} |
| | Local debiasing ($\alpha$) | {0.005, 0.01, 0.05} |
| **FCFL** | Fairness constraint ($\epsilon$) | {0.01, 0.03, 0.05, 0.07} |
| **praFFL** | Diversity ($\tau_p$) | {10, 15, 20} |
| **FedFACT (In)** | Classifier number | 1 |
| | $w_k^t$ learning rate ($\eta_w$) | {0.03, 0.3} |
| | Dual parameter bound | 5 |
| **FedFACT (Post)** | Temperature $\beta$ | 0.1 |
| | Dual parameter bound | 5 |

For the fairness-control parameters, e.g., the parameter $\lambda$ in praFFL [85] and the global and local fairness constraints in Cost [25], we impose stringent fairness requirements on the model in our overall comparative experiments, and we adjust the parameters governing the fairness metrics in the Pareto-curve experiments.

### C.1.4 Experiments Compute Resources

We conducted our experiments on a GPU server equipped with 8 CPUs and two NVIDIA RTX 4090s (24G).

### C.2 Discussion about FedFACT and LoGoFair [95]

LogoFair [95] is designed for binary-classification in federated learning under both global and local fairness constraints, seeking the Bayes-optimal classifier. By deriving a closed-form solution for the fair Bayes classifier, LogoFair reformulates the post-processing fairness adjustment as a bilevel optimization problem jointly solved by the server and clients, which is an approach conceptually analogous to our post-processing framework. In binary classification, FedFact and LogoFair both target Bayes-optimal classifiers under constraints disparity metrics expressed in linear form. Theoretically, for an identical fairness metric, our Bayes-optimal fair classifier characterization covers that of LogoFair. Consequently, we refrain from performing a comparative evaluation of the two approaches.

Our method differs by defining the loss at the client level, thereby achieving lower estimation error than the local group-specific objective in [95]. Crucially, by formulating the post-processing model over the probabilistic simplex instead of restricting outputs to the unit interval $[0, 1]$ in the binary case, our framework achieves enhanced scalability and naturally adaptable to multi-group, multiclass settings.

Note that, whether for binary or multiclass settings, our implementation of FedFACT is based on calibrating confusion matrices over the multi-dimensional probabilistic simplex.

## C.3 Heterogeneous Split of Client Distribution

We propose a partitioning method that introduces heterogeneous correlations between the sensitive attribute $A$ and label $Y$, thereby further elucidating the trade-off between global fairness and local fairness [33].

**Heterogeneous Split.** We assume a dataset $D$ of $n$ samples, each with a binary attribute $A$ and a binary label $Y$. We denote by $n_{ij} = |\{x_\ell, a_\ell, y_\ell : (a_\ell = i, \, y_\ell = j)\}|$ the number of samples in joint class $(i, j)$ for $i, j \in \{0, 1\}$. Our goal is to partition $D$ into $N$ disjoint subsets (one per client) such that in client $k \in [N]$, the correlation between $A$ and $Y$ is controlled by a target parameter $\gamma_k \in [a, b] \subseteq [0, 1]$. To achieve this, we first assign each client $k$ a weight $\gamma_k$

$$w_k^{(i,j)} = \begin{cases} \gamma_k, & (i,j) \in \{(0,0), (1,1)\}, \\ 1 - \gamma_k, & (i,j) \in \{(1,0), (0,1)\}. \end{cases}$$

Then for each joint class $(i, j)$ we compute the total weight $W^{(i,j)} = \sum_{k=1}^{n} w_k^{(i,j)}$ and assign to client $k$ a preliminary count $c_k^{(i,j)} = \lfloor (w_k^{(i,j)}/W^{(i,j)}) \, n_{ij} \rfloor$. Any remaining samples are distributed one by one to the clients with the largest fractional remainders, so that $\sum_{k=1}^{N} c_k^{(i,j)} = n_{ij}$. Finally, for each class $(i, j)$ we shuffle its $n_{ij}$ sample indices and slice them into blocks of size $c_k^{(i,j)}$. Client $k$ then collects all its four blocks across $(i, j)$, yielding a partition that in expectation realizes the desired within-client correlation $\gamma_k$ between $A$ and $Y$.

This approach can be regarded as a generalization of the synergy-level-based heterogeneous split in [33] to the multi-client setting, where the $A$-$Y$ correlation for each client is governed by a parameter randomly drawn from $[a, b] \subseteq [0, 1]$, thereby yielding a more pronounced balance between global fairness and local fairness. Throughout the experimental evaluation, we set $\gamma_k \in [0.2, 0.8]$ to guarantee that every client has a sufficient number of sensitive group samples to assess local fairness.

# D Detailed Experiments Results

## D.1 Comparison Result and binary EOP criterion

**Parato Curves of DP.** We have already presented the numerical comparison between our proposed method and the baselines in the main text; here, we report the Pareto curves illustrating the trade-off between global fairness and accuracy. More precisely, we compare the trade-off between accuracy and the global fairness measure, as well as the trade-off between accuracy and the local fairness measure, as a function of the fairness constraint.

The Pareto curve for the global DP criterion is shown in Figure 2 where the horizontal axis denotes accuracy and the vertical axis represents the fairness metric. Consequently, models located closer to the upper-right corner exhibit superior accuracy-fairness trade-offs. As illustrated in Figure 2, our method outperforms all existing state-of-the-art approaches when comparing accuracy against either global fairness in isolation.

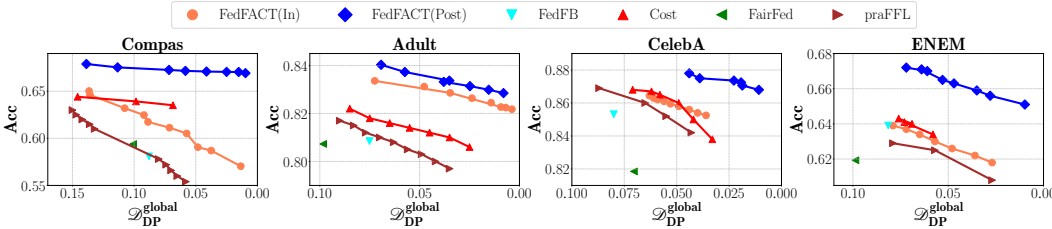

Figure 2: The Pareto frontier on Compas, Adult, CelebA and ENEM datasets. The curve closer to the upper right corner indicates a better trade-off between accuracy and fairness.

This result not only demonstrates that our model achieves a more favorable accuracy-fairness balance but also highlights its controllability: by tuning the fairness constraints, one can satisfy diverse fairness requirements.

**Parato Curves of EOP.** In Figure 3, we illustrate the Pareto curve for the Equalized Odds (EO) criterion-accuracy. Because EOP enforces tighter constraints than DP, precise adherence in a federated context requires large per-group sample counts at each client. Hence, we also compare the global EOP here. Our framework still exceeds all state-of-the-art baselines in trading off accuracy against fairness.

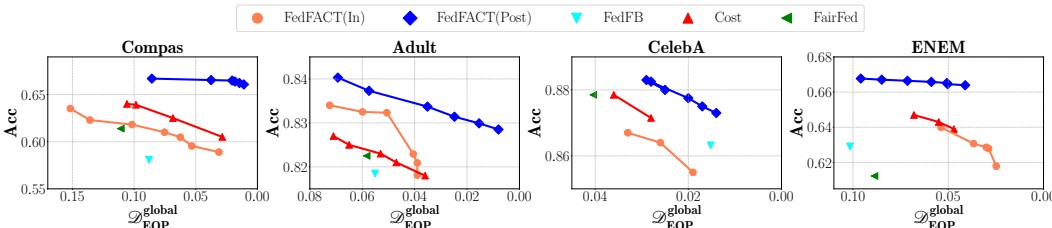

Figure 3: The Pareto frontier on Compas, Adult, CelebA and ENEM datasets. The curve closer to the upper right corner indicates a better trade-off between accuracy and fairness.

### D.2 Details for Multi-Class Classification

**Multi-Class fair datasets.** We illustrate how FedFACT performs on multi-class prediction using CelebA and ENEM. For CelebA, with 'Gender' still serving as the sensitive attribute, We employ the binary attributes "Smile" and "Big_Nose" to construct a multiclass task by mapping their joint values $\{0, 1\} \times \{0, 1\}$ onto a four-class label set $\{0, 1, 2, 3\}$, thereby formulating a multiclass classification problem on the CelebA dataset. These attributes are commonly used in centralized machine learning literature [13, 97] to construct fairness-aware classification tasks. For ENEM, we follow [3] to quantize the Humanities exam score to 5 classes. In order to guarantee adequate per-group sample sizes at each client in heterogeneous settings for fairness evaluation (or some clients only hold less than 10 samples for specific group under heterogeneous partitioning), we adopt the four race labels "Branca," "Preta," "Parda," and "Amarela" from the Race attribute as the sensitive groups. These datasets are partitioned into five clients under a heterogeneous split with $\gamma = 1$.

**Evaluation.** In terms of baselines, only the Cost [25] algorithm is theoretically applicable to fairness optimization in multiclass federated learning scenarios. However, their experiments and code are limited to binary classification, and have already been used as binary baselines for comparison with our method. Consequently, we focus exclusively on reporting FedFACT's performance along with FedAvg in multiclass fairness, establishing it as a pioneering approach in this setting.

### D.3 Additional Experiments for Adjusting Accuracy-Fairness Trade-Off

In Table 5, we present additional experiments on the Compas and Adult datasets under the heterogeneous split to illustrate the adjustment of the accuracy-fairness trade-off. Compared to the results in the main text, this partitioning yields a more pronounced trade-off between global and local fairness.

Table 5: Additional Accuracy-Fairness Balance.

| Dataset | Compas (In-) | | | Adult (In-) | | | Compas (Post-) | | | Adult (Post-) | | |
|---|---|---|---|---|---|---|---|---|---|---|---|---|
| $(\xi^g, \xi^l)$ | Acc | $\mathscr{D}^{global}$ | $\mathscr{D}^{local}$ | Acc | $\mathscr{D}^{global}$ | $\mathscr{D}^{local}$ | Acc | $\mathscr{D}^{global}$ | $\mathscr{D}^{local}$ | Acc | $\mathscr{D}^{global}$ | $\mathscr{D}^{local}$ |
| (0.00,0.00) | 60.22 | 0.0404 | 0.0745 | 80.99 | 0.0021 | 0.0407 | 64.56 | 0.0083 | 0.0075 | 81.25 | 0.0139 | 0.0275 |
| (0.02,0.00) | 60.61 | 0.0436 | 0.0734 | 81.04 | 0.0021 | 0.0423 | 64.78 | 0.0091 | 0.0099 | 81.56 | 0.0146 | 0.0285 |
| (0.04,0.00) | 60.90 | 0.0490 | 0.0737 | 81.09 | 0.0039 | 0.0446 | 65.04 | 0.0123 | 0.0099 | 81.62 | 0.0147 | 0.0285 |
| (0.00,0.02) | 60.80 | 0.0499 | 0.0744 | 81.18 | 0.0046 | 0.0411 | 64.94 | 0.0146 | 0.0214 | 81.82 | 0.0238 | 0.0381 |
| (0.02,0.02) | 61.03 | 0.0503 | 0.0726 | 81.64 | 0.0315 | 0.0463 | 65.12 | 0.0311 | 0.0306 | 82.04 | 0.0240 | 0.0381 |
| (0.04,0.02) | 61.32 | 0.0555 | 0.0774 | 81.65 | 0.0318 | 0.0467 | 65.57 | 0.0378 | 0.0371 | 82.16 | 0.0257 | 0.0397 |
| (0.00,0.04) | 61.18 | 0.0581 | 0.0804 | 81.31 | 0.0053 | 0.0444 | 65.16 | 0.0294 | 0.0517 | 82.46 | 0.0350 | 0.0492 |
| (0.02,0.04) | 61.39 | 0.0644 | 0.0753 | 81.67 | 0.0177 | 0.0452 | 65.16 | 0.0412 | 0.0419 | 82.49 | 0.0346 | 0.0497 |
| (0.04,0.04) | 62.39 | 0.0878 | 0.0966 | 82.14 | 0.0486 | 0.0497 | 65.82 | 0.0507 | 0.0574 | 82.63 | 0.0350 | 0.0518 |

Note that the gap between the imposed constraints and the observed fairness metrics stems from the **inevitable generalization error** incurred with finite local samples. Consequently, global fairness exhibits greater controllability than local fairness. In practice, FedFACT remains capable of

tuning the accuracy-fairness balance according to the specified fairness constraints, highlighting the controllability inherent in our approach.

## D.4 Hyper-Parameter Experiments

In this subsection, we examine the impact of the number of classifiers in the in-processing method. Specifically, we incrementally increase the size of the weighted ensemble—from using only the most recently trained classifier up to including the ten preceding classifiers. Let $N_h$ represent the number of classifiers comprising the weighted ensemble. As reported in Table 6, we observe that augmenting the ensemble with multiple classifiers yields negligible improvements and can even degrade performance when earlier classifiers have not been fully trained. Consequently, in light of these empirical findings, all in-processing experiments in this work utilize only the single most recently obtained classifier.

Table 6: Hyper-Parameter Experimental Results.

| | Compas | | | Adult | | | CelebA | | | ENEM | | |
|---|---|---|---|---|---|---|---|---|---|---|---|---|
| $N_h$ | Acc | $\mathscr{D}^{global}$ | $\mathscr{D}^{local}$ | Acc | $\mathscr{D}^{global}$ | $\mathscr{D}^{local}$ | Acc | $\mathscr{D}^{global}$ | $\mathscr{D}^{local}$ | Acc | $\mathscr{D}^{global}$ | $\mathscr{D}^{local}$ |
| 1 | 61.17 | 0.0407 | 0.0732 | 82.04 | 0.0014 | 0.0401 | 86.15 | 0.0382 | 0.0473 | 65.33 | 0.0293 | 0.0387 |
| 2 | 61.29 | 0.0408 | 0.0731 | 81.24 | 0.0015 | 0.0416 | 85.54 | 0.0382 | 0.0482 | 65.54 | 0.0285 | 0.0392 |
| 5 | 61.18 | 0.0410 | 0.0723 | 81.63 | 0.0032 | 0.0397 | 85.91 | 0.0377 | 0.0472 | 65.41 | 0.0307 | 0.0390 |
| 10 | 61.14 | 0.0404 | 0.0736 | 81.91 | 0.0048 | 0.0399 | 86.59 | 0.0384 | 0.0471 | 65.11 | 0.0398 | 0.0383 |

## D.5 Efficiency and Scalability Study

In this section, we conduct out experiments with DP criterion to examine the communication cost and scalability of FedFACT.

**Efficiency.** We evaluate the communication efficiency of FedFACT by monitoring its performance across varying numbers of communication rounds $T$. As illustrated in Figure 4, the post-processing method, built upon a fully trained pre-trained model, consistently achieves convergence in fewer than 10 communication rounds, underscoring its high efficiency. The in-processing method likewise converges in under 40 iterations; given that it requires training the federated model from scratch, this performance is comparable to the convergence speed of FedAvg, making it highly effective compared to existing federated learning algorithms.

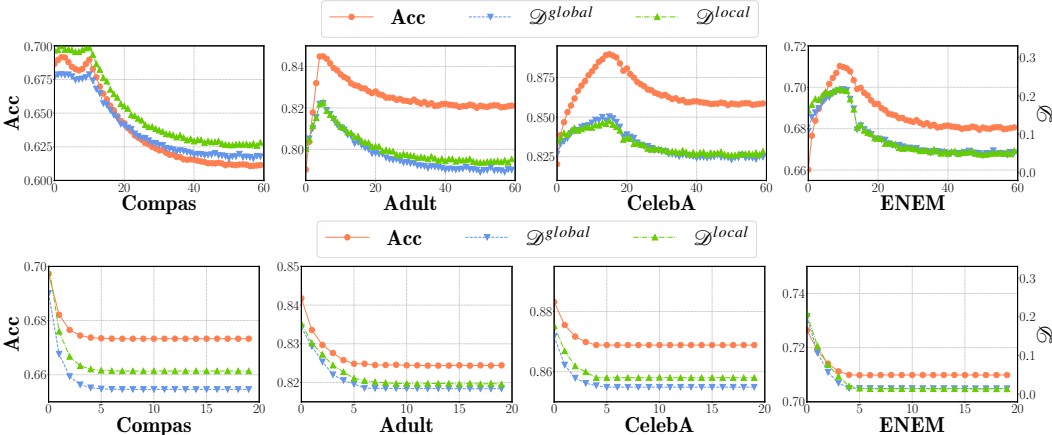

Figure 4: Communication Effectiveness Analysis. The convergence rates of both the in-processing (top row) and post-processing (bottom row) methods with respect to communication rounds on Compas, Adult, CelebA, and ENEM datasets.

Overall, whether employing the in-processing or post-processing method, all three performance metrics rapidly converge to stable values across each of the four datasets, empirically confirming both the communication efficiency and the overall effectiveness of FedFACT.

**Scalability.** We evaluate FedFACT's performance as the number of clients varies from 2 to 50 on all four datasets, with heterogeneity parameter $\gamma = 5$ to ensure that each local client has adequate samples for assessing local fairness. The results, shown in Figure 5, indicate that on each dataset, there is an upward shift in the metric as the client count increases. Enforcing fairness constraints, especially via the in-processing method, sometimes necessitates a modest loss in accuracy, and the post-processing approach on the Compas dataset exhibits pronounced fairness fluctuations due to substantial generalization error when sample sizes are small. Aside from this, our method reliably bounds the model's fairness, underscoring its robustness to variations in client population.

# E    Broader Impacts and Limitations

**Broader Impacts.** This paper addresses critical fairness issues in FL. By embedding fairness constraints at both the global and client levels, our framework delivers models that distribute accuracy more equitably, bolstering user confidence and mitigating bias amplification. The contributions of this research enhance user satisfaction and promote social equity. This fairness-aware approach extends readily to high-stakes classification tasks beyond FL: for instance, clinical decision support in hospital networks, vision-based detection systems, and financial fraud alerts. Integrating fairness into decentralized model training promotes privacy-preserving, equitable AI, helps satisfy emerging regulatory requirements, and encourages broader adoption of responsible machine learning across diverse application domains.

**Limitations.** The primary limitation of FedFACT is the fairness representation, which contains the linear disparities such as communly used DP, EOP and EOP criteria, but it excludes some nonlinear formulations of fairness, e.g. Predictive Parity [22] and individual fairness [29]. Moreover, based on our generalization-error analysis, although the proposed method enables a controllable accuracy-fairness trade-off for a given fairness metric, it still requires a sufficiently large local sample size to accurately estimate local fairness (whereas global fairness demands only an adequate overall sample size). While our empirical results compare favorably against existing approaches, exploiting dataset characteristics to optimize fairness may reduce the sample complexity needed for local fairness optimization. Addressing these limitations remains an important avenue for future work.

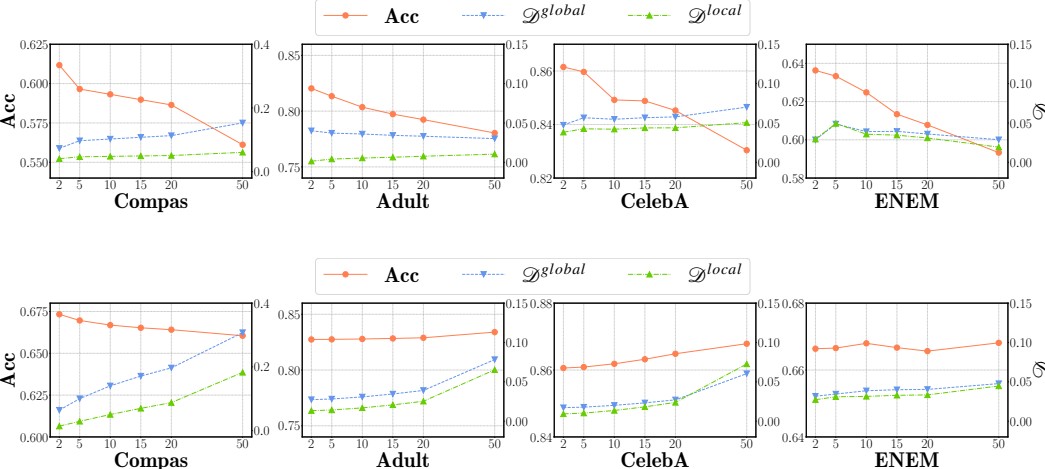

Figure 5: Scalability Analysis. The behavior of both the in-processing (top row) and post-processing (bottom row) methods as the number of clients increases from 2 to 50 across Compas, Adult, CelebA, and ENEM datasets.

