# OpenReview forum: "FedFACT: A Provable Framework for Controllable Group-Fairness Calibration in Federated Learning"
_NeurIPS.cc/2025/Conference — NeurIPS 2025 poster_

### Official Review · Reviewer_vZsk · 2025-06-06

**Clarity:** 2
**Significance:** 3
**Originality:** 3
**Rating:** 4
**Confidence:** 2

**Summary:**

The paper FedFACT jointly enforces global and local group fairness in multi-class classification settings. It introduces both in-processing and post-processing methods that approximate Bayes-optimal fair classifiers by reformulating the fairness-constrained learning problem into personalized cost-sensitive classification and bi-level optimization. The authors support their proposed algorithm with both theoretical analysis and empirical evaluation.

**Questions:**

1: The paper is difficult to follow, particularly in Section 4.3, due to the heavy use of subscript and superscript notation. With clearer explanations and better organization of the derivations and formulations, this section would be significantly easier to understand.

2: The in-processing approach produces personalized classifiers for each client, while the post-processing method enforces fairness constraints. However, in the experiments, the authors evaluate FedFACT (In) and FedFACT (Post) separately. Why not consider a combined approach—FedFACT (In & Post)? Would this integration potentially yield better performance than using either method alone? If not, what are the limitations or trade-offs that prevent such a combination?

**Ethical Concerns:**

["NO or VERY MINOR ethics concerns only"]

**Final Justification:**

The authors have thoroughly addressed my concerns in the rebuttal. Therefore, I am raising my score to 4.

**Quality:**

3

**Strengths And Weaknesses:**

Strengths:

1: The paper provides a unified framework for addressing both global and local group fairness in multi-class federated learning.

2: FedFACT approximates the Bayes-optimal fair classifier in both the in-processing and post-processing stages by reformulating the problem as a personalized cost-sensitive learning task and a bi-level optimization, respectively.

3: The authors offer a theoretical analysis to support the design and effectiveness of the proposed FedFACT algorithm.

Weakness:

See the Questions section

---

> ### Author Rebuttal · Authors · 2025-07-31
>
> ### For Questions
> > **Q1:** The paper is difficult to follow, particularly in Section 4.3, due to the heavy use of subscript and superscript notation.With clearer explanations and better organization of the derivations and formulations, this section would be significantly easier to understand.
>
> **Response:** Thank you for thoroughly reviewing our paper and for pointing out the heavy use of notation. We apologize for the density of notation, since the inherent complexity of the multi‑class fairness problem necessitates a certain level of notational detail. Meanwhile, we fully agree with your feedback and will improve the notation and explanations in the following three aspects to enhance the paper’s readability and organization.
>
> **(i) Notation Table.** To improve clarity, we plan to introduce the following notation table, which summarizes **the main symbols used throughout the paper**, particularly in Section 4 (Methodology), along with their descriptions. Note that, to avoid redundancy of including all symbol definitions, we have limited this table to the key symbols that appear frequently in the paper. Readers can refer to it as needed to quickly review the meaning of these notations utilized in the subsequent derivations.
>
> | Notation | Description | Notation | Description |
> | ------ | ------ | ------ | ------ |
> | $(X,A,Y,K)$ | Joint Distribution (Data, Group, Label, Client) | $h$ | Random Classifier |
> | $(\mathcal{X},\mathcal{A},\mathcal{Y},\mathcal{K})$ | Corresponding Sample Space | $\mathcal{H}$ | Set of Random Classifiers |
> | $N$ | Number of Client | $p_\delta$ | Probability of Event $\delta$ (Detailed in Appendix 1.1) |
> | $n_k$ | Local Sample Size | $\Delta_m$ | Probability Simplex |
> | $\mathcal{D}_k$ | Local Dataset $\{(x_{k,i}, a_{k,i}, y_{k,i})\}_{i=1}^{n_k}$ | $\widehat{Y}$ | Model Prediction $h(x)$ |
> | $\eta$ | Bayes score function | $\mathbf{C^{a,k}}(h)$ | Decentralized Group-Specific Confusion Matrix |
> | $\mathbf{D}^{a,k}_{u_g}$ | Global Fairness Constraint Matrix (${u_g}\in \mathcal{U}_g$) | $\mathbf{D}^{a,k}_{u_k}$ | Local Fairness Constraint Matrix (${u_k}\in \mathcal{U}_k$)|
> | $\mathscr{D}^{g}_{u_g}(h)$ | Global Fairness Function $\sum_{a,k} \langle \mathbf{D}^{a,k}_{u_g}, \mathbf{C}^{a,k}(h) \rangle$ | $\mathscr{D}_{u_k}^k(h)$ | Local Fairness Function $\sum_{a,k} \langle \mathbf{D}^{a,k}_{u_k}, \mathbf{C}^{a,k}(h) \rangle$ |
> | $\mathcal{U}_g$ | Index Set for Global Fairness Constraints | $\mathcal{U}_k$ | Index Set for Local Fairness Constraints |
> | $\xi^{g}$ | Global Fairness Bounds | $\xi^k$ | Local Fairness Bounds ($k \in [N]$) |
> | $\lambda$ | Global Dual Parameter $(\lambda^{(1)},\lambda^{(2)})$ | $\mu$ | Local Dual Parameter $\bigcup_{k=1}^N \mu_k, \ \mu_k=(\mu^{(1)}_k,\mu^{(2)}_k)$ |
> | $\Lambda$ | Value Domain of Global Dual Parameter | $\mathcal{M}$ | Value Domain of Local Dual Parameter |
> | $\mathbf{h}$ | Federated Model $(h_1,\cdots,h_N)$ | $\mathcal{L}(\mathbf{h},\lambda,\mu)$ | Lagrangian Dual Function |
> | $\Pi_{\mathcal{X}}(x)$ | Projection Operator | $\widehat{\mathcal{L}}(\mathbf{h},\lambda,\mu)$ | Empirical Lagrangian Dual Function |
> | $\mathbf{M}^{\lambda,\mu}(a,k)$ | Fairness Calibration Matrix | $\mathcal{P}^X_{a,k}$ | Probability Measure $P(X \mid A=a, K=k)$ |
>
> Note that we improve the subscript or superscript $(l,k)$ to $k$ for more explicit formulation.
>
> **(ii) Clearer Presentation of the Formulas.** To enhance clarity, we will further make several modifications to the presentation of the formulas. First, we will revise certain notations to enhance readability, ensuring that the symbols are more intuitive and consistent throughout the paper (e.g.,   $\mathbf{M}^{\lambda,\mu}(a,k) \to \mathbf{M}^{a,k}(\lambda,\mu)$ and the notation $(l,k)$ mentioned above). Some typos in Algorithm 2 will be corrected. Moreover, we also find that some of the more complex equations can be simplified. For example, the lengthy equation (7) can be simplified by using 1-norm $\| \lambda\|_1$ and $\| \mu_k\|_1$.
>
> **(iii) Intuitive Explanation:** We will insert an concise, intuitive overview of the theoretical results in Section 4.3 to guide readers in understanding the subsequent analysis: The optimal classifier under fairness constraints from a probabilistic perspective (the Bayes-optimal fair classifier [1,2,3])  in post-processing is derived in Theorem 5. We then propose a practical, unbiased objective in equation (8) to learn this optimal classifier. Proposition 6 and Theorem 10 in Appendix B.5 provide the theoretical evidence that the post‑processing method will approximate the optimal accuracy‑fairness trade-off as the number of iterations and sample size increase.
>
> > **Q2:** The in-processing approach produces personalized classifiers for each client, while the post-processing method enforces fairness constraints. However, in the experiments, the authors evaluate FedFACT (In) and FedFACT (Post) separately. Why not consider a combined approach—FedFACT (In & Post)? Would this integration potentially yield better performance than using either method alone? If not, what are the limitations or trade-offs that prevent such a combination?
>
> **Response:** We appreciate your attention to the details of our experimental section and your valuable question. In fact, combining in‑processing and post‑processing is theoretically unsound. In the following, we clarify the inherent theoretical limitations and support our claim through empirical results. We hope this resolves your concerns.
>
> **(i) Theoretical Unsoundness.** Combining these two methods can only be achieved by **using the classifier learned from the in‑processing phase as the pre‑trained model for the post‑processing method**, which then performs further fairness calibration. Nevertheless, this approach is theoretically unjustifiable. Note that **post‑processing requires the pre‑trained classifier to approximate the ground‑truth Bayes score function** [1,2,3,4]. A widely adopted approach in fairness post‑processing is to train probabilistic classifiers aimed at maximizing accuracy without fairness calibration, typically using cross‑entropy loss,  and to treat the model’s probabilistic output as an estimate of the Bayes score function. While **the in‑processing method is designed to learn an optimal classifier that fulfills fairness constraints [5,6], it does not aim to learn a probabilistic classifier that accurately approximates the Bayes score function.** Intuitively, the objective of the in‑processing method is to approximate Equation (3) in our paper, rather than the ground‑truth Bayes score function $\eta(X,A,K)$. Consequently, integrating the two approaches does not improve the model’s performance; instead, it magnifies the errors introduced by the pre‑trained model, leading to a reduction in performance.
>
> **(ii) Experimental Evaluation.** We conducted comparative experiments to provide empirical support for the aforementioned conclusion. The experiment is conducted on all four datasets, using a Dirichlet partition $\gamma=0.5$ to construct the heterogeneous data. We set $(\xi^g,\xi^l)=(0.05,0.05)$ for in‑processing and$(\xi^g,\xi^l)=(0.01,0.01)$ for post‑processing to construct FedFACT(In & Post)$_1$ and $(\xi^g,\xi^l)=(0.01,0.01)$ for both methods to construct FedFACT(In & Post)$_2$, with all other settings identical to those in the original paper.
>
>
> |Dataset|Method|Acc|$\mathscr{D}^{global}$|$\mathscr{D}^{local}$|
> |---|---|---|---|---|
> |**Compas**|FedFACT(In)|61.17|0.0407|0.0732|
> | |FedFACT(In & Post)$_1$|62.08|0.0334|0.0719|
> | |FedFACT(In & Post)$_2$|60.81|0.0275|0.0613*|
> | |FedFACT(Post)|67.33*|0.0139*|0.0641|
> |**Adult**|FedFACT(In)|82.04|0.0014|0.0401|
> | |FedFACT(In & Post)$_1$|82.32|0.0205|0.0386|
> | |FedFACT(In & Post)$_2$|81.72|0.0088*|0.0331|
> | |FedFACT(Post)|82.74*|0.0134|0.0274*|
> |**CelebA**|FedFACT(In)|86.15|0.0382|0.0473|
> | |FedFACT(In & Post)$_1$|86.74|0.0297|0.0363|
> | |FedFACT(In & Post)$_2$|85.80|0.0242|0.0267|
> | |FedFACT(Post)|87.06*|0.0093*|0.0172*|
> |**ENEM**|FedFACT(In)|68.33|0.0493|0.0487|
> | |FedFACT(In & Post)$_1$|68.41|0.0252|0.0206|
> | |FedFACT(In & Post)$_2$|67.98|0.0127|0.0213|
> | |FedFACT(Post)|71.21*|0.0056*|0.0198*|
>
> The results indicate that while combining post‑processing and in‑processing can improve fairness at the cost of accuracy, the accuracy-fairness balance of FedFACT (In & Post) is significantly inferior to FedFACT (Post) in most cases. Additionally, on the Adult dataset, the global fairness metric of the in‑processing method initially outperforms the post‑processing method, but when combined with post‑processing, global fairness is weakened. These findings provide additional support for the conclusions of our previous theoretical analysis.
>
> We sincerely appreciate your valuable questions. The modifications, discussions, and experiments outlined above will be incorporated into the revised version of the paper.
>
> [1] Post-hoc Bias Scoring Is Optimal For Fair Classification, ICLR 2024.
>
> [2] Fairness Guarantees in Multi-class Classiﬁcation with Demographic Parity, JMLR 2024.
>
> [3] Fair and Optimal Classification via Post-Processing, ICML 2023.
>
> [4] Fair Bayes-Optimal Classifiers Under Predictive Parity, NeurIPS 2022.
>
> [5] A Reductions Approach to Fair Classification, ICML 2018.
>
> [6] Fair Bilevel Neural Network (FairBiNN): On Balancing fairness and accuracy via Stackelberg Equilibrium, NeurIPS 2024.

---

> > ### Comment · Reviewer_vZsk · 2025-08-05
> >
> > Thank you for the detailed responses. The authors have thoroughly addressed my concerns in the rebuttal. Therefore, I am raising my score to 4.

---

> > > ### Author Response · Authors · 2025-08-07
> > >
> > > Dear reviewer vZsk,
> > >
> > > Thank you for your positive final comment and for raising your score. We are very grateful for your support.
> > >
> > > Your insightful feedback throughout the review process is quite valuable in helping us strengthen the paper. As you suggested, we will be sure to integrate the discussion and experiments from our responses into the final manuscript. We will further improve the notation used in this paper, simplify the complex formulas, and present our theoretical results and their implications more clearly.
> > >
> > > Thank you once again for your constructive feedback.

---

### Official Review · Reviewer_facd · 2025-06-27

**Clarity:** 4
**Significance:** 3
**Originality:** 4
**Rating:** 5
**Confidence:** 1

**Summary:**

This paper introduces FedFACT, a novel framework for achieving controllable group fairness in federated learning under multi-class classification. Addressing key challenges—harmonizing global and local fairness, and enabling an optimal trade-off between fairness and accuracy—FedFACT identifies Bayes-optimal classifiers under fairness constraints. It reformulates fair FL as a personalized cost-sensitive learning problem (for in-processing) and as a bi-level optimization problem (for post-processing). The authors provide theoretical convergence and generalization guarantees, and experimental results show that FedFACT outperforms existing methods in maintaining both accuracy and fairness across heterogeneous datasets.

**Questions:**

1) Integration of in-processing and post-processing methods: While the paper evaluates the in-processing and post-processing algorithms independently, the discussion suggests that they are complementary. However, it remains unclear whether and how these two approaches can be effectively combined. Could the authors clarify if a combined approach is feasible, and if so, provide empirical evidence or discussion on its performance? Demonstrating synergy between the two could strengthen the practical utility of the framework.


2) Scalability to a larger number of clients: The current evaluation is conducted with a very limited number of clients (2 for COMPAS and 5 for other datasets). This raises concerns about the generalizability of the proposed approach to more realistic federated learning scenarios involving dozens or hundreds of clients. Could the authors provide insights into:

* How the method scales as the number of clients increases?

* Whether the fairness-accuracy trade-off holds under larger, more heterogeneous client populations?

* Any plans or limitations in scaling up the evaluation?

3) Sufficiency of evaluation samples: Relatedly, the paper does not clearly justify whether the current number of clients and data samples per client is sufficient for meaningful statistical evaluation. Could the authors elaborate on the sampling criteria and suggest whether additional clients or data partitioning strategies might yield more reliable insights?

**Ethical Concerns:**

["NO or VERY MINOR ethics concerns only"]

**Final Justification:**

I thank the authors for their thorough rebuttal, which clarifies design choices, provides large-scale ACSIncome and 100-client results, and details parameter tuning. The explanation of integrating in-processing and post-processing methods is clear and convincing. Overall, the responses reinforce the robustness, scalability, and soundness of the approach, and I have no remaining concerns regarding the work.

**Limitations:**

yes

**Quality:**

4

**Strengths And Weaknesses:**

Strengths

* Strong empirical performance: The proposed FedFACT framework demonstrates outstanding empirical results, consistently outperforming baseline methods in both accuracy and fairness trade-offs across a variety of experimental settings. The ability to balance global and local fairness without significant performance degradation is a notable contribution.


* Theoretical grounding: The paper provides solid theoretical guarantees, including convergence and generalization bounds for the proposed framework, which adds rigor and credibility to the approach.


* Novel formulation: Reformulating fair federated learning as a cost-sensitive learning problem and introducing a bi-level optimization framework is an original and intellectually interesting contribution. This provides a principled path for balancing fairness and accuracy.

Weaknesses

* Limited evaluation scale: The experimental evaluation appears to be conducted on relatively small datasets or a limited number of clients. This raises concerns about the scalability and generalizability of the approach to large, heterogeneous real-world deployments.


* Practical usability and parameter tuning: The framework relies on tuning fairness relaxation parameters (ξg ,ξl ), but the paper does not provide a systematic or automated way to set these values. As a result, achieving the desired trade-off between accuracy, global fairness, and local fairness appears to require manual trial-and-error. This could hinder the practical adoption of FedFACT in real-world applications as such tuning can be tedious.

---

> ### Author Rebuttal · Authors · 2025-07-31
>
> ### For Weaknesses
>
> > **W1:** Limited evaluation scale.
>
> **Response:** We appreciate your attention to the details of our experimental section and your valuable question. We answer your question in the following two aspects.
>
> **(i) Scale of data:** We apologize for any misunderstanding. We evaluate our method on four available real-world datasets, with sizes ranging from approximately 6K to over 100K. Notably, both CelebA and ENEM comprise over 200K samples, and according to existing benchmark studies [1], they are almost the largest publicly available datasets in the fairness domain in terms of sample size.
>
> **(ii) Large, heterogeneous real-world deployments.** To demonstrate the real-world deployment of our method, we conduct experiments on the **ACSIncome dataset** [2] with 1,664,500 datapoints, divided into **50 clients representing U.S. states**, using race (white/non-white) as the sensitive attribute. The fairness metric is demographic parity (DP), with $\xi^g=\xi^l=0.01$ for both algorithms.
>
> |Method|Acc|$\mathscr{D}^{global}$|$\mathscr{D}^{local}$|
> |---|---|---|---|
> |**FedAvg**|79.80|0.1032|0.1857|
> |FairFed|75.22|0.0883|0.1460|
> |FedFB|76.53*|0.0676|0.1148|
> |FCFL|71.95|0.0651|0.0832|
> |praFFed|74.48|0.0733|0.0914|
> |Cost|75.40|0.0290|0.0642|
> | FedFACT$_{g l}$ (In)|76.39|0.0232|0.0360|
> | FedFACT$_{g l}$ (Post)|76.50|0.0108*|0.0249*|
>
> The results show that FedFACT significantly improves the accuracy‑fairness trade‑off compared to other methods in real‑world deployments.
>
> > **W2:**  Practical usability and parameter tuning.
>
> **Response:** Thank you for your valuable question on parameter tuning. **The parameters $(\xi^g, \xi^l)$ control the tolerance for global and local disparity, and should be set based on real-world needs.** Our experiments show that tuning these parameters effectively maintains fairness within the desired thresholds, particularly when using the post-processing method.
>
> Furthermore, we provide a more fine‑grained parameter tuning strategy:
>
> (1) To minimize global or local group‑wise disparity, one can set the parameters $(\xi^g, \xi^l)$ to values close to zero (e.g., 0.01 or 0.001).
>
> (2) Considering that there exist practical fairness tolerance $(\kappa^g, \kappa^l)$. First, initialize $(\xi^g, \xi^l)$ to near‑zero values $\rho$. After running the algorithm, the observed fairness gap $(Gap^g, Gap^l):=(\mathscr{D}^g-\rho, \mathscr{D}^l-\rho)$ predominantly reflects the generalization error due to finite sample size, and independent of the settings of $(\xi^g, \xi^l)$ according to our generalization error analysis (Theorems 9 and 10 in Appendix). One can set $(\xi^g, \xi^l) = \big(\max(\rho, \kappa^g-Gap^g), \max(\rho, \kappa^l-Gap^l)\big)$ to meet real‑world fairness tolerance.
>
> ### For Questions
>
> > **Q1:** Integration of in-processing and post-processing methods.
>
> **Response:** We appreciate your attention to the details of our experimental section and your valuable question. We answer your question in the following two aspects.
>
> **(i) Theoretical Unsoundness.** Combining these two methods can only be achieved by using the in‑processing classifier as the pre‑trained model for post‑processing. However, this is theoretically unjustifiable. Note that post‑processing requires the pre‑trained classifier to approximate the ground‑truth Bayes score function [3,4]. While the in‑processing method is designed to learn a group-fairness calibrated classifier [5,6], without aiming to learn the Bayes score function. Consequently, integrating the two approaches does not enhance performance; instead, it magnifies the errors introduced by the pre‑trained model, leading to performance decline.
>
> **(ii) Experimental Evaluation.** We conducted comparison experiments to provide empirical support. We set $(\xi^g,\xi^l)=(0.05,0.05)$ for in‑processing and $(\xi^g,\xi^l)=(0.01,0.01)$ for post‑processing to construct FedFACT(In & Post)$_1$ and $(\xi^g,\xi^l)=(0.01,0.01)$ for both methods to construct the method FedFACT(In & Post)$_2$, with $\gamma=0.5$.
>
> |Dataset|Method|Acc|$\mathscr{D}^{global}$|$\mathscr{D}^{local}$|
> |---|---|---|---|---|
> |**Compas**|FedFACT(In)|61.17|0.0407|0.0732|
> | |FedFACT(In & Post)$_1$|62.08|0.0334|0.0719|
> | |FedFACT(In & Post)$_2$|60.81|0.0275|0.0613*|
> | |FedFACT(Post)|67.33*|0.0139*|0.0641|
> |**Adult**|FedFACT(In)|82.04|0.0014|0.0401|
> | |FedFACT(In & Post)$_1$|82.32|0.0205|0.0386|
> | |FedFACT(In & Post)$_2$|81.72|0.0088*|0.0331|
> | |FedFACT(Post)|82.74*|0.0134|0.0274*|
> |**CelebA**|FedFACT(In)|86.15|0.0382|0.0473|
> | |FedFACT(In & Post)$_1$|86.74|0.0297|0.0363|
> | |FedFACT(In & Post)$_2$|85.80|0.0242|0.0267|
> | |FedFACT(Post)|87.06*|0.0093*|0.0172*|
> |**ENEM**|FedFACT(In)|68.33|0.0493|0.0487|
> | |FedFACT(In & Post)$_1$|68.41|0.0252|0.0206|
> | |FedFACT(In & Post)$_2$|67.98|0.0127|0.0213|
> | |FedFACT(Post)|71.21*|0.0056*|0.0198*|
>
> The results indicate that while combining post‑processing and in‑processing, the accuracy-fairness balance of FedFACT (In & Post) is significantly inferior to FedFACT (In) or (Post) in most cases.
>
> > **Q2:** Scalability to a larger number of clients.
>
> **Response:** We appreciate your thorough review and the valuable question regarding scalability. We fully acknowledge your feedback. Due to space limitations, **detailed evaluations of model stability in terms of global fairness, local fairness, and accuracy with 2 to 50 clients are provided in Appendix D.5.** To strengthen our experiments, we have conducted additional tests on four datasets here, scaling up to 100 clients.
> - **Limitations in scaling up the evaluation.** As data heterogeneity and client number increase, local datasets exhibit pronounced group or label skewness and vacancies, such as missing multiple groups or classes, making local fairness evaluation infeasible.
> Therefore, to scale our evaluation appropriately, we conduct additional experiments focusing on the stability of accuracy and global fairness with client numbers ranging from 50 to 100, extending the experiments in Appendix D.5.
>
> | |Dataset|Compas| |Adult| |CelebA| |ENEM| |
> |---|---|---|---|---|---|---|---|---|---|
> |**Clients**|**Method**|**Acc**|$\mathscr{D}^{global}$|**Acc**|$\mathscr{D}^{global}$|**Acc**|$\mathscr{D}^{global}$|**Acc**|$\mathscr{D}^{global}$|
> |**50**|FedFACT(In)|60.33|0.0704|80.04|0.0104|84.53|0.0438|62.72|0.0607|
> |**50**|FedFACT(Post)|65.81|0.0447|81.40|0.0429|86.06|0.0338|66.05|0.0503|
> |**75**|FedFACT(In)|59.28|0.0848|79.65| 0.0189|82.76|0.0652|61.53| 0.0742|
> |**75**|FedFACT(Post)|64.33|0.0610|80.35|0.0572|84.34|0.0473|65.42|0.0588|
> |**100**|FedFACT(In)|56.66|0.1017|78.10| 0.0428|81.68|0.0789|60.05|0.0823|
> |**100**|FedFACT(Post)|63.85|0.0712|79.8|0.0812|83.06|0.0650|65.29|0.0877|
>
> **Method performance and accuracy-fairness trade-off.**  Overall, the results show that while the model’s performance declines slightly with an increase in the number of clients, it remains stable in terms of fairness, demonstrating its robustness. Specifically, although the model's accuracy is inevitably impacted as the number of clients grows, the global fairness metric does not experience a sharp decline. This indicates that the model can maintain a balance between accuracy and fairness even with a large number of clients, highlighting the scalability of our approach with respect to client number.
>
> > **Q3:** Sufficiency of evaluation samples.
> **Response:** Thank you for your attention to the details of our experimental section and your valuable question. We answer your question in the following three aspects.
>
> **(i) The sampling criteria and Data partitioning strategies.** Our experiments utilize sampling schemes based on two data-partitioning methods.(1) **Dirichlet partition** [7], where a proportion vector $\alpha$ is generated from a Dirichlet distribution $Dir(\gamma)$, and samples are allocated to clients based on these proportions; (2) **Heterogeneous correlation split**, a novel method that introduces random heterogeneous correlations between $A$ and $Y$ as detailed in Appendix D.3. We employ these methods to introduce heterogeneity and decouple global and local fairness, since significant differences can be observed only under a certain degree of group and label heterogeneity, as shown in [8]. Thus, our experiments explore the accuracy-fairness balance under such conditions, ensuring representativeness.
>
> **(ii) Client number:** We apologize for any misunderstanding. We restricted our comparative experiments to 5 clients in order to ensure that each client retains enough samples from every subgroup to reliably assess local fairness under heterogeneous data partitions. This is consistent with prior federated learning fairness research [7,9,10], which generally conducts comparisons with a limited number of clients (less than 4). Furthermore, In the scalability study (Appendix D.5), we have conducted detailed experiments on the impact of the number of clients on the model.
>
> We sincerely appreciate your valuable questions. The discussions and experiments mentioned above will be incorporated into the revised version of the paper.
>
> [1] FFB: A Fair Fairness Benchmark for In-Processing Group Fairness Methods, ICLR 2024.
>
> [2] Retiring Adult: New Datasets for Fair Machine Learning, NeurIPS 2021.
>
> [3] Post-hoc Bias Scoring Is Optimal For Fair Classification, ICLR 2024.
>
> [4] Fairness Guarantees in Multi-class Classiﬁcation with Demographic Parity, JMLR 2024.
>
> [5] A Reductions Approach to Fair Classification, ICML 2018.
>
> [6] Fair Bilevel Neural Network (FairBiNN): On Balancing fairness and accuracy via Stackelberg Equilibrium, NeurIPS 2024.
>
> [7] FairFed: Enabling Group Fairness in Federated Learning, AAAI 2023.
>
> [8] Demystifying Local and Global Fairness Trade-offs in Federated Learning Using Partial Information Decomposition, ICLR 2024.
>
> [9] Addressing Algorithmic Disparity and Performance Inconsistency in Federated Learning, NeurIPS 2021.
>
> [10] PraFFL: A Preference-Aware Scheme in Fair Federated Learning, KDD 2025.

---

> > ### Comment · Reviewer_facd · 2025-08-05
> >
> > Thank you for your detailed responses. They addressed my questions well.

---

> > > ### Author Response · Authors · 2025-08-07
> > >
> > > Dear reviewer facd,
> > >
> > > Thank you very much for your valuable feedback. It has been tremendously helpful in improving the quality of our work, and the above discussions and additional experimental results about real-world deployment, client number and results on integration of in-processing and post-processing methods will be incorporated into the revised manuscript.

---

### Official Review · Reviewer_LAAJ · 2025-07-03

**Clarity:** 2
**Significance:** 3
**Originality:** 2
**Rating:** 4
**Confidence:** 2

**Summary:**

The paper introduces FedFACT, a unified framework for controllable group fairness calibration in federated learning. The proposed method optimizes both global and local level fairness for multi-class problems. They propose two algorithms, in-processing and post-processing based methods, that approach Bayes-optimal fair classifiers. Both algorithms come with provable convergence and consistency guarantees. The authors compare the proposed algorithms with existing baselines that are targeted for only global or local fairness. The numerical results on various datasets show that the proposed methods enjoy a better tradeoff than existing methods between global-local fairness and accuracy.

**Questions:**

Please see weaknesses. Beyond that:
- How would the proposed methods behave if some clients are adversarial?

**Ethical Concerns:**

["NO or VERY MINOR ethics concerns only"]

**Final Justification:**

I think that the authors clarified all the questions raised by me. They explained how the proposed method can be extended for fairness notions that are not linear in the covariance matrix. They presented new results under adversarial clients and how to mitigate their effects.

**Limitations:**

yes

**Paper Formatting Concerns:**

No major formatting issues

**Quality:**

3

**Strengths And Weaknesses:**

Strengths
- To my knowledge, the paper is the first joint consideration of global and local fairness in federated learning. It is a novel contribution.
- The proposed methods are practical, each communication round requires only one client-server communication, and the post-processing method doesn't require labels.
- The experiments section is quite comprehensive, authors consider multiple datasets, baselines, and data split methods.

Weaknesses
- The framework supports only fairness constraints that are linear in the confusion matrix (DP, EO, EOP); important nonlinear notions such as predictive parity are out of scope. Can the confusion-matrix formulation be extended to ratios or conditional metrics such as predictive parity via a surrogate function that is linear in terms of the confusion matrix?

---

> ### Author Rebuttal · Authors · 2025-07-31
>
> ### For Weaknesses
>
> > **W1:** The framework supports only fairness constraints that are linear in the confusion matrix (DP, EO, EOP); important nonlinear notions such as predictive parity are out of scope. Can the confusion-matrix formulation be extended to ratios or conditional metrics such as predictive parity via a surrogate function that is linear in terms of the confusion matrix?
>
>
> **Response:** Thank you for your detailed attention to the scope of our framework and for providing such valuable feedback. We fully agree with your viewpoint and endeavour to broaden our scope in the following two aspects, while evaluating the feasibility of covering predictive parity to address your concerns.
>
> **(i) Ratios Metrics.** We indeed find that **fairness notions formulated as ratios metrics can be converted into linear constraints under certain conditions**, and our framework is well suited to enforce these fairness constraints in federated learning environments. Specifically, the ratios metric or constraints, which are formulated as $\left|\frac{\sum_a\langle \mathbf{D}^a, \mathbf{C}^{a}(h) \rangle}{\sum_a\langle \mathbf{G}^a, \mathbf{C}^{a}(h) \rangle}\right| \leq \xi$ with constant metrix $\mathbf{D}^a, \mathbf{G}^a$ and group-specific confusion matrix $\mathbf{C}^{a}(h)$ depend on classifier $h$, can certainly be transformed into multiple linear constraints if the sign of $\sum_a\langle \mathbf{G}^a, \mathbf{C}^{a}(h) \rangle$ is unchanged for any $h$. When the denominator’s sign is uncertain, the feasible domain of $\mathbf{C}^{a}(h)$ is non‑convex, precluding its expression via linear constraints. In fact, since each entry of $\mathbf{C}^{a}(h)$ lies in [0,1], whenever the entries of $ \mathbf{G}^a$ are sign‑consistent, the corresponding ratio constraint admits a linear‑constraint representation. For example, the **Calibration within Groups (CG)** which was proposed in [1] and further explored in [2], is a fairness metric in binary classification and can be formulated as $\frac{FN^a}{FN^a + TN^a} = v_0; \frac{TP^a}{TP^a + FP^a} = v_1$, where $TP^a, FP^a, FN^a, TN^a$ are derived from binary group-specific confusion matrix $\mathbf{C}^{a}$ and $0 ≤ v_0 < v_1 ≤ 1$ have no implicit dependence on any entries of the fairness-confusion tensor. Because this fairness criterion appears as a ratio metric and every element of corresponding $\mathbf{G}^a$ is non‑negative, it admits a linear‑constraint representation and can be realized in our proposed distributed framework. Moreover, the ratio metrics presented in [3] also can be formulated into multiple linear constraints based on the above analysis.
>
> **(ii) Predictive Parity.** In binary classification, Predictive Parity [4] is a typical conditional metric states that the likelihood of being in the positive class given the positive prediction is the same for each group: $|P(Y = 1|\widehat{Y} = 1, A = 1) - P(Y = 1|\widehat{Y} = 1)| \leq \xi$. Prior work [2] explored this form of constraint, demonstrating that **this notion of fairness is not linear in the group-specific confusion matrix** and presenting a quadratic-form representation based on the group-specific confusion matrix. Additionally, the other research [5] showed that the Bayes-optimal classifier under Predictive Parity constraint even cannot be achieved by threshold calibration. In multi-class classification, we did not find any existing literature that defines a multi‑class formulation of Predictive Parity. A possible natural extension could be expressed as $\max_{y,a}|P(Y = y|\widehat{Y} = y, A = a) - P(Y = y|\widehat{Y} = y)| \leq \xi$. Following the same derivation in [2], it is obvious that Predictive Parity also cannot be expressed in multiple linear or ratio constraints under multi-class circumstances.
>
> Overall, certain ratio metrics can be reformulated as linear constraints and enforced within our framework. However, existing research indicates that conditional metrics like predictive parity cannot be represented in the linear form of group-specific confusion matrix.
>
> ### For Questions
>
> > **Q1:** How would the proposed methods behave if some clients are adversarial?
>
> **Response:** Thank you for raising this important question regarding the behavior of our method under adversarial clients. We respectfully note that this paper emphasizes group-fairness calibration in federated learning and proposes a general framework that achieves accuracy-fairness balance. Meanwhile, we acknowledge that the presence of adversarial clients can interfere with the achievement of fairness in federated learning. Thus, we will investigate the presence of adversarial clients in two respects and provide a detailed response to your question.
>
> **(i) Applicability of robust aggregation methods:** we aim to implement group-fairness calibration and propose a general framework that leverages cost-sensitive loss and personalized optimization to achieve accuracy-fairness balance, without relying on specialized aggregation schemes. While FedAvg is employed here due to its straightforward implementation and effective performance under data heterogeneity, adversarial-resilient aggregation techniques (e.g., $(f, \kappa)$-robust aggregator [6]) can also be integrated into our framework. Moreover, the personalized procedure in our framework inherently enhances the model’s robustness [7].
>
> **(ii) Experimental evaluation with adversarial clients:** We present experimental results demonstrating our model’s performance in adversarial‑client scenarios, as well as its behavior when employing defense mechanisms such as Byzantine‑robust aggregation methods CM and Nearest Neighbor Mixing (NMM) in [6]. We partition the Compas and Adult datasets into 5 and 10 clients, respectively, under a heterogeneous environment $(\gamma=1)$, and employ **Label Flipping** as the attack method, with adversarial clients constituting 40% of the total. Results are shown in the following table. All results are the average of three repeated experiments.
>
> The model performance in the absence of adversarial clients:
>
> |Dataset|Method|Acc|$\mathscr{D}^{global}$|$\mathscr{D}^{local}$|
> |---|---|---|---|---|
> |**Compas**|FedFACT(In)|60.13|0.0589|0.0738|
> | |FedFACT(Post)|67.20|0.0344|0.0614|
> |**Adult**|FedFACT(In)|82.39|0.0180|0.0499|
> | |FedFACT(Post)|81.87|0.0191|0.0223|
>
> The model performance with adversarial clients and defense methods:
>
> |Dataset|Method|Acc|$\mathscr{D}^{global}$|$\mathscr{D}^{local}$|
> |---|---|---|---|---|
> |**Compas**|FedFACT(In)|56.33|0.1143|0.2224|
> | |FedFACT(In)+CM|58.18|0.0670|0.1032|
> | |FedFACT(In)+CM+NMM|59.25| 0.0809|0.0901|
> | |FedFACT(Post)|63.33|0.0632|0.0830|
> | |FedFACT(Post)+CM|65.56|0.0425|0.0749|
> | |FedFACT(Post)+CM+NMM|66.43|0.0399|0.0685|
> |**Adult**|FedFACT(In)|81.29|0.0855|0.1271|
> | |FedFACT(In)+CM|82.27|0.0389|0.0709|
> | |FedFACT(In)+CM+NMM|82.33|0.0217|0.0652|
> | |FedFACT(Post)|80.18|0.0393|0.0451|
> | |FedFACT(Post)+CM|80.89|0.0229|0.0346|
> | |FedFACT(Post)+CM+NMM|80.95|0.0206|0.0269|
>
> Experimental results indicate that, under adversarial‑client conditions, our model’s accuracy and fairness experience only modest declines rather than severe degradation, demonstrating that **the personalized procedure in our framework inherently enhances robustness**. When employing the Byzantine‑robust aggregation methods [6], both accuracy and fairness further recover to some extent, supporting our prior analysis.
>
> We appreciate your insightful feedback once more. The above discussions and experiments will be incorporated into the revised version of the paper.
>
> [1] Inherent trade-offs in the fair determination of risk scores, ITCS 2017.
>
> [2] FACT: A Diagnostic for Group Fairness Trade-offs, ICML 2020.
>
> [3] Beyond Adult and COMPAS: Fair Multi-Class Prediction via Information Projection, NeurIPS 2022.
>
> [4] Fair prediction with disparate impact: A study of bias in recidivism prediction instruments, Big Data 2017.
>
> [5] Fair Bayes-Optimal Classifiers Under Predictive Parity, NeurIPS 2022.
>
> [6] Fixing by Mixing: A Recipe for Optimal Byzantine ML under Heterogeneity, AISTATS 2023.
>
> [7] Ditto: Fair and Robust Federated Learning Through Personalization, ICML 2021.

---

> > ### Comment · Reviewer_LAAJ · 2025-08-04
> >
> > Thanks for the detailed rebuttal and new results. I appreciate the explanation regarding how fairness notions that are not linear in the confusion matrix could be integrated into your framework. I think that this discussion could be included in the paper. I found the new numerical results under adversarial clients promising; I value the use of robust aggregation schemes. I will maintain my positive score for this paper for now.

---

> > > ### Author Response · Authors · 2025-08-07
> > >
> > > Dear reviwer LAAJ,
> > >
> > > Thank you very much for your insightful review and for acknowledging our rebuttal. We are delighted that we were able to address your concerns. We fully agree with your important points regarding how fairness notions that are not linear in the confusion matrix could be integrated into our framework, and the impact of adversarial clients. As you suggested, we will be sure to integrate the detailed discussion on non-linear fairness notions and our new results with adversarial clients into the revised manuscript.

---

### Official Review · Reviewer_eRHZ · 2025-07-09

**Clarity:** 2
**Significance:** 2
**Originality:** 3
**Rating:** 5
**Confidence:** 3

**Summary:**

* The paper proposes a federated fair learning method that takes into account local and global fairness constraints in multi-class classification settings.
* The convergence and consistency guarantees are shown under smoothness assumptions for both in-processing and post-processing settings
* Empirical comparisons across 4 public benchmarks demonstrate better accuracy-fairness tradeoffs across both local and global constraints

**Questions:**

Typos in Table 3 (line 7) and elsewhere in the results section's writeup reduces readability of the paper

**Ethical Concerns:**

["NO or VERY MINOR ethics concerns only"]

**Final Justification:**

Update after the author response: The response has addressed my concerns, and I increase the score as it provides a sound approach at improving fairness-accuracy tradeoff in multi-class classification under federated learning settings.

**Limitations:**

Limitations around gradient conflict, and other sub-optimal optimization procedures need to be addressed beyond the Bayes-optimal fair classifier settings to bridge the gap between theoretical guarantees and practical applications.

**Quality:**

3

**Strengths And Weaknesses:**

Strengths:

* The method proposed is able to provide fine-grained sensitivity based on local and global fairness constraints of federated equality of opportunity and demographic parity based on decentralized confusion matrices, without regressing on accuracy
* The algorithm proposes to learn Bayes optimal accurate classifiers under both local and global fairness settings, which is a novel contribution
* The usage of weighted classifiers in both in-processing and post-processing settings with minimal parameter exchange in each training rounds ensures that the remaining generalization error can be mitigated by increased sampling and training rounds (T).

Weaknesses:
* The paper is very dense to read - with the implications of the theoretical results hard to parse. The bulk of the empirical results related to multi-class classification has been relegated to the Appendix. This is a bit concerning as the main claim of the paper is that it addresses global and local fairness in federated learning in multi-class settings. The difficulty of extending prior work beyond binary classification is not explained in the paper. Further, adaptations of prior work to multi-class settings are not provided or explained as to why it is difficult to obtain.
* The aggregation mechanism at the server adopts the aggregation mechanism as proposed in FedAvg, and does not discuss the challenges of gradient conflicts while optimizing for local and global fairness constraints.
* Lack of confidence intervals in the results in-general reduce the statistical significance of the results. This is particularly concerning since 2 of the 4 datasets (presented in the main paper) have small evaluation dataset sizes, which needs further investigation.

---

> ### Author Rebuttal · Authors · 2025-07-31
>
> ### For Weaknesses
>
> > **W1:** Implications of the theoretical results: relegation of most empirical results related to multi‑class classification to the Appendix; lack of explanation regarding the difficulty of extending prior work; absence of adaptations of prior work to multi‑class settings.
>
> **Response:** We greatly appreciate your careful reading and for raising these insightful feedback. We apologize for the dense presentation of our theoretical section. Below, we provide responses to each issue highlighted here.
>
> **(i) Theoretical implications.** Overall, we derive the explicit representation of the optimal probabilistic classifier under joint local-global fairness constraints and propose theoretically grounded, unbiased learning algorithms at in- & post-processing phase to learn the optimal classifier. Our convergence and generalization analyses implicate that, as the number of iterations and sample size increase, the method converges to the optimal accuracy-fairness trade‑off. To the best of our knowledge, we are the first to derive the Bayes‑optimal fair classifier in multi‑class federated learning settings and to propose both in- and post-processing optimization methods with rigorous theoretical guarantees.
>
> **(ii) The multi-class classification results.** We fully acknowledge your feedback and apologize for delaying the multi-class classification results to the Appendix due to space limitation. **In the revised version, we will move key multi-class results with a specific analysis into the main text** to ensure that the multi-class results are given adequate attention and context.
>
> **(iii) The difficulty of extending prior work beyond binary classification.** From a technical standpoint, fairness calibration techniques tailored for binary classification in federated learning could not scale to multiclass problems due to the significantly greater complexity induced by multi-dimensional model output and multi‑class fairness constraints. For in‑processing approaches, most prior works use a differentiable surrogate to approximate the fairness metrics [4,5,6], but multi‑class fairness entails multiple constraints with cross‑dependencies, making surrogate construction challenging. In post‑processing, fairness correction typically involves setting group‑specific thresholds on output scores [7,8]. However, in multi‑class scenarios, where model outputs lie on the probability simplex, a single threshold or even a simple set of thresholds cannot capture the $\arg \max$ decision boundary, thereby impeding the feasible trade‑off between fairness and accuracy.
>
>
> **(iv) Supplementary experiments.** We extend the method proposed in [8] (denoted as Cost) to the multi‑group, multi‑class scenario. Since it is designed for binary DP, EO, and EOP, we did not compare it in the original paper. We compare this method with ours in the multi‑class experimental setup (Appendix D.2), with the results  presented below.
>
> |Metric|Dataset|Method|Acc|$\mathscr{D}^{global}$|$\mathscr{D}^{local}$|
> |---|---|---|---|---|---|
> |**DP**|**CelebA**|Cost|58.28|0.0832|0.1241|
> | | |FedFACT(In)|57.32|0.0727|0.0541*|
> | | |FedFACT(Post)|60.10*|0.0337*|0.0810|
> | |**ENEM**|Cost|37.32|0.0786|0.1050|
> | | |FedFACT(In)|38.86|0.0402|0.0651*|
> | | |FedFACT(Post)|41.85*|0.0186*|0.0887|
> |**EOP**|**CelebA**|Cost|59.96|0.0814|0.1551|
> | | |FedFACT(In)|60.32|0.0430*|0.0559*|
> | | |FedFACT(Post)|60.55*|0.0637|0.1240|
> | |**ENEM**|Cost|40.62|0.0980|0.1262|
> | | |FedFACT(In)|40.82|0.0619|0.0409*|
> | | |FedFACT(Post)|41.85*|0.0480*|0.0987|
>
> The results clearly show that our model outperforms the Cost model, as our approach can approximate the optimal accuracy‑fairness boundary.
>
> > **W2:** The challenge of gradient conflicts.
>
> **Response:** Thank you for pointing out the valuable issue regarding the aggregation mechanism and the potential challenges of gradient conflicts. We discuss this challenge in two aspects:
>
> **(i) Motivation for adopting FedAvg:** In this paper, we underline group-fairness calibration and propose a general framework that leverages cost-sensitive loss and personalized optimization to achieve accuracy-fairness balance, **without relying on specialized aggregation schemes.** The FedAvg aggregation mechanism is adopted here due to its simplicity and effectiveness in heterogeneous federated learning scenarios. This allows us to focus on fairness calibration without introducing additional complexity in the aggregation process.
>
> **(ii) Discussion on gradient conflicts and additional experiments:** Gradient conflicts are ubiquitous in heterogeneous environments [10,11]. During the parameter aggregation stage of FedFACT, when each client sends gradients that push toward its own local fairness target and diverge the global fairness objective, e.g., improving local parity on one client may hurt overall parity, naive averaging can decelerate model convergence or produce sub-optimal trade‑offs. However, **it should be noted that our learning objective constitutes an unbiased estimate of the Bayes‑optimal fair classifier that balances global and local fairness, thereby mitigating the conflicts caused by optimizing global and local fairness to some extent.** Moreover, as discussed above, our model does not rely on any specialized aggregation scheme. Therefore, aggregation methods developed to mitigate gradient conflicts (e.g. [11]) can be directly adapted to our framework to further address this challenge.
>
> In conclusion, we respectfully note that, our framework emphasizes group‑fairness calibration and utilizes FedAvg for simplicity, while our proposed learning objective inherently harmonizes the conflicts caused by optimizing global and local fairness.
>
> > **W3:** Lack of confidence intervals in the results.
>
> **Response:** We appreciate your attention to the details of our experimental section and for noting the omission of confidence intervals. Space limitations led us to present only the averaged metrics of multiple experiments in the main text. We fully agree that, particularly in datasets with small data sizes, including confidence intervals is vital for demonstrating the robustness of our conclusions. To enhance the persuasiveness of our experiments and address your concern, **we further provide additional experimental details with 95% confidence interval on Compas and Adult dataset mentioned here.** Due to character limitation, we consider scenarios under the Dirichlet heterogeneous split ($\gamma=0.5$). The results are presented in the Table below.
>
> |Dataset| |Compas| | |Adult| |
> |---|---|---|---|---|---|---|
> |**Method**|**Acc**|$\mathscr{D}^{global}$|$\mathscr{D}^{local}$|**Acc**|$\mathscr{D}^{global}$|$\mathscr{D}^{local}$|
> |**FedAvg**|69.7±0.9|0.276±0.024|0.359±0.031|84.5±0.2|0.177±0.019|0.231±0.028|
> |FairFed|59.4±1.8|0.103±0.025|0.111±0.037|80.2±0.4|0.098±0.034|0.143±0.049|
> |FedFB|58.3±1.5|0.089±0.024|0.098±0.028|81.9±0.5|0.075±0.017|0.116±0.021|
> |FCFL|56.5±0.8|0.065±0.019|0.061±0.011|81.5±0.7|0.085±0.010|0.126±0.039|
> |praFFed|59.9±0.9|0.082±0.023|0.097±0.026|81.0±0.6|0.059±0.017|0.076±0.023|
> |Cost|64.5±0.4| 0.059±0.008|0.094±0.011|81.1±0.2|0.026±0.005|0.059±0.008|
> |FedFACT$_g$ (In)|61.2±0.7|0.034±0.006|0.076±0.024|82.1±0.3|0.002±0.001|0.041±0.003|
> |FedFACT$_l$ (In)|61.8±0.6|0.060±0.015|0.064±0.016| 82.4±0.4|0.014±0.004|0.051±0.005|
> | FedFACT$_{g l}$ (In)|61.2±0.7|0.041±0.008|0.073±0.022| 82.0±0.2|0.001±0.000*|0.040±0.003|
> | FedFACT$_g$ (Post)|67.3±0.3|0.013±0.003*|0.066±0.006|82.8±0.1*|0.017±0.003|0.028±0.004|
> | FedFACT$_l$ (Post) |67.5±0.4*|0.032±0.007|0.055±0.008*|82.8±0.1*|0.015±0.002|0.027±0.003*|
> | FedFACT$_{g l}$ (Post)|67.3±0.3|0.014±0.004|0.064±0.009|82.7±0.1|0.013±0.002|0.027±0.003*|
>
> These results further demonstrate that our model outperforms all baselines, achieving a better balance between accuracy and fairness.
>
> ### For Questions
> >  **Q1:** Typos in Table 3 (line 7) and elsewhere in the results section.
>
> **Response:** Thank you for your careful reading of our paper. We apologize for having these typos and we promise to correct them in the revised version. We will also review the paper in detail and revise other typos to enhance the paper’s readability.
>
> > **Limitations:** Limitations around gradient conflict, and other sub-optimal optimization procedures need to be addressed.
>
> **Response:** We sincerely appreciate your feedback. In response to Weakness 2, we discuss the challenge raised by gradient conflict. We acknowledge that certain sub‑optimal optimization procedures may impact model performance. However, **our method has demonstrated relatively significant improvements to prior methods, owing to our in‑depth theoretical investigations.** Moreover, as the proposed framework is already moderately complex, and to prevent further increases in time or memory complexity, we leave these optimization refinements for future work.
>
> Sincere appreciation for your valuable feedback. We will integrate the aforementioned analyses, discussions and additional experiments into the paper’s revised version. We hope this resolves your concerns.
>
> [1] Post-hoc Bias Scoring Is Optimal For Fair Classification, ICLR 2024.
>
> [2] Fairness Guarantees in Multi-class Classiﬁcation with Demographic Parity, JMLR 2024.
>
> [3] Fair Bayes-Optimal Classifiers Under Predictive Parity, NeurIPS 2022.
>
> [4] Addressing Algorithmic Disparity and Performance Inconsistency in Federated Learning, NeurIPS 2021.
>
> [5] FairFed: Enabling Group Fairness in Federated Learning, AAAI 2023.
>
> [6] PraFFL: A Preference-Aware Scheme in Fair Federated Learning, KDD 2025.
>
> [7] LoGoFair: Post-Processing for global and local Fairness in Federated Learning, AAAI 2025.
>
> [8] The Cost of global and local Fairness in Federated Learning, AISTATS 2025.
>
> [9] FACT: A Diagnostic for Group Fairness Trade-offs, ICML 2020.
>
> [10] SCAFFOLD: Stochastic Controlled Averaging for Federated Learning. ICML 2020.
>
> [11] Federated Multi-Objective Learning, NeurIPS 2023.

---

> > ### Author Response · Authors · 2025-08-07
> > **Experimental Results on Gradient Conflicts**
> >
> > Dear Reviewer eRHZ,
> >
> > Thank you again for your valuable feedback. To further clarify the impact of gradient conflicts towards the FedFACT method and present the compatibility of our method with aggregation techniques for mitigating gradient conflict , we conducted experiments on all four datasets. We combined the both proposed methods (FedFACT In & Post) with FedSMGDA proposed in  [1], denoting as MGDA in the following experimental results. Considering a heterogeneous data scenario ($\gamma$ = 0.5) and setting $\delta^g=\delta^l=0.01$ with all other settings consistent to the original study. The results are shown below.
> >
> >
> > |Dataset|Compas| | |Adult| | |CelebA| | |ENEM| | |
> > |---|---|---|---|---|---|---|---|---|---|---|---|---|
> > |**Method**|**Acc**|$\mathscr{D}^{global}$|$\mathscr{D}^{local}$|**Acc**|$\mathscr{D}^{global}$|$\mathscr{D}^{local}$|**Acc**|$\mathscr{D}^{global}$|$\mathscr{D}^{local}$|**Acc**|$\mathscr{D}^{global}$|$\mathscr{D}^{local}$|
> > |FedFACT(In)|61.2±0.7|0.041±0.008|0.073±0.022| 82.0±0.2|0.001±0.000|0.040±0.003|86.2±0.8 | 0.038±0.014 | 0.047±0.019| 68.3±0.4 | 0.049±0.007 | 0.048±0.012|
> > |FedFACT(In)+MGDA|61.6±0.5|0.043±0.013|0.067±0.021|82.5±0.3|0.019±0.005|0.039±0.004|86.8±0.7 | 0.042±0.016|0.045±0.018|68.7±0.3 | 0.052±0.010 | 0.041±0.014 |
> > |FedFACT(Post)|67.3±0.3|0.014±0.004|0.064±0.009|82.7±0.1|0.013±0.002|0.027±0.003|87.1±0.3 | 0.009±0.002 | 0.017±0.004|71.2±0.2 | 0.006±0.001 |  0.020±0.003|
> > |FedFACT(Post)+MGDA|67.5±0.3|0.017±0.005|0.066±0.010|82.9±0.2|0.016±0.004|0.025±0.004|87.2±0.3 | 0.011±0.002 | 0.017±0.004|71.2±0.2 | 0.007±0.001 |  0.018±0.002|
> >
> >
> > The experimental results show that, for the In-processing method, incorporating the MGDA-based aggregation approach leads to improvements in model accuracy, as MGDA aggregation mitigates the impact of gradient conflicts.However, the improvement in fairness is not significant, further illustrating that our proposed method inherently harmonizes the conflicts arising from optimizing both global and local fairness.
> >
> >
> > For the Post-processing method, the improvement with MGDA aggregation is not significant. This is most likely due to the fact that the optimization parameter $\lambda$ is a low-dimensional (<10) vector, which reduces the impact of gradient conflicts, and the post-processing errors are primarily introduced by the sample size rather than optimization, as analyzed in the original paper.
> >
> >
> > These findings provide additional support for our previous analysis in rebuttal: Our framework emphasizes group-fairness calibration without relying on specific aggregation schemes, making it compatible to aggregation methods that reduce the impact of gradient conflicts, while our proposed learning objective inherently harmonizes the conflicts caused by optimizing both global and local fairness.
> >
> > [1]  Federated Multi-Objective Learning, NeurIPS 2023.

---

> > > ### Comment · Reviewer_eRHZ · 2025-08-08
> > >
> > > Thank you for the detailed response. I look forward to the edits in the revised paper.

---

> > > > ### Author Response · Authors · 2025-08-08
> > > >
> > > > Dear reviewer eRHZ,
> > > >
> > > > Sincere appreciation for your valuable feedback! It has greatly contributed to enhancing the quality of our work. As you suggested, we will make sure to incorporate the corresponding discussion and experiments in the revised manuscript, in accordance with the above content about experiments in the multi-class setting, the discussion on gradient conflicts, and the confidence intervals in the experiments.
> > > >
> > > > Thank you once again for your constructive review and support.

---

### Decision · Program_Chairs · 2025-09-17

**Decision:**

Accept (poster)

**Comment:**

This paper presents a novel controllable federated group-fairness calibration framework called FedFACT. It identifies the Bayes-optimal classifiers under both global and local fairness constraints in multi-class case. It balances fairness and accuracy by reformulating fair federated leanring as personalized cost-sensitive learning problem for in-processing, and bi-level optimization for post-processing. Convergence and generalization guarantees for FedFACT are developed. Extensive experimental results show that FedFACT consistently outperforms baselines in balancing accuracy and global-local fairness.

---

Overall, the work is solid and carries substantial merits. All reviewers are on the positive side. I agree with their assessments.

A few issues are raised in the reviews, and authors are encouraged to address them in the final version. In particular, authors are encouraged to improve the readability of certain sections, and discuss possible adaptions of related works on binary classification to the multi-class setting studied in this paper (e.g. what are the challenges, how would the adapted results compare to the present results). It would also be nice to discuss possible integrations of the proposed global fairness and local fairness solutions.